# Towards Understanding Why Mask Reconstruction Pretraining Helps in Downstream Tasks

**Jiachun Pan**[1,2*]        **Pan Zhou**[1*]         **Shuicheng Yan**[1]
[1] Sea AI Lab                    [2]National University of Singapore
pan.jiachun@u.nus.edu                {zhoupan,yansc}@sea.com

## Abstract

For unsupervised pretraining, mask-reconstruction pretraining (MRP) approaches, *e.g.* MAE (He et al., 2021) and data2vec (Baevski et al., 2022), randomly mask input patches and then reconstruct the pixels or semantic features of these masked patches via an auto-encoder. Then for a downstream task, supervised fine-tuning the pretrained encoder remarkably surpasses the conventional "supervised learning" (SL) trained from scratch. However, it is still unclear 1) how MRP performs semantic feature learning in the pretraining phase and 2) why it helps in downstream tasks. To solve these problems, we first theoretically show that on an auto-encoder of a two/one-layered convolution encoder/decoder, MRP can capture all discriminative features of each potential semantic class in the pretraining dataset. Then considering the fact that the pretraining dataset is of huge size and high diversity and thus covers most features in downstream dataset, in fine-tuning phase, the pretrained encoder can capture as much features as it can in downstream datasets, and would not lost these features with theoretical guarantees. In contrast, SL only randomly captures some features due to lottery ticket hypothesis. So MRP provably achieves better performance than SL on the classification tasks. Experimental results testify to our data assumptions and also our theoretical implications.

## 1 Introduction

Self-supervised learning (SSL) has emerged as a popular and effective method to learn unsupervised representations, with great success witnessed by many downstream tasks, *e.g.* image classification (He et al., 2016a), object detection (Girshick et al., 2015; Tan et al., 2020) and segmentation (Ronneberger et al., 2015; He et al., 2017). In SSL, one often needs to first create an artificial supervised learning problem, a.k.a. a pretext task, that can obtain pseudo data labels via well designing the task itself, and then train a network for learning how to capture useful data features from this artificial supervised task. For example, one representative SSL, contrastive learning (He et al., 2020a; Chen et al., 2020b), constructs a supervised problem on an unlabeled dataset via regarding random augmentations of an image as a separate class, and then performs supervised instance discrimination. Owing to the unnecessity of manual annotations and its great success, SSL has already paved a new way to solve unsupervised learning problems, and also has attracted increasing research interests.

In this work, we are particularly interested in the recently proposed mask-reconstruction pretraining (MRP) of SSL families (Xie et al., 2021; Dong et al., 2021), *e.g.* MAE (He et al., 2021) and data2vec (Baevski et al., 2022). The core idea of this MRP family is to randomly mask the patches of the input image and then reconstruct the pixels or semantic features of these masked patches via an auto-encoder. After pretraining on a large-scale unsupervised dataset, MRP fine-tunes the encoder on a specific downstream task to learn more task-specific representations. This pretraining mechanism generally enjoys remarkable test performance improvement on the same downstream task and also a much superior generalization ability on out-of-distribution data than the standard end-to-end "supervised learning". Actually, it also reveals better fine-tuning performance than other state-of-the-art SSL approaches, including contrastive learning (He et al., 2020a; Chen et al., 2020b) and clustering learning (Caron et al., 2018; Wu et al., 2018). Because of its simplicity and strong

---

*Equal contribution. Pan Jiachun did this work during an internship at Sea AI Lab.

compatibility, MRP has attracted wide interests and is seeing increasingly more applications. However, theoretical analyses and understandings on MRP still largely lag their practical applications. To be specific, it is not clear how MRP performs feature learning via the mask reconstruction task, though heavily desired. Moreover, the theoretical reasons for the superiority in test performance of MRP over end-to-end supervised learning are rarely investigated. Most existing theoretical works (Wen & Li, 2021; Arora et al., 2019; Tosh et al., 2021a;b) focus on analyzing contrastive learning, and few works study MRP which differs much from contrastive learning. Cao et al. (2022) analyzed the patch-based attention in MAE via an integral kernel but did not study the core questions in this work, *i.e.*1) what features does MRP learn and 2) why does MRP beat conventional supervised learning.

**Contributions.** In this work, we provide a theoretical viewpoint to understand the semantic (feature) learning process of MRP. Moreover, we analyze test performance of MRP to show its superiority over supervised learning on the downstream classification tasks. Our contributions are highlighted below.

Firstly, based on the multi-view data assumption from (Allen-Zhu & Li, 2020) where multi/single discriminative features exist in multi-view/single-view data, we prove that on an auto-encoder with a two/one-layered convolution encoder/decoder, the pretrained encoder in MRP can capture all the discriminative features of each semantic class in the pretraining dataset. Moreover, a convolution kernel in the encoder captures at most a feature. These properties benefit the downstream tasks. As the pretraining dataset is often much larger than downstream dataset, pretraining dataset (approximately) covers all the features in the downstream dataset. So the kernels of the pretrained encoder also well grab the features in downstream datasets. Besides, as a kernel is associated with at most a feature, then the semantic features would not be fused together, allowing a network to easily establish the relation among kernels and semantic class labels in downstream classification task.

Secondly, we theoretically show that after fine-tuning on the downstream dataset, MRP enjoys superior test performance to that of end-to-end supervised learning on the downstream tasks by using classification as an example. Assuming pretraining and downstream datasets share the same distribution, we prove that after fine-tuning, MRP can classify the new samples correctly with high probability for both multi-view and single-view test data. This result is superior to (Allen-Zhu & Li, 2020), which shows the conventional SL only has a half test accuracy on single-view test data.

## 2 RELATED WORKS

**SSL approaches.** According to the pretext tasks, the current SSL approaches can be grouped into contrastive learning, *e.g.* (Hjelm et al., 2018; Oord et al., 2018), clustering learning, *e.g.* (Caron et al., 2018; Wu et al., 2018) and mask-reconstruction pretraining (MRP) (He et al., 2021; Baevski et al., 2022). Given random augmentations of an image, contrastive learning, *e.g.*, MoCo (He et al., 2020a) and SimCLR (Chen et al., 2020a), brings the different crops of the same image together, and pushes the crops of different images far away from each other in the feature space. For clustering learning, it aims to cluster similar samples into the same group. However, both contrastive learning and clustering learning heavily depend on the multi-crop augmentations. The recently proposed MRP is a simpler SSL. This MRP family, *e.g.* MAE (He et al., 2021) and SimMIM (Xie et al., 2021), randomly masks image patches and then reconstructs the masked patches via an auto-encoder. Later, both MaskFeat (Wei et al., 2021) and data2vec (Baevski et al., 2022) empirically find better performance by reconstructing semantic feature. Now MRP has surpassed the end-to-end supervised learning on many downstream tasks, *e.g.* image classification (Dong et al., 2021) and object detection (He et al., 2021), and is seeing more applications because of its effectiveness and strong compatibility.

**SSL analysis.** Despite its remarkable success in practice, the theoretical understanding of SSL is still largely absent. Arora et al. (2019) provided generalization guarantees for contrastive learning on linear classification models with the assumption that different positives belong to the same latent class. Wang & Isola (2020) showed that contrastive learning can trade-off the alignment and uniformity of features on a hypersphere. HaoChen et al. (2021) proposed and analyzed a spectral version of contrastive loss with provable accuracy guarantees under linear probing evaluation. Tian et al. (2020) proved that SimCLR only captures feature variability across data points. However, these theoretical works mainly study contrastive learning which essentially differs from MRP. The most closely relevant work to ours is (Lee et al., 2021; Cao et al., 2022). Cao et al. (2022) analyzed the patch-based attention in MAE via an integral kernel by showing the benefits of patchifying, the equivalence between the attention mechanism in MAE and a learnable integral kernel transform, *etc*. However, they did not reveal any feature properties of MRP and the superiority reasons of MRP over conventional supervised learning. Lee et al. (2021) showed the benefits of reconstruction partial

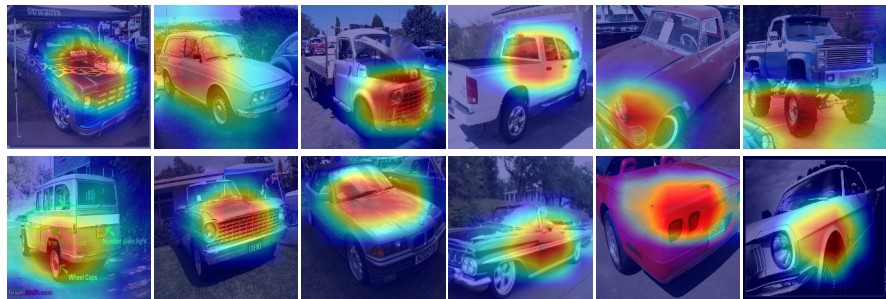

Figure 1: Visualization of ResNet50 (He et al., 2016b) trained by conventional supervised learning. We use Eigen-CAM to localize class-specific image regions which show why the model predicts the image as the corresponding class. Though ResNet50 predicts all car images correctly, it actually locates different regions, *e.g.* front, side window, car nose, taillight, and wheel, for different images, indicating multiple independent features for each class and thus the "multi-view" data assumption.

data from another partial data on reducing sample complexity of the downstream tasks under the condition where two pieces are independent conditioned on their semantic label. But this independent condition is not the real cases where the two parts of the same image will share a significant amount of information not explained by the label (Bansal et al., 2020). Moreover, these works do not study how features are learned by networks, which is essential to understanding MRP in practice.

## 3 PROBLEM SETUP

Here we first introduce "multi-view" data assumption introduced in (Allen-Zhu & Li, 2020), and then present the pretraining framework of mask-reconstruction pretraining (MRP). Finally, following most MRP works, we use a $k$-classification task as a downstream task for analysis. In this work, we use $O, \Omega, \Theta$ to hide constants w.r.t. $k$, and $\tilde{O}, \tilde{\Omega}, \tilde{\Theta}$ to hide polylogarithmic factors w.r.t. $k$. We use $\mathrm{poly}(k)$ ($\mathrm{polylog}(k)$) to denote $\Theta(k^C)$ ($\Theta(\log^C k)$) with constant $C > 0$. $[n]$ denotes $\{1, 2, \ldots, n\}$.

### 3.1 MULTI-VIEW DATA DISTRIBUTION

On realistic data, for each semantic class, there are actually several independent features being in effect during the classification process. As shown in Fig. 1, we adopt Eigen-CAM (Muhammad & Yeasin, 2020) to localize class-specific regions which tell us why a model predicts the image as the corresponding class. Here we test on ResNet50 (He et al., 2016b) trained by the Pytorch Team[1] in a supervised training manner. For all the car images in Fig. 1, though ResNet50 predicts them correctly, Eigen-CAM locates different class-specific regions, *e.g.* car front, side window, taillight, and wheel, on different images. These results directly testify to the multiple independent discriminative features in a semantic class. Such a data structure is called "multi-view data" and is firstly testified in (Allen-Zhu & Li, 2020). In the following, we make the multi-view assumption on the realistic data for analysis. The mathematical formulation is similar to (Allen-Zhu & Li, 2020).

Assume that there are $k$ semantic classes, and each data pair is denoted by $(X, y)$, where $X = (x_1, x_2, \ldots, x_P) \in (\mathbb{R}^d)^P$ has $P$ patches (*e.g.* (non-)overlap image patches) in which each patch is $d$-dimensional, and $y \in [k]$ is the label of $X$. Then suppose there are multiple discriminative features associated with each semantic class. For simplicity, here we say two features and define the two feature vectors as $v_{i,1}, v_{i,2} \in \mathbb{R}^d$ for each class $i \in [k]$. Note, our analysis technique can also be extended to multiple features. We further assume feature vectors are orthonormal, *i.e.*,

$$\forall i, i' \in [k], \quad \forall l, l' \in [2], \quad \|v_{i,l}\|_2 = 1, \quad \text{and} \quad v_{i,l} \perp v_{i',l'}, \text{ when } (i, l) \neq (i', l'). \quad (1)$$

Denote the set of all discriminative features of the $k$ classes as $\mathcal{V} = \{v_{i,1}, v_{i,2}\}_{i=1}^k$.

Now we introduce the multi-view distribution $\mathcal{D}_m$ and single-view distributions $\mathcal{D}_s$, where samples from $\mathcal{D}_m$ have multiple features, samples from $\mathcal{D}_s$ has only a single main feature. Let $C_p$ be a universal constant, $s$ be a universal parameter to control feature sparsity, $\sigma_p = \frac{1}{\sqrt{d}\mathrm{polylog}(k)}$ be a parameter to control magnitude of random noise, and $\gamma$ be a parameter to control the feature noise.

**Definition 3.1** (Multi-view data (Allen-Zhu & Li, 2020))**.** *Data distribution $\mathcal{D}$ consists of data from multi-view data $\mathcal{D}_m$ with probability $1-\mu$ and from single-view data $\mathcal{D}_s$ with probability $\mu$. We define $(X, y) \sim \mathcal{D}$ by randomly uniformly selecting a label $y \in [k]$ and generating data $X$ as follows.*

---

[1] https://pytorch.org/hub/pytorch_vision_resnet/

*1) Sample a set of features $\mathcal{V}'$ uniformly at random from $\{v_{i,1}, v_{i,2}\}_{i \neq y}$ each with probability $\frac{s}{k}$.*

*2) Denote $\mathcal{V}(X) = \mathcal{V}' \cup \{v_{y,1}, v_{y,2}\}$ as the set of feature vectors used in data $X$.*

*3) For each $v \in \mathcal{V}(X)$, pick $C_p$ disjoint patches in $[P]$ and denote it as $\mathcal{P}_v(X)$ (the distribution of these patches can be arbitrary). We denote $\mathcal{P}(X) = \cup_{v \in \mathcal{V}(X)} \mathcal{P}_v(X)$.*

*4) If $\mathcal{D} = \mathcal{D}_s$ is the single-view distribution, pick a value $\hat{l} = \hat{l}(X) \in [2]$ uniformly at random.*

*5) For each $p \in \mathcal{P}_v(X)$ for some $v \in \mathcal{V}(X)$, given feature noise $\alpha_{p,v'} \in [0, \gamma]$, we set*

$$x_p = z_p v + \sum_{v' \in \mathcal{V}} \alpha_{p,v'} v' + \xi_p,$$

*where $\xi_p \in \mathcal{N}(0, \sigma_p \mathbf{I})$ is an independent random Gaussian noise. The coefficients $z_p \geq 0$ satisfy*

- *For "multi-view" data $(X, y) \in \mathcal{D}_m$, $\sum_{p \in \mathcal{P}_v(X)} z_p \in [1, O(1)]$ and $\sum_{p \in \mathcal{P}_v(X)} z_p^q \in [1, O(1)]$ for an integer $q \geq 2$, when $v \in \{v_{y,1}, v_{y,2}\}$, $z_p$ is uniformly distributed over $C_p$ patches and the marginal distribution of $\sum_{p \in \mathcal{P}_v(X)} z_p$ is left-close.*

- *For "single-view" data $(X, y) \in \mathcal{D}_s$, when $v = v_{y,\hat{i}}$, $\sum_{p \in \mathcal{P}_v(X)} z_p \in [1, O(1)]$, $\sum_{p \in \mathcal{P}_v(X)} z_p^q \in [1, O(1)]$ for $q \geq 2$. When $v = v_{y,3-\hat{i}}$, $\sum_{p \in \mathcal{P}_v(X)} z_p \in [\rho, O(\rho)]$ (here we set $\rho = k^{-0.01}$ for simplicity). $z_p$ is uniformly distributed over $C_p$ patches.*

- *$\sum_{p \in \mathcal{P}_v(X)} z_p \in [\Omega(1), 0.4]$ when $v \in \mathcal{V}(X) \setminus \{v_{y,1}, v_{y,2}\}$, and the marginal distribution of $\sum_{p \in \mathcal{P}_v(X)} z_p$ is right-close.*

*6) For each $p \in [P] \setminus \mathcal{P}(X)$, with an independent random Gaussian noise $\xi_p \sim \mathcal{N}(0, \frac{\gamma^2 k^2}{d} \mathbf{I})$, we set*

$$x_p = \sum_{v' \in \mathcal{V}} \alpha_{p,v'} v' + \xi_p,$$

*where each $\alpha_{p,v'} \in [0, \gamma]$ is the feature noise.*

Intuitively, multi-view data $\mathcal{D}_m$ refers to the data with multiple features distributed over patches plus some noise from other features and background noise, while only a single main feature exists in single-view data $\mathcal{D}_s$. Their mixed distribution $\mathcal{D}$ can well characterize realistic data. Based on distribution $\mathcal{D}$, in Sec. 4 we will define the datasets used for pretraining and downstream fine-tuning.

### 3.2 Mask-Construction Pretraining Framework

As a representative MRP, MAE (He et al., 2021) randomly masks the patches of an input image and then reconstructs the pixels of these masked patches via an auto-encoder. Recently, many works show that reconstructing the semantic features often achieves higher performance, where the semantic feature can be obtained by feeding the vanilla full input into a teacher network, *e.g.* a pretrained network (Wei et al., 2021) or the exponential moving average (EMA) of encoder in MAE (Dong et al., 2021; Baevski et al., 2022). In this paper, we analyze both Teacher-Student framework and MAE but will focus more on the former one because of its slightly higher performance.

**Network Architectures.** Formally, as shown in Fig 2, we implement the encoder in student network by a two-layer convolution smoothed ReLU network with $km$ kernels denoted by $w_r \in \mathbb{R}^d, r \in [km]$. for the encoder, its output is defined as

$$H(X) = [h_1(X), h_2(X), \dots, h_{km}(X)], \qquad \text{where} \quad h_r(X) = \sum_{p \in [P]} \overline{\text{ReLU}}(\langle w_r, x_p \rangle).$$

Here $\overline{\text{ReLU}}$ is a smoothed ReLU (Allen-Zhu & Li, 2020) and is defined as follows: for an integer $q \geq 2$ and a threshold $\varrho = \frac{1}{\text{polylog}(k)}$, $\overline{\text{ReLU}}(z) = 0$ if $z \leq 0$, $\overline{\text{ReLU}}(z) = \frac{z^q}{q \varrho^{q-1}}$ if $z \in [0, \varrho]$ and $\overline{\text{ReLU}}(z) = z - (1 - 1/q)\varrho$ if $z \geq \varrho$. The desirable properties of smoothed ReLU function is that when $z$ is large it is linear with $z$ and when $z$ is small, it will be much smaller. Thus, it will make the low-magnitude feature noises much smaller to better separate the true features from feature noises. The decoder is a linear layer parameterized by $b_r$ ($r \in [km]$), and its output is

$$\boldsymbol{h}'(X) = [h_1'(X), h_2'(X), \dots, h_{km}'(X)], \qquad \text{where} \quad h_r'(X) = b_r h_r(X), \quad r \in [km].$$

Following the practice in MRP (Baevski et al., 2022; Dong et al., 2021), teacher network shares the same architecture with student network, and is a smoothed ReLU network parameterized by $\hat{w}_r, r \in [km]$. Its output is defined as

$$\boldsymbol{h}(X) = [\hat{h}_1(X), \hat{h}_2(X), \dots, \hat{h}_{km}(X)], \qquad \text{where} \quad \hat{h}_r(X) = \sum_{p \in [P]} \overline{\text{ReLU}}(\langle \hat{w}_r, x_p \rangle).$$

**Pretraining of MRP on Pretext Task.** Now we define the pretraining loss. Let $\boldsymbol{\epsilon} = (\epsilon_1, \epsilon_2, \dots, \epsilon_p)$, where $\epsilon_i$ is an independent Bernoulli variable with $\Pr(\epsilon_i = 1) = \theta$. In the pretraining, for the linear

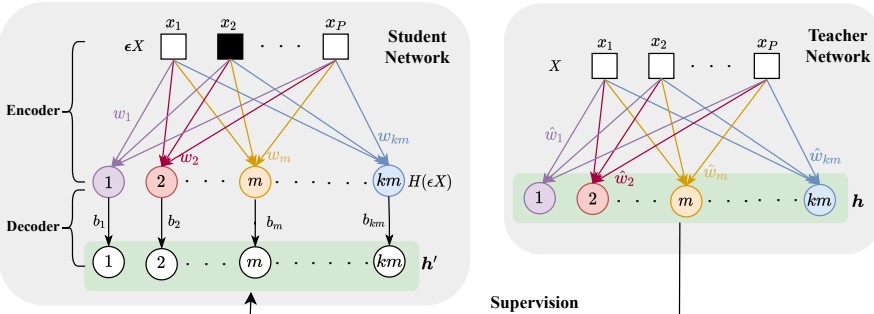

Figure 2: **Teacher-Student framework** studied in this work. Given an input $X = [x_1, \ldots, x_P]$ (image or text tokens) with $P$ patches, this framework randomly masks patches to obtain $\epsilon X = [\epsilon_1 x_1, \ldots, \epsilon_p x_P]$ with Bernoulli variable $\epsilon_p$ to mask, and feeds $\epsilon X$ into student encoder $H$ for a latent vector $H(\epsilon X)$. Then, student decoder takes $H(\epsilon X)$ as input and outputs $\boldsymbol{h}'$ of all patches to predict the output $\boldsymbol{h}$ of a teacher with vanilla input $X$ as input. The encoder is two-layer CNN, and the decoder is a linear layer. For **MAE framework**, it has encoder-decoder networks and the decoder have an additional layer to map output of the encoder to recover $P$ patches (see Fig 6 in Appendix).

student decoder, we set its all parameters as $b_r = c(\theta) = \frac{1}{\theta}$ for simplicity, which is provably sufficient for improving downstream tasks. Now we define the empirical mean squared pretraining loss:

$$L(H; \boldsymbol{\epsilon}) = \frac{1}{2N} \sum_{n \in [N]} L(H; X_n, \boldsymbol{\epsilon}) = \frac{1}{2N} \sum_{n \in [N]} \sum_{r \in [km]} \|\hat{h}_r(X_n) - h'_r(\boldsymbol{\epsilon} X_n)\|_2^2,$$

where $N$ is the number of data points for pretraining and $\boldsymbol{\epsilon} X = (\epsilon_1 x_1, \epsilon_2 x_2, \ldots, \epsilon_P x_P)$.

Now we discuss how to pretrain it. Following MRP (Baevski et al., 2022; Dong et al., 2021), we use student to update teacher by updating teacher kernel parameters $\hat{w}_r$ as $\hat{w}_r^{(t)} = \tau w_r^{(t)}$, where we set $\tau = 1 + c_0$ and $c_0 = \frac{1-\theta}{C_p \theta} + \Theta\left(\frac{1}{t+1}\right)$. Then we use gradient descent to update student encoder parameters:

$$w_r^{(t+1)} = w_r^{(t)} - \eta \mathbb{E}_{\boldsymbol{\epsilon}} \left[ \nabla_{w_r} L(H; \boldsymbol{\epsilon}) \right]. \tag{2}$$

**Fine-tuning of MRP on Classification Downstream Tasks.** Here we consider a classification downstream task. Specifically, we fine-tune the pretrained student encoder with an extra linear layer using $N_2$ labeled samples. We fine-tune the network by minimizing the empirical cross-entropy loss:

$$L_{\text{down}}(F) = \frac{1}{N_2} \sum_{n \in [N_2]} L_{\text{down}}(F; X_n, y_n), \quad \text{where} \quad F_i(X) = \sum_{r \in [km]} u_{i,r} h_r(X), i \in [k].$$

Here $L_{\text{down}}(F; X, y) = -\log \frac{e^{F_y(X)}}{\sum_{j \in [k]} e^{F_j(X)}}$, and $u_{i,r}, r \in [km], i \in [k]$ denotes the weights of the extra linear layer. Then we adopt the gradient descent to fine-tune the kernels $w_r$ of the pretrained encoder and update the parameters $u_{i,r}$:

$$w_r^{(t+1)} = w_r^{(t)} - \eta_1 \nabla_{w_r} L_{\text{down}}(F), \quad u_{i,r}^{(t+1)} = u_{i,r}^{(t)} - \eta_2 \nabla_{u_{i,r}} L_{\text{down}}(F),$$

where the learning rate $\eta_1$ is often much smaller than $\eta_2$ in practice.

## 4 MAIN RESULTS

Here we first reveal the semantic feature learning process of mask-reconstruction pretraining (MRP), and then theoretically show why MRP helps downstream tasks by taking the classification task as an example. Finally, we intuitively discuss the benefits of MRP to other downstream tasks.

### 4.1 FEATURE LEARNING PROCESS OF PRETRAINING

Here we mainly show that pretraining can capture the whole features $\mathcal{V}$ (defined in (1)) in the pretraining dataset by showing that the correlation scores between the features and the kernels of the student encoder gradually increase during training process. For brevity, we first define

$$\mathcal{M}_{i,l}^{(0)} := \left\{ r \in [km] : \langle w_r^{(0)}, v_{i,l} \rangle \geq \Lambda_{i,l}^{(0)} \left(1 - O(1/\log k)\right) \right\}, \quad \text{where} \quad \Lambda_{i,l}^{(t)} := \max_{r \in [km]} [\langle w_r^{(t)}, v_{i,l} \rangle]^+.$$

Here $w_r^{(t)}$ denotes the $r$-th convolution kernel of the student encoder at the $t$-th iteration. $\Lambda_{i,l}^{(t)}$ denotes the highest positive correlation score between the $l$-th feature $v_{i,l}$ of the $i$-th class and all the $km$ kernels $w_r^{(t)}$. Larger $\Lambda_{i,l}^{(t)}$ means the network can better capture the feature $v_{i,l}$. For $\mathcal{M}_{i,l}^{(0)}$, it is

composed of the kernels which have slightly smaller correlation scores than the maximum score $\Lambda_{i,l}^{(0)}$ at the initial stage. For analysis, we pose some assumptions on data and the network as follows.

**Assumption 1.** *(1) The pretraining dataset $\mathcal{Z}$ have $N$ samples which are i.i.d. drawn from the distribution $\mathcal{D}$ defined in Definition 3.1 and let $N \geq \mathrm{poly}(k)$.*

*(2) Each kernel $w_r^{(0)}(r \in [km])$ is initialized by a Gaussian distribution $\mathcal{N}(0, \sigma_0^2 \mathbf{I})$ with $\sigma_0 = O(1/\sqrt{k})$. Moreover, $m$ satisfies $m \in [\mathrm{polylog}(k), \sqrt{k}]$.*

Assumption 1 means that there are about $(1 - \mu)N$ "multi-view" samples and $\mu N$ "single-view" data points in the pretraining dataset $\mathcal{Z}$. According to Definition 3.1, a multi-view sample contains multiple discriminative features distributed over patches plus some noise from other features and background noise, while for a single-view sample, it has only a single main feature and some noises. We use Gaussian initialization as it is the standard initialization used in practice. Note, for pretraining, we do not use any labels. Theorem 1 states the feature learning process in MRP.

**Theorem 1.** *Suppose Assumption 1 holds, learning rate $\eta \leq \frac{1}{\mathrm{poly}(k)}$ in gradient decent steps (2). After $T = \frac{\mathrm{poly}(k)}{\eta}$ iterations, for sufficiently large $k$, the learned kernels $\{w_r^{(T)}\}_{r \in [km]}$ satisfy the following properties with high probability.*

*1) **Under Teacher-Student framework**, when $q \geq 3$, for every $v_{i,l} \in \mathcal{V}$ and every $(X, y) \in \mathcal{Z}$,*

*(a) $\Lambda_{i,l}^{(0)} \in [\tilde{\Omega}(\sigma_0), \tilde{O}(\sigma_0)]$, $\Lambda_{i,l}^{(T)} \in [1/\mathrm{polylog}(k), \tilde{O}(1)]$ and $r^* \in \mathcal{M}_{i,l}^{(0)}$, where $r^* = \mathrm{argmax}_{r \in [km]}[\langle w_r^{(T)}, v_{i,l} \rangle]^+$.*

*(b) For each $r \in \mathcal{M}_{i,l}^{(0)}$, $\langle w_r^{(T)}, v_{i',l'} \rangle \leq \tilde{O}(\sigma_0)$ when $(i, l) \neq (i', l')$.*

*(c) For each $r \notin \mathcal{M}_{i,l}^{(0)}$, $\langle w_r^{(T)}, v_{i,l} \rangle \leq \tilde{O}(\sigma_0)$.*

*2) **Under MAE framework**, when $q \geq 4$, the properties (a)-(c) also hold.*

See the proofs of Teacher-Student framework in Appendix F and of MAE framework in Appendix H. Theorem 1 states that for both frameworks, the pretrained model can capture all features. But MAE needs slightly restrictive assumption, since 1) it requires $q \geq 4$ where $q$ is the smooth parameter in ReLU (see Sec. 3.2), and 2) larger $q$ compresses small feature noises more heavily to better separate the true features from feature noises. This functionality is implemented by teacher in Teacher-Student framework, as teacher can filter out feature noise and gives more clean targets.

Theorem 1 (a) shows that for those kernels winning the lottery ticket at the random initialization stage (*i.e.* kernels $w_r^{(0)} \in \mathcal{M}_{i,l}^{(0)}$), at least one of them would win out through the course of training and capture the feature $v_{i,l}$. Specifically, at initialization, for any feature $v_{i,l}$ in the whole features $\mathcal{V}$ of the pretraining dataset, its correlation score $\Lambda_{i,l}^{(0)}$ with any kernel $w_r$ in $\{w_r\}_{r=1}^{mk}$ is at most $\tilde{O}(1/\sqrt{k})$. After MRP pretraining, for any feature $v_{i,l}$, there always exists at least a kernel $w_r^{(T)}$ so that the correlation score $\Lambda_{i,l}^{(T)}$ between $v_{i,l}$ and $w_r^{(T)}$ is increased to at least $1/\mathrm{polylog}(k)$. So each feature in $\mathcal{V}$ is captured by at least a convolution kernel in the student encoder. Besides, as the pretraining dataset is often much larger than the downstream dataset, the features in pretraining dataset actually (approximately) cover all the features in downstream dataset. So the kernels of the pretrained student encoder is able to capture as much features as possible in downstream datasets.

For Theorem 1 (b) and (c), they mainly guarantee some kinds of corresponding relations among kernels and features: *a kernel captures at most a feature*. Specifically, from Theorem 1 (b) indicates that for these kernels $w_r$ in $\mathcal{M}_{i,l}^{(0)}$ which mainly capture the semantic feature $v_{i,l}$, they actually only capture little information of other features $v_{i',l'}$ where $v_{i',l'} \neq v_{i,l}$, since for any $w_r \in \mathcal{M}_{i,l}^{(0)}$, its correlation score with $v_{i',l'}$ is no larger than $\tilde{O}(\sigma_0)$ and keeps small during the training phase. Theorem 1 (c) shows that for these kernels $w_r \notin \mathcal{M}_{i,l}^{(0)}$, they keep losing the lottery ticket during training, and only capture little information of feature $v_{i,l}$. Theorem 1 (b) and (c) together guarantee that a kernel mainly captures at most a feature and can only grab very little information of other features. So the multiple features captured by the encoder kernels is separated and not involved with each other. This property is very important for fine-tuning, since intuitively, a kernel is only associated with at most a feature, and accordingly, a linear classifier can directly establish the relations among kernels and semantic class labels. See more discussion in Sec. 4.2.

### 4.2 BENEFIT JUSTIFICATION OF MRP ON DOWNSTREAM TASKS

**Classification Downstream Task.** Here we first analyze the performance of MRP on classification downstream task. After pretraining, following the practice in (Wei et al., 2021; Dong et al., 2021; He et al., 2021; Xie et al., 2021; Baevski et al., 2022), we only fine-tune the student encoder with an extra linear layer on the labeled training data of the downstream datasets. See the details of fine tuning in Sec. 3.2. Before analysis, we first make some mild assumptions.

**Assumption 2.** *(1) The downstream dataset $\mathcal{Z}_{\text{down}}$ of $N_2$ samples is i.i.d. drawn from the distribution $\mathcal{D}$ defined in Definition 3.1. Let $N_2 \geq k$.*
*(2) We initialize $u_{i,r}^{(0)}, i \in [k], r \in [km]$ by 0 and initialize $w_r^{(0)}$ by the pretrained encoder $w_r^{(T)}$.*

Assumption 2 actually assumes the pretraining and downstream datasets share the same distribution $\mathcal{D}$. This data assumption accords with the practice in many SSL works (Wei et al., 2021; Dong et al., 2021), *e.g.* MAE (He et al., 2021), SimMIM (Xie et al., 2021) and data2vec (Baevski et al., 2022), which pretrain and fine-tune on the same dataset, *e.g.* ImageNet, but with significant improvement over the conventional supervised learning. Then based on Theorem 1, we analyze the test performance on the classification downstream task, and summarize the results in Theorem 2. We denote the fine-tuning network as function $F(\cdot) \in \mathbb{R}^k$ which outputs $k$-dimensional prediction for $k$ classes.

**Theorem 2** (Test performance analysis). *Suppose Assumption 2 holds. When $F(\cdot)$ is either the student encoder in **Teacher-Student framework** or the encoder in **MAE** with an extra linear layer, by fine-tuning $F(\cdot)$ with $N_2$ labeled samples, for any new data point $(X, y) \sim \mathcal{D}$, $F(\cdot)$ satisfies*

$$\Pr_{(X,y)\sim\mathcal{D}} \left[ F_y(X) \geq \max_{j \neq y} F_j(X) + \tilde{O}(1) \right] \geq 1 - e^{-\Omega(\log^2 k)},$$

*where $F_y(X)$ denotes the $y$-th element in $F(X)$, i.e. the predicted probability for the class $y$.*

See its proof in Appendix G. Theorem 2 guarantees that no matter for single-view or multi-view data $(X, y) \sim \mathcal{D}$, the fine-tuned classifier $F(\cdot)$ always correctly predicts the label $y$ with high probability. This is because intuitively, as proved in Theorem 1 (a), after pretraining, for each discriminative feature $v_{i,l}$ in the feature set $\mathcal{V}$, at least a kernel $w_r$ in the pretrained student encoder can capture it. This means that even at the beginning of the fine tuning, the encoder in the function $F(\cdot)$ is already capable to discover and grab all features in $\mathcal{V}$. Then as shown in Theorem 1 (b) and (c), a kernel captures at most a feature. In this way, for single-view sample containing a single feature denoted by $v_{i,\hat{i}}$, the corresponding kernels in the encoder would capture it and output a large correlation score ( $\geq 1/\text{polylog}(k)$) at the corresponding positions and small correlation scores ( $\leq \tilde{O}(1/\sqrt{k})$) at the remaining positions. Similarly, for multi-view samples including several features, the corresponding kernels have large correlation scores at some specific kernels while small ones for remaining kernels. For a class, these positions of high scores would not change, because all features are captured and a kernel grabs at most a feature. Based on this, the last linear layer in $F(\cdot)$ can easily establish the corresponding relation between large score positions and feature labels, and learns to classify.

Then we compare the test performance with conventional end-to-end "supervised learning" (SL) on the same downstream dataset. Under the same data distribution and the same network $F$, Allen-Zhu & Li (2020) analyzed the test performance of SL (see Lemma 1).

**Lemma 1.** *Suppose the data assumption in Assumption 2 holds and now let sample number $N_2 \geq \text{poly}(k)$. Let the learning rate $\eta \leq \frac{1}{\text{poly}(k)}$. Then by training $F$ after $T = \frac{\text{poly}(k)}{\eta}$ iterations with supervised training, with probability $\geq 1 - e^{-\Omega(\log^2 k)}$, the supervised trained model $F_{\text{SL}}^{(T)}$ satisfies*

$$\Pr_{(X,y)\sim\mathcal{D}} \left( \exists i \in [k] : F_{\text{SL},y}^{(T)}(X) < F_{\text{SL},i}^{(T)}(X) \right) \in [0.49\mu, 0.51\mu].$$

Lemma 1 shows that the supervised trained model $F_{\text{SL}}^{(T)}$ has only about 50% accuracy on single-view data whose ratio among all data is $\mu$. Both our Theorems and Lemma 1 are proved using lottery ticket hypothesis (Allen-Zhu & Li, 2020; Wen & Li, 2021; Allen-Zhu & Li, 2022; Frankle & Carbin, 2019) and we discuss the detailed difference between our work and (Allen-Zhu & Li, 2020) in terms of network architecture, objective loss and fine-tuning on downstream tasks in Appendix C. For supervised training, when the convolution kernels are randomly initialized, for each class, a feature correlates more with kernels than other features. With more training iterations, this feature become a winning lottery over other features. Thus, for those single-view data without the captured features, the classification will make an error and obtain only half accuracy for total dataset. By comparison, in MRP including both teacher-student framework and MAE, all features are captured in the pretraining,

and thus has stronger capacity to capture more kinds of features for each class. This also accords with our empirical observations in Sec. 5. Specifically, Fig. 3 visualizes the class-specific image regions for both models trained by SL and MRP. By comparison, MRP captures multiple discriminative features in an image, while SL only captures single feature.

**Discussion on Other Downstream Tasks.** Besides classification downstream task, our conclusion could intuitively generalize to other downstream tasks, e.g. transfer learning and detection, because in the pretraiing phase, our encoder have provably captured all feature features in each images.

For transfer learning, the representative task is classification task $\mathcal{T}_{cls}$ (He et al., 2020b; 2021) which pretrains a model on a large-scale unlabeled data $\mathcal{D}_{pre}$ and then fine-tunes the pretrained model on a classification downstream dataset $\mathcal{D}_{fine}$. Denote the feature set of dataset $\mathcal{D}_{fine}$ as $\mathcal{V}'$. We discuss this transfer learning in three cases: 1) the datasets $\mathcal{D}_{fine}$ and $\mathcal{D}_{pre}$ share the same feature set $\mathcal{V}'$ (*i.e.* $\mathcal{V}' = \mathcal{V}$) but can have different data distribution (*e.g.* different ratio of single- and multi-view data); 2) $\mathcal{V} \subset \mathcal{V}'$; 3) the pretraining and downstream tasks share the partial semantic set, i.e., $\mathcal{V} \cap \mathcal{V}' \neq \emptyset$. For the case (1) and (2), following the similar proof process of Theorem 2, the fine-tuned model can also obtain high classification accuracy on downstream tasks. For the case (3), as our training networks are over-parameterized, i.e., $m \in [\mathrm{polylog}(k), \sqrt{k}]$ and the size of the lottery ticket winning set is $O(\mathrm{polylog}(k))$ at the end of pre-training, the number of kernel weights that finally capture the features is much smaller than the total number of kernels. For the remaining kernels, they still can capture new features in the downstream fine-tuning phase. In this way, MRP still improves the overall transfer learning performance. The experimental results on transfer learning in Table 1 also support our above analysis. While the pretraining dataset, i.e. ImageNet, and the downstream dataset, VOC07, share many different categories and thus have different features, MRP still performs better than the supervised case, which validates our analysis.

For object detection downstream task, it has two targets: 1) finding the bounding boxes of possible objects, and 2) classifying bounding boxes. For bounding box detection, since a) the encoder pretrained by MRP can grab all features and b) the desired bounding boxes should contain features, the encoder can detect the bounding boxes precisely, at least does not lose many bounding boxes of objects. In contrast, supervised learning often randomly captures feature features (Allen-Zhu & Li, 2020) and thus cannot well detect bounding boxes. For the second target, *i.e.* classification, we can draw similar results on the transformer learning classification task that MRP often performs better than supervised learning. So by considering the advantages of MRP over supervised learning on the two targets in object detection, MRP should expect better performance than supervised learning.

**Comparison to Other SSL Analysis.** Only a few works have studied MRP. Cao et al. (2022) revealed the benefits of patchifying and the equivalence between the attention mechanism in MAE, *etc*. But they did not analyze any feature learning process of MRP and the superiority reasons of MRP over supervised learning. Lee et al. (2021) showed that by splitting an input into two pieces, the pretext tasks of reconstructing one piece from another piece can decrease the sample complexity of the downstream tasks. Though promising, they require the two pieces to be approximately independent conditioned on their feature label, which contradicts many realistic cases where the two parts of the same image share a significant amount of information not explained by the label (Bansal et al., 2020). In contrast, under multi-view data assumptions verified by (Allen-Zhu & Li, 2020) and our experimental results, we reveal how features are learned by MRP, and also explicitly show the performance improvement on downstream tasks, which is essential to understanding MRP in practice.

**Discussion on CNN Architectures.** We analyze CNNs because of two reasons. One is for easy comparison with supervised results of same encoder network in (Allen-Zhu & Li, 2020). Another one is the difficulty of analyzing the feature learning process of Transformer due to its correlated manipulations and highly nonlinear attentions. But this work is the first one that analyzes the feature learning process of MRP on explicit non-linear networks. Moreover, as shown in (Jing et al., 2022; Fang et al., 2022) (see our Appendix B) and our Sec. 5, CNNs pretrained by MRP empirically surpass supervised one on various downstream tasks, *e.g.* classification, detection and segmentation. Finally, Appendix B shows that Transformers pretrined by MRP also captures more features than supervised one which accords with our observations on CNNs and validates our theory.

Table 1: Performance on Various Downstream Tasks. We use official MRP setting to train ResNet50.

| Downstream Tasks | Classification Acc. (%) on ImageNet | Detection AP$_{75}$ on VOC07+12 | Transfer learning Classification Acc. (%) on VOC07 |
|---|---|---|---|
| Supervised Training | 76.2 | 58.8 | 87.5 |
| MRPs | 78.0 | 63.2 | 91.1 |

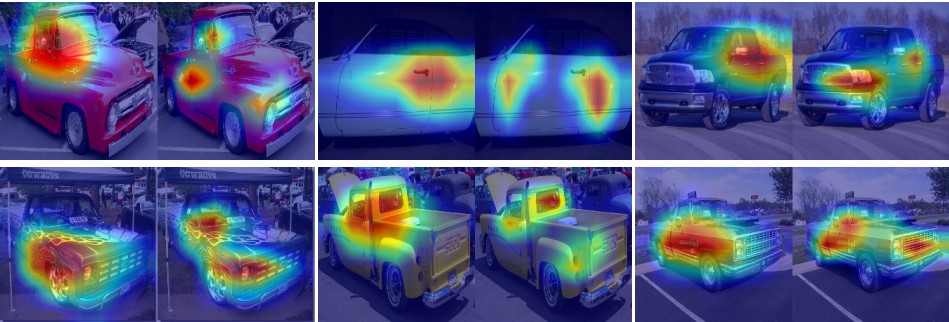

Figure 3: Visualization of ResNet50 (He et al., 2016b) respectively trained by supervised learning and MRP. We use Eigen-CAM to localize class-specific image regions. For each pair, the left figure is given by the supervised model, while the right figure comes from the pretrained model. By comparison, the pretrained model often captures more kinds of features than the supervised model.

## 5 EXPERIMENTS

**Assumption Investigation.** To verify our "multi-view" data assumption, we investigate whether there are multiple discriminative features for some classes in ImageNet (Deng et al., 2009). To this end, we use the widely used Eigen-CAM (Muhammad & Yeasin, 2020) to visualize which part of an image plays a key role in deciding its predicted class. We follow the default setting in Eigen-CAM and use the network parameters of the forth block to compute the project of an image in ResNet50 (He et al., 2016b) released by PyTorch Team[1]. As shown in Fig. 1 of Sec. 3.1, though ResNet50 predicts all the car images correctly, Eigen-CAM locates different class-specific regions, *e.g.* car front, side window, car nose, taillight, and wheel, for different images. It indicates the existence of multiple independent discriminative features in a semantic class and validates our "multi-view" data assumption.

**Results on CNN.** We investigate the performance of MRP on CNNs. We use the recently proposed SimMIM (Xie et al., 2021), a representative MRP, on ResNet50. We use SimMIM rather than MAE (He et al., 2021), as MAE removes the masked patches before encoder but the convolution operations in CNN encoder cannot handle masked input, while SimMIM replaces the masked patches by a mask token and can use CNNs. Then we use ResNet50 (4-layered transformer) to implement the encoder (decoder). Next, we pretrain for 300 epochs on ImageNet, and fine-tune pretrained ResNet50 for 100 epochs on ImageNet. Table 1 reports the top-1 accuracy on ImageNet, and shows that on ResNet50, MRP improves supervised training by a large margin. Moreover, we fine-tune our pretrained ResNet50 on transfer learning classification downstream task on VoC07 and detection task on VoC07+12. The results show that CNNs pretrained by MRP generalizes well on various downstream tasks and indeed often surpasses supervised baselines. These results accord with our theoretical implications that MRP can help downstream tasks by enjoying superior performance than conventional SL. See more results on downstream tasks in Appendix B.

Finally, we use Eigen-CAM to localize class-specific image regions for both models trained by supervised learning (SL) and MRP. For each pair in Fig. 3, the left image is the visualization of SL, while the right one is from MRP. By comparison, MRP often captures several discriminative features in an image, *e.g.* front window and door handle in the first pair, while SL only grabs a feature, *e.g.* front window in the first pair. *See similar observations on transformer in Appendix B.* These results accord with our theory that the advantages of MRP come from its stronger capacity to capture more kinds of class features in the pretraining.

## 6 CONCLUSION

In this work, we analyze the feature learning process of mask-reconstruction pretraining (MRP) and show its superiority in downstream tasks. For pretraining, the pretrained encoder in MRP provably captures all discriminative features in the pretraining dataset. Because of its huge size and high diversity, the pretraining dataset covers the features in downstream dataset. So for fine-tuning, the encoder well grabs as much features as it can in downstream datasets, while supervised learning only randomly captures some features due to its random initialization. Thus, the fine-tuned encoder in MRP provably achieves better accuracy than supervised learning on the classification tasks. Experimental results validate our assumptions and also our theoretical implications.

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

## A  OUTLINE OF APPENDIX

This supplementary document contains the main proofs for two main theorems. It is structured as follows. Appendix B first provides more experimental results to compare the learnt features between the conventional supervised learning and the mask-reconstruction pretraining. We also provide the results on other downstream tasks under backbone. Moreover, we also visualize the results under Transformer backbone. In Appendix C, we present the necessary assumptions and main results on feature learning process of mask-reconstruction pretraining. In Appendix D, we introduce the main ideas to prove our main results. In Appendix E, we show some technical results. Then we prove Theorem C.4 in Appendix F. Finally, we show the performance on downstream classification tasks (proof of Theorem G.2) in Appendix G. We prove the similar results under MAE framework and have a discussion on BEiT in Appendix H.

## B  MORE EXPERIMENTAL RESULTS AND DETAILS

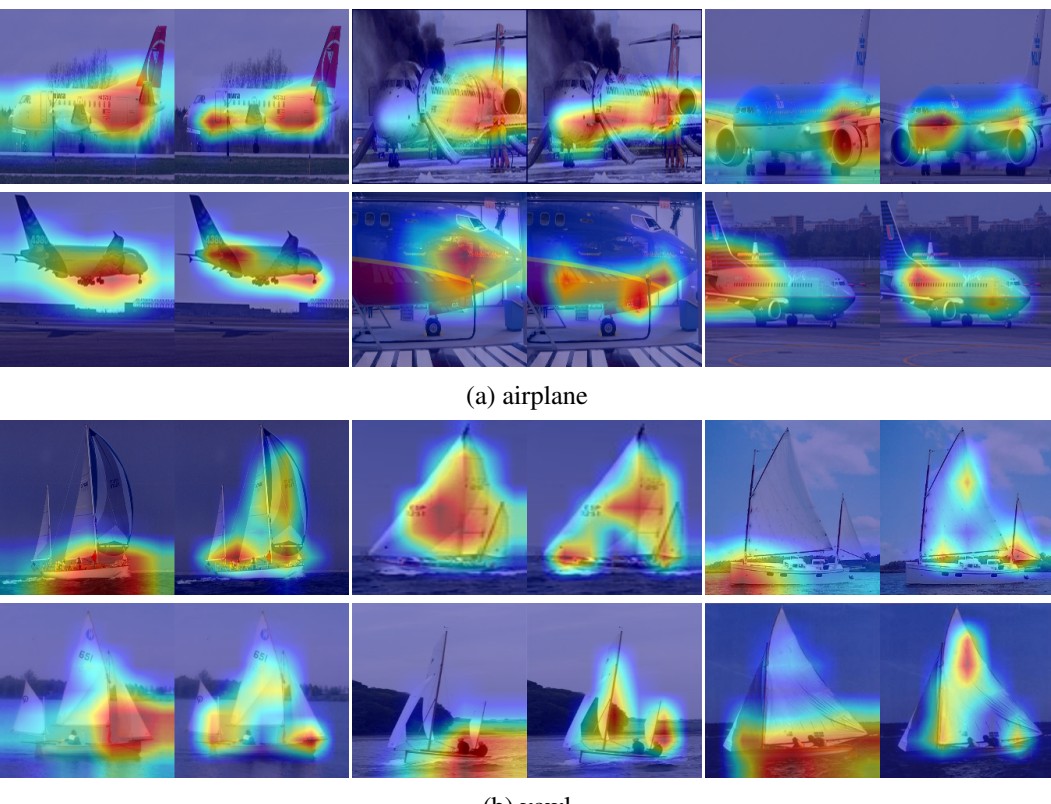

(a) airplane

(b) yawl

Figure 4: Class-specific visualization of ResNet50 (He et al., 2016b) trained by MRP. For each pair, the left figure is given by supervised model, while the right figure comes from the pretrained model. By comparison, pretrained model often captures more kinds of features than supervised model.

In this section, we provide more visualizations to compare the learnt features by conventional supervised learning (SL) and the mask-reconstruction pretraining (MRP). Moreover, we provide more results on other downstream tasks.

**More visualization results.** Same as in the manuscript (Sec. 5), here we use Eigen-CAM to localize class-specific image regions for both models trained by SL and MRP. Note, since 1) CAM-alike methods all need to a well-trained classifier at the top of the backbone, and 2) the pretrained model for MRP has only encoder backbone but no a classifier, we visualize the fine tuned model instead of pretrained model for MRP.

For each pair in Fig. 4, the left image is the visualization of SL, while the right one is from MRP. By comparison, MRP can often capture several discriminative features in an image, *e.g.* the airplane

head and airplane tail in the first pair, while SL only captures a feature, *e.g.* airplane tail in the first pair. These results also accord with our theory that the advantages of MRP come from its stronger capacity to capture more kinds of features for each class in the pretraining phase.

**Results on other downstream tasks.** Actually, some works, *e.g.* (Jing et al., 2022; Fang et al., 2022), also empirically find that CNNs pretrained by MRP can generalize well on various downstream tasks, e.g., detection and segmentation. Specifically, Fang et al. (2022) showed that on ResNet50, MRP achieves 38.0 mIoU on ADE20K semantic segmentation task and greatly improves the 36.1 mIoU of supervised learning baseline. See these results in its Table 3 (b). Indeed, Table 4 in work (Jing et al., 2022) also demonstrates that on VoC07+12 detection task, COCO detection task and COCO instance segmentation tasks, ResNet50 pretrained by MRP respectively achieves 64.4 $AP_{75}$, 42.1 $AP_{75}^{bb}$ and 36.4 $AP_{75}^{mk}$, while supervised ResNet50 respectively achieves 58.8 $AP_{75}$, 41.2 $AP_{75}^{bb}$ and 35.2 $AP_{75}^{mk}$.

**Visualization under Transformer backbone.** Besides ResNet, we further provide visualization results on Transformer (Dosovitskiy et al., 2020) to display the localize class-specific image regions for both models trained by SL and MRP. For each group in Fig. 5, the left image is the visualization of SL, while the middle one is from MAE (He et al., 2021) and the right one is from data2vec (Baevski et al., 2022). Here we directly use the official released ViT-base models (Dosovitskiy et al., 2020) of SL[2], MAE[3] and data2vec[4].

Then one can observe that both MAE and data2vec usually capture several discriminative features in an image, while SL only captures a feature. For example, in the first comparison group, SL only captures one side of car tail. In contrast, MAE grabs two sides of car tail, and data2vec locates both two sides of car tail and captures more, including the car wheel and car window. These results are consistent with the results on ResNet50. All these results show the generality of the implication of our theory on MRP.

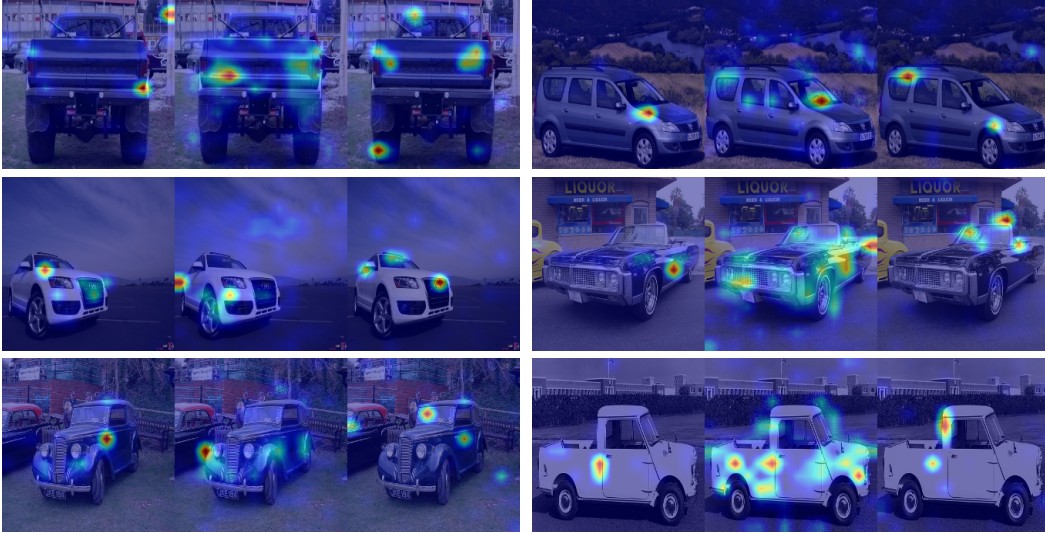

Figure 5: Class-specific visualization of ViT-base (Dosovitskiy et al., 2020) trained by MRP. For each group, the left image is the visualization of supervised model, while the middle one is from MAE and the right one is from data2vec. By comparison, the model trained by MAE and data2vec often captures more kinds of features than supervised model.

---

[2]For official trained SL model, you can download it at `https://github.com/facebookresearch/deit/blob/main/README_deit.md`.

[3]For official trained MAE model, you can download it at `https://github.com/facebookresearch/mae/blob/main/FINETUNE.md`.

[4]For official trained data2vec model, you can download it at `https://github.com/facebookresearch/fairseq/tree/main/examples/data2vec`.

## C  MAIN RESULT ON FEATURE LEARNING PROCESS OF MASK-RECONSTRUCTION PRETRAINING

In this section, we first show the main result on feature learning process mask-reconstruction pretraining (MRP). To introduce our main result, we first characterize the kernels at random initialization and during the training process. We first define

$$\Lambda_{i,l}^{(t)} := \max_{r \in [km]} [\langle w_r^{(t)}, v_{i,l} \rangle]^+,$$

where $w_r^{(t)}$ denotes the $r$-th convolution kernel of the student encoder at the $t$-th iteration. Here $\Lambda_{i,l}$ is the largest positive correlation score between the $l$-feature $v_{i,l}$ of $i$-th class and all the $km$ kernels $w_r^{(t)}$. We also define

$$\mathcal{M}_{i,l}^{(0)} := \left\{ r \in [km] : \langle w_r^{(0)}, v_{i,l} \rangle \geq \Lambda_{i,l}^{(0)} \left( 1 - O(\frac{1}{\log k}) \right) \right\}.$$

Here the set $\mathcal{M}_{i,l}^{(0)}$ is formed by the kernels which have slightly smaller correlation scores than the maximum score $\Lambda_{i,l}^{(0)}$ at the intial stage. If a kernel $w_r$ is not in $\mathcal{M}_{i,l}^{(0)}$, it means that the magnitude of $v_{i,l}$ inside the random initialization $w_r^{(0)}$ is non-trivially lagging behind, comparing to other kernels. Later we will prove that through the course of training, those kernels $w_r$ will lose the lottery and not learn anything useful for feature $v_{i,l}$.

We also have some properties of kernels at initialization ($t = 0$). The following lemma has been proved in (Allen-Zhu & Li, 2020, Fact B.1.).

**Lemma C.1** (The size of $\mathcal{M}_{i,l}^{(0)}$ at initialization). *With high probability at least $1 - e^{-\Omega(\log^5 k)}$, we have $|\mathcal{M}_{i,l}^{(0)}| \leq m_0$, where $m_0 := O(\log^5 k)$.*

Then to show our main result, we need some assumptions on the parameters and an induction hypothesis.

**Assumption C.2** (Parameter Assumption). *The parameters introduced in the paper need to satisfy the following conditions:*

- *$\varrho$ is the threshold for the smoothed ReLU activation. We assume $\varrho = \frac{1}{\text{polylog}(k)}$.*

- *$q \geq 3$ and $\sigma_0^{q-2} \leq \frac{1}{k}$.*

- *$\gamma$ controls feature noise. $\gamma \leq \tilde{O}\left(\frac{\sigma_0}{k}\right)$.*

- *$s$ controls feature sparsity. $s = \Theta(\text{polylog}(k))$.*

- *$N \geq \tilde{\omega}\left(\frac{k}{\sigma_0^{2q-1}}\right), \sqrt{d} \geq \tilde{\omega}(k/\sigma_0^{2q-1}), \sqrt{d} \geq \tilde{\omega}(k^{5/2}/\eta^{1/q})$ and $P \leq \sigma_0^{-q+1/2}$.*

- *$\text{polylog}(k) \leq m \leq \sqrt{k}$.*

- *$\eta \geq \frac{1}{k^{q(q-2)}}$ and $\eta \leq \frac{1}{\text{poly}(k)}$.*

- *$c(\theta) = \frac{1}{\theta}$.*

- *$\tau$ is the parameter controls the update of weights of Teacher network. $\tau = 1 + \frac{1-\theta}{C_p \theta} + \Theta(\frac{1}{t^{1/q}+1})$.*

The following induction hypothesis is important as it shows the main properties of kernels during the training course.

**Induction Hypothesis C.3.** *For every $v_{i,l} \in \mathcal{V}$, for each $r \in \mathcal{M}_{i,l}^{(0)}$, for every $(X, y) \in \mathcal{Z}$,*

*(a) For every $p \in \mathcal{P}_{v_{i,l}}(X)$, we have*

$$\langle w_r^{(t)}, x_p \rangle = \langle w_r^{(t)}, v_{i,l} \rangle z_p + \tilde{o}(\sigma_0).$$

(b) *For every $p \in \mathcal{P}(X) \setminus \mathcal{P}_{v_{i,l}}(X)$, we have*

$$|\langle w_r^{(t)}, x_p \rangle| \leq \tilde{O}(\sigma_0).$$

(c) *For every $p \in [P] \setminus \mathcal{P}(X)$, we have*

$$|\langle w_r^{(t)}, x_p \rangle| \leq \tilde{O}(\sigma_0 \gamma k).$$

*For every $r \notin \cup_{i \in [k], l \in [2]} \mathcal{M}_{i,l}^{(0)}$, for every $(X, y) \in \mathcal{Z}$,*

(d) *for every $p \in \mathcal{P}(X)$, $|\langle w_r^{(t)}, x_p \rangle| \leq \tilde{O}(\sigma_0)$.*

(e) *for every $p \in [P] \setminus \mathcal{P}(X)$, $|\langle w_r^{(t)}, x_p \rangle| \leq \tilde{O}(\sigma_0 \gamma k)$.*

*Moreover, for every $v_{i,l} \in \mathcal{V}$:*

(f) $\Lambda_{i,l}^{(t)} \in [\tilde{\Omega}(\sigma_0), \tilde{O}(1)]$.

(g) *for each $r \in \mathcal{M}_{i,l}^{(0)}$, $\langle w_r^{(t)}, v_{i,l} \rangle \geq -\tilde{O}(\sigma_0)$.*

(h) *for each $r \notin \mathcal{M}_{i,l}^{(0)}$, $\langle w_r^{(t)}, v_{i,l} \rangle \leq \tilde{O}(\sigma_0)$.*

Now we have the following result on the feature learning process of MRP.

**Theorem C.4** (Feature learning process of MRP). *Suppose Assumption C.2 holds. By running the gradient descent step in (2) with learning rate $\eta \leq \frac{1}{\text{poly}(k)}$, after $T = \frac{\text{poly}(k)}{\eta}$ iterations, for sufficiently large $k > 0$, Induction Hypothesis C.3 holds for all iterations $t = 0, 1, \ldots, T$ with high probability.*

**Differences from the work (Allen-Zhu & Li, 2020).** We use the similar multi-view data assumption 3.1 as (Allen-Zhu & Li, 2020), since we find it is a reasonable and practical assumption and it is helpful in proving what we actually have learned in the masked-reconstruction based pretraining. Besides, we also adopt the same network (two-layer CNNs with smoothed ReLu activation function) in (Allen-Zhu & Li, 2020) as our encoder. It is mainly for easy to compare the supervised learning results in (Allen-Zhu & Li, 2020) and self-supervised results proved by us. This can better illustrate the benefits of self-supervised pretraining.

But there are three main differences between our works and (Allen-Zhu & Li, 2020), including network architecture, objective loss and teacher. For network architecture, MRP contains both encoder and decoder, while supervised learning in (Allen-Zhu & Li, 2020) only considers the encoder. As for objective loss, MRP is a reconstruction loss of a masked input, while supervised learning in (Allen-Zhu & Li, 2020) uses distillation loss and cross-entropy loss of a non-masked input. Finally, in terms of teacher, MRP uses an online teacher whose parameters are changed along with training and thus is more dynamic and complex, while supervised learning in (Allen-Zhu & Li, 2020) uses a well-trained teacher whose parameter is fixed and thus gives a fixed target of an image. These three big differences cause the different lottery tickets winning or losing process during the training courses. This point can be observed in different practical intuition from our Induction Hypothesis C.3 and from Induction Hypothesis B.3 of (Allen-Zhu & Li, 2020). In our Induction Hypothesis C.3, no semantic features will lose the lottery tickets, while in (Allen-Zhu & Li, 2020) some of semantic features will be missed during the training courses. Based on different Induction Hypothesis, analysis of mask-reconstruction pretraining is non-trivial which is one part of our novel contributions.

Another part of our contributions is that after pretraining, we further need to show the test performance on downstream classification task. In this part, we use the same cross-entropy loss as (Allen-Zhu & Li, 2020), which is also popularly adopted in supervised training. But different from (Allen-Zhu & Li, 2020), which simply fixed the linear coefficients of output of convolution kernels of the encoder (backbone) by 1, here we need to train the weights of an extra linear layer and fine-tune the weights of convolution kernels of the encoder at the same time (see (15) in Appendix G).

# D   PROOF OVERVIEW OF THEOREM C.4

In this section, we introduce the main steps to prove Theorem C.4. The proof of Theorem C.4 includes two process. First, when $t \le T_0$, where $T_0 = \Theta(\frac{k}{\eta\sigma_0^{2q-2}})$ and when $t \in [T_0, T]$, where $T - T_0 \le \tilde{O}(\frac{kT_0^{1/q}}{\eta})$.

## D.1   INITIAL STAGE

The initial stage of the training process is defined as the training iterations $t \le T_0$, where $T_0 = \Theta(\frac{k}{\eta\sigma_0^{2q-2}})$. In this stage, kernels in $\mathcal{M}_{i,l}^{(0)}$ will focus on learning feature $v_{i,l}$. More formally, we will prove that at least one of kernels in $\mathcal{M}_{i,l}^{(0)}$ will capture the feature $v_{i,l}$, i.e.,

$$\max_{r \in \mathcal{M}_{i,l}^{(0)}} \langle w_r^{(T_0)}, v_{i,l} \rangle \ge \varrho = \frac{1}{\text{polylog}(k)}.$$

This indicates that the maximum correlation between kernels inside $\mathcal{M}_{i,l}^{(0)}$ and feature $v_{i,l}$ will grow.

Next, since the kernels inside $\mathcal{M}_{i,l}^{(0)}$ have captured the main correlations with feature $v_{i,l}$, what about the kernels outside $\mathcal{M}_{i,l}^{(0)}$? To answer this question, we will show that for $w_r \notin \mathcal{M}_{i,l}^{(0)}$,

$$\langle w_r^{(T_0)}, v_{i,l} \rangle \le \tilde{O}(\sigma_0) = \tilde{O}(\frac{1}{\sqrt{k}}),$$

which means that the correlations with feature $v_{i,l}$ will keep small. For those kernels that the magnitude of $v_{i,l}$ is lagging behind at initialization, it will loss the lottery and capture little feature $v_{i,l}$.

Furthermore, will those kernels inside $\mathcal{M}_{i,l}^{(0)}$ capture other feature $v_{j,l'} \ne v_{i,l}$? The answer is no. So to show this point, we also prove that for $r \in \mathcal{M}_{i,l}^{(0)}$,

$$\langle w_r^{(T_0)}, v_{j,l'} \rangle \le \tilde{O}(\sigma_0), \quad \forall v_{j,l'} \ne v_{i,l}.$$

Besides, the kernels will also not be influenced by the noises, i,e., for all $r, r \in [km]$, for every $p \in [P]$,

$$\langle w_r^{(T_0)}, \xi_p \rangle \le \tilde{O}\left(\frac{1}{\text{poly}(k)}\right).$$

## D.2   CONVERGENCE STAGE

In this stage, when $t \in [T_0, T]$, since part of kernels have won the lottery, its correlations with corresponding feature will continue to hold in this stage. But in this stage, the gradient will become small which drives the learning process to converge.

The intuition here is that, when the correlation between weights and its corresponding features grows over the threshold $\varrho$, the gradients will become to be small. That is when the kernels learned the corresponding feature, the increasing in the correlation will be small and thus drive the learning process to converge. We will show that after $t \ge T_0$,

$$\langle \nabla_{w_r^{(t)}} L(H), v_{i,l} \rangle \le \tilde{O}\left(\frac{1}{T_0^{1/q}}\right) = \tilde{O}\left(\frac{1}{\text{poly}(k)}\right), \quad \text{for } w_r \in \mathcal{M}_{i,l}^{(0)}.$$

While the gradients with other features (features not captured by this kernels) keep to be smaller. In this way, the correlation between weights and its corresponding features will not grow too large and we will show that

$$\max_{r \in \mathcal{M}_{i,l}^{(0)}} \langle w_r^{(T)}, v_{i,l} \rangle \le \tilde{O}(1).$$

## E    SOME TECHNICAL RESULTS

In this section, we first show the gradient and its approximations. We also state some consequences from our Induction Hypothesis C.3. They all are useful in our later proof of the main results.

### E.1    GRADIENTS AND ITS APPROXIMATIONS

Recall

$$L(H; X, \epsilon) = \frac{1}{2} \sum_{r \in [km]} \left( \sum_{p \in [P]} \overline{\mathrm{ReLU}}(\langle \hat{w}_r, x_p \rangle) - \sum_{p \in [P]} c(\theta) \overline{\mathrm{ReLU}}(\langle w_r, \epsilon_p x_p \rangle) \right)^2.$$

and

$$L(H; X) = \mathbb{E}_\epsilon[L(H; X, \epsilon)] = \frac{1}{2} \sum_{r \in [km]} \left( \sum_{p \in [P]} \overline{\mathrm{ReLU}}(\langle \hat{w}_r, x_p \rangle) - \sum_{p \in [P]} \overline{\mathrm{ReLU}}(\langle w_r, x_p \rangle) \right)^2$$
$$+ \frac{1}{2} \left( \frac{1}{\theta} - 1 \right) \sum_{r \in [km]} \sum_{p \in [P]} \left( \overline{\mathrm{ReLU}}(\langle w_r, x_p \rangle) \right)^2. \tag{3}$$

**The function of $\theta$.** As we make an expectation on the pretraining loss function $L(H; X, \epsilon)$ over $\epsilon$ when doing gradient descent, the expectation of $\epsilon$ ( here $\mathbb{E}[\epsilon] = \theta$) really matters in the analysis. More specifically, based on (3) and from the definition of $\theta$, when $\theta \to 1$, this means $P(\epsilon_p = 1) \to 1$, i.e., there is nearly no mask. Then according to our choice of the teacher's network parameters, when $\theta \to 1$, $\hat{w}_r \approx w_r$, which means that the loss keeps small and so there is nearly no update of parameters of student and teacher models. In this case, the student model cannot learn the useful semantic features in the data. When $\theta \to 0$ (i.e., we mask all the data), $\hat{w}_r \to \infty$ and $L(H; X) \to \infty$. In this case, the student model also cannot learn useful semantic features in the data. Overall, there is a tradeoff on the mask ratio $\theta$. The analysis also holds under the MAE model in Appendix. H. When $\theta \to 0$, $L(H; X) \to \infty$. When $\theta \to 1$, there is a trivial answer to make $L(H; X) \to 0$. In both above cases, no semantic features will be learned.

The derivation of above loss function is shown as follows. For simplicity of clarification, we denote $\hat{y}_{r,p} = \overline{\mathrm{ReLU}}(\langle \hat{w}_r, x_p \rangle)$ and $y_{r,p} = \overline{\mathrm{ReLU}}(\langle w_r, x_p \rangle)$. Then the loss function is

$$L(H; X, \epsilon) = \frac{1}{2} \sum_{r \in [km]} \left[ \left( \sum_{p \in [P]} \hat{y}_{r,p} \right)^2 - 2 \sum_{p \in [P]} \hat{y}_{r,p} \sum_{p \in [P]} c(\theta) \epsilon_p y_{r,p} + \left( \sum_{p \in [P]} c(\theta) \epsilon_p y_{r,p} \right)^2 \right].$$

Because 1) we set $c(\theta) = \frac{1}{\theta}$ and 2) each $\epsilon_p$ is i.i.d. Bernoulli and $\mathbb{E}[\epsilon_p] = \theta$, we obtain

$$\mathbb{E}_\epsilon \left[ 2 \sum_{p \in [P]} \hat{y}_{r,p} \sum_{p \in [P]} c(\theta) \epsilon_p y_{r,p} \right] = 2 \sum_{p \in [P]} \hat{y}_{r,p} \sum_{p \in [P]} y_{r,p}.$$

We also have

$$\mathbb{E}_{\boldsymbol{\epsilon}}\left[\left(\sum_{p\in[P]}c(\theta)\epsilon_p y_{r,p}\right)^2\right] = \frac{1}{\theta^2}\mathbb{E}_{\boldsymbol{\epsilon}}\left[\left(\sum_{p\in[P]}\epsilon_p y_{r,p}\right)\left(\sum_{p\in[P]}\epsilon_p y_{r,p}\right)\right]$$

$$= \frac{1}{\theta^2}\mathbb{E}_{\boldsymbol{\epsilon}}\left[\left(\sum_{p\in[P]}\epsilon_p y_{r,p}\right)\left(\sum_{p'\neq p}\epsilon_{p'} y_{r,p'}\right)\right]$$

$$+ \frac{1}{\theta^2}\mathbb{E}_{\boldsymbol{\epsilon}}\left[\left(\sum_{p\in[P]}\epsilon_p y_{r,p}\right)\left(\sum_{p'=p}\epsilon_{p'} y_{r,p'}\right)\right]$$

$$= \frac{1}{\theta^2}\mathbb{E}_{\boldsymbol{\epsilon}}\left[\sum_{p\in[P]}\sum_{p'\neq p}\epsilon_p\epsilon_{p'} y_{r,p}y_{r,p'}\right] + \frac{1}{\theta^2}\mathbb{E}_{\boldsymbol{\epsilon}}\left[\sum_{p\in[P]}\sum_{p'=p}\epsilon_p\epsilon_{p'} y_{r,p}y_{r,p'}\right]$$

$$\overset{(a)}{=} \left(\sum_{p\in[P]}y_{r,p}\right)\left(\sum_{p'\neq p}y_{r,p'}\right) + \frac{1}{\theta}\left[\left(\sum_{p\in[P]}y_{r,p}\right)\left(\sum_{p'=p}y_{r,p'}\right)\right]$$

$$= \left(\sum_{p\in[P]}y_{r,p}\right)\left(\sum_{p'}y_{r,p'}\right) + \left(\frac{1}{\theta}-1\right)\left[\left(\sum_{p\in[P]}y_{r,p}\right)\left(\sum_{p'=p}y_{r,p'}\right)\right]$$

$$= \left(\sum_{p\in[P]}y_{r,p}\right)\left(\sum_{p'}y_{r,p'}\right) + \left(\frac{1}{\theta}-1\right)\left[\sum_{p\in[P]}y_{r,p}^2\right]$$

where (a) is because $\mathbb{E}_{\boldsymbol{\epsilon}}[\epsilon_p\epsilon_{p'}] = \theta^2$ when $p \neq p'$ as we assume the variable in $\boldsymbol{\epsilon}$ is independent Bernoulli variable with $\Pr(\epsilon_p = 1) = \theta$ and $\mathbb{E}_{\boldsymbol{\epsilon}}[\epsilon_p\epsilon_{p'}] = \theta$ when $p = p'$. Combining the above results, we have

$$L(H;X) = \mathbb{E}_{\boldsymbol{\epsilon}}[L(H;X,\boldsymbol{\epsilon})] = \frac{1}{2}\sum_{r\in[km]}\left(\sum_{p\in[P]}\hat{y}_{r,p} - \sum_{p\in[P]}y_{r,p}\right)^2 + \frac{1}{2}\left(\frac{1}{\theta}-1\right)\sum_{r\in[km]}\sum_{p\in[P]}y_{r,p}^2,$$

which is our result.

We define

$$\Phi_r(X) := \sum_{p\in[P]}\overline{\mathrm{ReLU}}(\langle\hat{w}_r, x_p\rangle) - \sum_{p\in[P]}\overline{\mathrm{ReLU}}(\langle w_r, x_p\rangle).$$

**Fact 2.1** (Gradients). *Given the data point* $(X, y) \in \mathcal{D}$, *for every* $w_r, r \in [km]$,

$$-\nabla_{w_r}L(X) = \sum_{p\in[P]}\left(\Phi_r(X) - \left(\frac{1}{\theta}-1\right)\overline{\mathrm{ReLU}}(\langle w_r, x_p\rangle)\right)\overline{\mathrm{ReLU}}'(\langle w_r, x_p\rangle)x_p.$$

*where* $\overline{\mathrm{ReLU}}'$ *is the gradient of* $\overline{\mathrm{ReLU}}$. *Besides, we set* $\hat{w}_r^{(t)} = \tau w_r^{(t)}$.

We define several error terms that will be used in our proofs.

**Definition E.1.**

$$V_{r,i,l}(X) = \mathbb{I}_{\{v_{i,l}\in\mathcal{V}(X)\}}\sum_{p\in\mathcal{P}_{v_{i,l}}(X)}\overline{\mathrm{ReLU}}'(\langle w_r, x_p\rangle)z_p,$$

$$\hat{V}_{r,i,l}(X) = \mathbb{I}_{\{v_{i,l}\in\mathcal{V}(X)\}}\sum_{p\in\mathcal{P}_{v_{i,l}}(X)}\overline{\mathrm{ReLU}}'(\langle w_r, x_p\rangle),$$

$$W_{r,i,l}(X) = \left(\frac{1}{\theta}-1\right)\mathbb{I}_{\{v_{i,l}\in\mathcal{V}(X)\}}\sum_{p\in\mathcal{P}_{v_{i,l}}(X)}\overline{\mathrm{ReLU}}(\langle w_r, x_p\rangle)\overline{\mathrm{ReLU}}'(\langle w_r, x_p\rangle)z_p,$$

$$\hat{W}_{r,i,l}(X) = \left(\frac{1}{\theta}-1\right)\mathbb{I}_{\{v_{i,l}\in\mathcal{V}(X)\}}\sum_{p\in\mathcal{P}_{v_{i,l}}(X)}\overline{\mathrm{ReLU}}(\langle w_r, x_p\rangle)\overline{\mathrm{ReLU}}'(\langle w_r, x_p\rangle),$$

$$\Delta_{r,i,l}(X) = \mathbb{I}_{v_{i,l}\in\mathcal{V}(X)}\sum_{p\in\mathcal{P}_{v_{i,l}}(X)}\left[\overline{\mathrm{ReLU}}(\langle\hat{w}_r, x_p\rangle) - \overline{\mathrm{ReLU}}(\langle w_r, x_p\rangle)\right].$$

$$U_{i,l}(X) = \mathbb{I}_{\{v_{i,l}\in\mathcal{V}(X)\}}\cdot\tilde{O}(\sigma_0^{(2q-1)}).$$

We also define some small terms for easy of notation.

**Definition E.2.**

$$\mathcal{E}_1 = \tilde{O}(\sigma_0^{(q-1)})(\gamma + \sigma_p)s, \quad \mathcal{E}_2 = \tilde{O}((\sigma_0\gamma k)^{(q-1)})(\gamma + \sigma_p) \cdot P,$$
$$\mathcal{E}_3 = \tilde{O}(\sigma_0^{(2q-1)})(\gamma + \sigma_p)s, \quad \mathcal{E}_4 = \tilde{O}((\sigma_0\gamma k)^{(2q-1)})(\gamma + \sigma_p) \cdot P.$$
$$\mathcal{E}_5 = \tilde{O}(\sigma_0^q) \cdot (s+1), \quad \mathcal{E}_6 = \tilde{O}((\sigma_0\gamma k)^q) \cdot P.$$

We have the following lemma to approximate the gradient. We first approximate the term $\Phi_r(X)$.

**Claim E.3** (Bounds on $\Phi_r(X)$). *Suppose Assumption C.2 holds and Induction Hypothesis C.3 holds at iteration $t$. Then for every $v_{i,l} \in \mathcal{V}$, for every $r \in \mathcal{M}_{i,l}^{(0)}$, for every $(X, y) \in \mathcal{Z}$,*

$$\Phi_r(X) = \Delta_{r,i,l}(X) \pm \mathcal{E}_5 \pm \mathcal{E}_6.$$

*Proof of Claim E.3.* Using the induction hypothesis C.3, for every $v_{i,l} \in \mathcal{V}$, for every $r \in \mathcal{M}_{i,l}^{(0)}$, for every $(X, y) \in \mathcal{Z}$,

$$\Phi_r(X) = \sum_{p \in [P]} \overline{\text{ReLU}}(\langle \hat{w}_r, x_p \rangle) - \sum_{p \in [P]} \overline{\text{ReLU}}(\langle w_r, x_p \rangle)$$

$$= \mathbb{I}_{v_{i,l} \in \mathcal{V}(X)} \sum_{p \in \mathcal{P}_{v_{i,l}}(X)} \left[ \overline{\text{ReLU}}(\langle \hat{w}_r, x_p \rangle) - \overline{\text{ReLU}}(\langle w_r, x_p \rangle) \right]$$

$$+ \sum_{p \in \mathcal{P}(X) \backslash \mathcal{P}_{v_{i,l}}(X)} \left[ \overline{\text{ReLU}}(\langle \hat{w}_r, x_p \rangle) - \overline{\text{ReLU}}(\langle w_r, x_p \rangle) \right]$$

$$+ \sum_{p \in [P] \backslash \mathcal{P}(X)} \left[ \overline{\text{ReLU}}(\langle \hat{w}_r, x_p \rangle) - \overline{\text{ReLU}}(\langle w_r, x_p \rangle) \right]$$

$$\overset{(a)}{=} \mathbb{I}_{v_{i,l} \in \mathcal{V}(X)} \sum_{p \in \mathcal{P}_{v_{i,l}}(X)} \left[ \overline{\text{ReLU}}(\langle \hat{w}_r, x_p \rangle) - \overline{\text{ReLU}}(\langle w_r, x_p \rangle) \right]$$

$$\pm \tilde{O}(\sigma_0^q) \cdot (s+1) \pm \tilde{O}((\sigma_0\gamma k)^q) \cdot P,$$

where $(a)$ is as $C_p$ is a universal constant. $\qquad\qquad\square$

**Claim E.4** (Approximations of gradients). *Suppose Assumption C.2 holds and Induction Hypothesis C.3 holds at iteration $t$. Then for every $v_{i,l} \in \mathcal{V}$, for every $r \in \mathcal{M}_{i,l}^{(0)}$, for $(X, y) \in \mathcal{Z}$,*

*(a)*

$$\langle -\nabla_{w_r} L(X), v_{i,l} \rangle = V_{r,i,l}(X)\Delta_{r,i,l}(X) - W_{r,i,l}(X) + \Delta_{r,i,l}(X)(\mathcal{E}_1 + \mathcal{E}_2) - \mathcal{E}_3 - \mathcal{E}_4$$
$$\pm (V_{r,i,l}(X) + \hat{V}_{r,i,l}(X)(\gamma + \sigma_p))(\mathcal{E}_5 + \mathcal{E}_6) \pm (\mathcal{E}_5 + \mathcal{E}_6)(\mathcal{E}_1 + \mathcal{E}_2)$$

*(b) for $v_{j,l'} \neq v_{i,l}$ (note that $v_{j,l'} \neq v_{i,l}$ means that when $j = i$, $l' \neq l$ or $j \neq i$),*

$$|\langle -\nabla_{w_r} L(X), v_{j,l'} \rangle| = \left( \hat{V}_{r,i,l}(X)\Delta_{r,i,l}(X) - \hat{W}_{r,i,l}(X) \right)(\gamma + \sigma_p) \pm \hat{V}_{r,i,l}(X)(\mathcal{E}_5 + \mathcal{E}_6)(\gamma + \sigma_p)$$
$$\pm U_{r,j,l'}(X) + \Delta_{r,i,l}(X)(\mathcal{E}_1 + \mathcal{E}_2) \pm (\mathcal{E}_5 + \mathcal{E}_6)(\mathcal{E}_1 + \mathcal{E}_2) - \mathcal{E}_3 - \mathcal{E}_4.$$

*Proof of Claim E.4.* We first prove (a). Using the induction hypothesis C.3 and the fact $C_p$ is a universal constant, we have that for $v_{i,l} \in \mathcal{V}$, for every $r \in \mathcal{M}_{i,l}^{(0)}$, we have

$$\langle -\nabla_{w_r} L(X), v_{i,l} \rangle$$

$$= \sum_{p \in [P]} \left( \Phi_r(X) - \left(\frac{1}{\theta} - 1\right)\overline{\mathrm{ReLU}}(\langle w_r, x_p \rangle) \right)\overline{\mathrm{ReLU}}'(\langle w_r, x_p \rangle)\langle x_p, v_{i,l} \rangle$$

$$= \mathbb{I}_{\{v_{i,l} \in \mathcal{V}(X)\}} \sum_{p \in \mathcal{P}_{v_{i,l}}(X)} \left( \Phi_r(X) - \left(\frac{1}{\theta} - 1\right)\overline{\mathrm{ReLU}}(\langle w_r, x_p \rangle) \right)\overline{\mathrm{ReLU}}'(\langle w_r, x_p \rangle)(z_p + \alpha_{p,v_{i,l}} + \langle v_{i,l}, \xi_p \rangle)$$

$$+ \sum_{p \in \mathcal{P}(X) \backslash \mathcal{P}_{v_{i,l}}(X)} \left( \Phi_r(X) - \left(\frac{1}{\theta} - 1\right)\overline{\mathrm{ReLU}}(\langle w_r, x_p \rangle) \right)\overline{\mathrm{ReLU}}'(\langle w_r, x_p \rangle)(\alpha_{p,v_{i,l}} + \langle v_{i,l}, \xi_p \rangle)$$

$$+ \sum_{p \in [P] \backslash \mathcal{P}(X)} \left( \Phi_r(X) - \left(\frac{1}{\theta} - 1\right)\overline{\mathrm{ReLU}}(\langle w_r, x_p \rangle) \right)\overline{\mathrm{ReLU}}'(\langle w_r, x_p \rangle)(\alpha_{p,v_{i,l}} + \langle v_{i,l}, \xi_p \rangle)$$

$$= \mathbb{I}_{\{v_{i,l} \in \mathcal{V}(X)\}} \sum_{p \in \mathcal{P}_{v_{i,l}}(X)} \left( \Delta_{r,i,l}(X) - \left(\frac{1}{\theta} - 1\right)\overline{\mathrm{ReLU}}(\langle w_r, x_p \rangle) \right)\overline{\mathrm{ReLU}}'(\langle w_r, x_p \rangle)z_p$$

$$\pm V_{r,i,l}(X)(\mathcal{E}_5 + \mathcal{E}_6) \pm \hat{V}_{r,i,l}(X)(\mathcal{E}_5 + \mathcal{E}_6) \cdot (\gamma + \sigma_p)$$
$$+ (\Delta_{r,i,l}(X) \pm \mathcal{E}_5 \pm \mathcal{E}_6 - \tilde{O}(\sigma_0^q)) \cdot \tilde{O}(\sigma_0^{q-1}) \cdot (\gamma + \sigma_p) \cdot (s + 1)$$
$$+ (\Delta_{r,i,l}(X) \pm \mathcal{E}_5 \pm \mathcal{E}_6 - \tilde{O}((\sigma_0 \gamma k)^q)) \cdot \tilde{O}((\sigma_0 \gamma k)^{q-1}) \cdot (\gamma + \sigma_p) \cdot P$$
$$= V_{r,i,l}(X)\Delta_{r,i,l}(X) - W_{r,i,l}(X) + \Delta_{r,i,l}(X)(\mathcal{E}_1 + \mathcal{E}_2)$$
$$\pm (V_{r,i,l}(X) + \hat{V}_{r,i,l}(X)(\gamma + \sigma_p))(\mathcal{E}_5 + \mathcal{E}_6) \pm (\mathcal{E}_5 + \mathcal{E}_6)(\mathcal{E}_1 + \mathcal{E}_2) - \mathcal{E}_3 - \mathcal{E}_4$$

Now we show $(b)$. Using the induction hypothesis C.3, for $v_{i,l} \in \mathcal{V}$, for every $r \in \mathcal{M}_{i,l}^{(0)}$, when $v_{j,l'} \neq v_{i,l}$, we have

$$\langle -\nabla_{w_r} L(X), v_{j,l'} \rangle$$

$$= \sum_{p \in [P]} \left( \Phi_r(X) - \left(\frac{1}{\theta} - 1\right)\overline{\mathrm{ReLU}}(\langle w_r, x_p \rangle) \right)\overline{\mathrm{ReLU}}'(\langle w_r, x_p \rangle)\langle x_p, v_{j,l'} \rangle$$

$$= \mathbb{I}_{\{v_{i,l} \in \mathcal{V}(X)\}} \sum_{p \in \mathcal{P}_{v_{i,l}}(X)} \left( \Phi_r(X) - \left(\frac{1}{\theta} - 1\right)\overline{\mathrm{ReLU}}(\langle w_r, x_p \rangle) \right)\overline{\mathrm{ReLU}}'(\langle w_r, x_p \rangle)(\alpha_{p,v_{j,l'}} + \langle v_{j,l'}, \xi_p \rangle)$$

$$+ \mathbb{I}_{\{v_{j,l'} \in \mathcal{V}(X)\}} \sum_{p \in \mathcal{P}_{v_{j,l'}}(X)} \left( \Phi_r(X) - \left(\frac{1}{\theta} - 1\right)\overline{\mathrm{ReLU}}(\langle w_r, x_p \rangle) \right)\overline{\mathrm{ReLU}}'(\langle w_r, x_p \rangle)(z_p + \alpha_{p,v_{j,l'}} + \langle v_{j,l'}, \xi_p \rangle)$$

$$+ \sum_{p \in \mathcal{P}(X) \backslash \{\mathcal{P}_{v_{i,l}}(X) \cup \mathcal{P}_{v_{j,l'}}(X)\}} \left( \Phi_r(X) - \left(\frac{1}{\theta} - 1\right)\overline{\mathrm{ReLU}}(\langle w_r, x_p \rangle) \right)\overline{\mathrm{ReLU}}'(\langle w_r, x_p \rangle)(\alpha_{p,v_{j,l'}} + \langle v_{j,l'}, \xi_p \rangle)$$

$$+ \sum_{p \in [P] \backslash \mathcal{P}(X)} \left( \Phi_r(X) - \left(\frac{1}{\theta} - 1\right)\overline{\mathrm{ReLU}}(\langle w_r, x_p \rangle) \right)\overline{\mathrm{ReLU}}'(\langle w_r, x_p \rangle)(\alpha_{p,v_{j,l'}} + \langle v_{j,l'}, \xi_p \rangle)$$

$$= \left( \hat{V}_{r,i,l}(X)\Delta_{r,i,l}(X) - \hat{W}_{r,i,l}(X) \right)(\gamma + \sigma_p) \pm \hat{V}_{r,i,l}(X)(\mathcal{E}_5 + \mathcal{E}_6)(\gamma + \sigma_p)$$
$$\pm U_{r,j,l'}(X) + \Delta_{r,i,l}(X)(\mathcal{E}_1 + \mathcal{E}_2) \pm (\mathcal{E}_5 + \mathcal{E}_6)(\mathcal{E}_1 + \mathcal{E}_2) - \mathcal{E}_3 - \mathcal{E}_4.$$

$\square$

## E.2 SOME RESULTS FROM INDUCTION HYPOTHESIS C.3

### E.2.1 GROWTH OF $\Lambda_{i,l}^{(t)}$

The following claim shows about at which iteration $\Lambda_{i,l}^{(t)}$ will be greater than the threshold $\varrho$ in the definition of smooth ReLU function.

**Claim E.5.** *Suppose Assumption C.2 holds and induction hypothesis C.3 holds at iteration t. For every $v_{i,l}$, suppose $\Lambda_{i,l}^{(t)} \leq \varrho$. Then we have*

$$\Lambda_{i,l}^{(t+1)} = \Lambda_{i,l}^{(t)} + \tilde{\Theta}\left(\frac{\eta}{k}\right)\overline{\text{ReLU}}(\Lambda_{i,l}^{(t)})\overline{\text{ReLU}}'(\Lambda_{i,l}^{(t)}).$$

*Proof of Claim E.5.* Recall that $\Lambda_{i,l}^{(t)} := \max_{r \in [km]}[\langle w_r^{(t)}, v_{i,l}\rangle]^+$. We choose any $r \in [km]$ that makes $\langle w_r^{(t)}, v_{i,l}\rangle \geq \tilde{\Omega}(\sigma_0)$. Now we show the updates. We know that

$$\langle w_r^{(t+1)}, v_{i,l}\rangle = \langle w_r^{(t)}, v_{i,l}\rangle + \eta\mathbb{E}_{(X,y)\sim\mathcal{Z}}\left[\langle -\nabla_{w_r}L(X), v_{i,l}\rangle\right]$$

Using Claim E.4, we have

$$\langle -\nabla_{w_r}L(X), v_{i,l}\rangle = V_{r,i,l}(X)\Delta_{r,i,l}(X) - W_{r,i,l}(X) + \Delta_{r,i,l}(X)(\mathcal{E}_1 + \mathcal{E}_2) - \mathcal{E}_3 - \mathcal{E}_4$$
$$\pm (V_{r,i,l}(X) + \hat{V}_{r,i,l}(X)(\gamma + \sigma_p))(\mathcal{E}_5 + \mathcal{E}_6) \pm (\mathcal{E}_5 + \mathcal{E}_6)(\mathcal{E}_1 + \mathcal{E}_2)$$

Recall the definition of $V_{r,i,l}, \Delta_{r,i,l}, W_{r,i,l}$. As we assume $\Lambda_{i,l}^{(t)} \leq \varrho$ and based on our definition of smooth ReLU function, we could simplify the above inequalities by only keeping the main increasing term as

$$\langle -\nabla_{w_r}L(X), v_{i,l}\rangle = \Delta_{r,i,l}(X)V_{r,i,l}(X) - W_{r,i,l}(X).$$

This equation is obtained by setting $\langle w_r^{(t)}, v_{i,l}\rangle \geq \tilde{\Omega}(\sigma_0)$ and compare its order with the remaining term. It is indeed the main increasing term. For $(X, y) \in \mathcal{Z}$, we have

$$V_{r,i,l}(X) = \mathbb{I}_{\{v_{i,l}\in\mathcal{V}(X)\}}\overline{\text{ReLU}}'(\langle w_r^{(t)}, v_{i,l}\rangle)\sum_{p\in\mathcal{P}_{v_{i,l}}(X)} z_p^q \tag{4}$$

$$\Delta_{r,i,l}(X) = \mathbb{I}_{\{v_{i,l}\in\mathcal{V}(X)\}}\overline{\text{ReLU}}(\langle w_r^{(t)}, v_{i,l}\rangle)\sum_{p\in\mathcal{P}_{v_{i,l}}(X)}(\tau^q - 1)z_p^q, \tag{5}$$

$$W_{r,i,l}(X) = \mathbb{I}_{\{v_{i,l}\in\mathcal{V}(X)\}}\overline{\text{ReLU}}(\langle w_r^{(t)}, v_{i,l}\rangle)\overline{\text{ReLU}}'(\langle w_r^{(t)}, v_{i,l}\rangle)\left(\frac{1}{\theta} - 1\right)\sum_{p\in\mathcal{P}_{v_{i,l}}(X)} z_p^{2q} \tag{6}$$

Then

$$\Delta_{r,i,l}(X)V_{r,i,l}(X) - W_{r,i,l}(X) = \sum_{p\in\mathcal{P}_{v_{i,l}}(X)} z_p^q\left(\sum_{p'\in\mathcal{P}_{v_{i,l}}(X)}(\tau^q - 1)z_{p'}^q - \left(\frac{1}{\theta} - 1\right)z_p^q\right)$$
$$\times \mathbb{I}_{\{v_{i,l}\in\mathcal{V}(X)\}}\overline{\text{ReLU}}(\langle w_r^{(t)}, v_{i,l}\rangle)\overline{\text{ReLU}}'(\langle w_r^{(t)}, v_{i,l}\rangle).$$

According to our choice of $\tau$ and $z_p$ is uniformly distributed over $C_p$ patches, when $(X, y) \in \mathcal{Z}_m$ and $i = y$ or $(X, y) \in \mathcal{Z}_s$ and $i = y$ and $\hat{l} = l$, we have

$$\mathbb{E}_{(X,y)\in\mathcal{Z}}\left[\sum_{p\in\mathcal{P}_{v_{i,l}}(X)} z_p^q\left(\sum_{p'\in\mathcal{P}_{v_{i,l}}}(\tau^q - 1)z_{p'}^q - \left(\frac{1}{\theta} - 1\right)z_p^q\right)\mathbb{I}_{\{v_{i,l}\in\mathcal{V}(X)\}}\right] \in [\Omega(1), O(1)].$$

When $(X, y) \in \mathcal{Z}_s$ and $i = y$ and $\hat{l} = 3 - l$, we have

$$\mathbb{E}_{(X,y)\in\mathcal{Z}_s}\left[\sum_{p\in\mathcal{P}_{v_{i,l}}(X)} z_p^q\left(\sum_{p'\in\mathcal{P}_{v_{i,l}}}(\tau^q - 1)z_{p'}^q - \left(\frac{1}{\theta} - 1\right)z_p^q\right)\mathbb{I}_{\{v_{i,l}\in\mathcal{V}(X)\}}\right] \in [\Omega(\rho), O(\rho)].$$

When $(X, y) \in \mathcal{Z}$ and $i \neq y$, we have

$$\mathbb{E}_{(X,y)\in\mathcal{Z}}\left[\sum_{p\in\mathcal{P}_{v_{i,l}}(X)} z_p^q\left(\sum_{p'\in\mathcal{P}_{v_{i,l}}}(\tau^q - 1)z_{p'}^q - \left(\frac{1}{\theta} - 1\right)z_p^q\right)\mathbb{I}_{\{v_{i,l}\in\mathcal{V}(X)\}}\right] \in \frac{s}{k}[\Omega(1), O(1)].$$

Combining all above results, we have

$$\langle w_r^{(t+1)}, v_{i,l}\rangle = \langle w_r^{(t)}, v_{i,l}\rangle + \tilde{\Theta}\left(\frac{\eta}{k}\right)\overline{\text{ReLU}}(\langle w_r^{(t)}, v_{i,l}\rangle)\overline{\text{ReLU}}'(\langle w_r^{(t)}, v_{i,l}\rangle).$$

$\square$

Using Claim E.5, and $\tilde{\Omega}(\sigma_0) \leq \Lambda_{i,l}^{(0)} \leq \tilde{O}(\sigma_0)$, we have the following result:

**Claim E.6.** *Suppose Assumption C.2 holds and Induction Hypothesis C.3 holds for every iteration. Define $T_0 := \tilde{\Theta}\left(\frac{k}{\eta\sigma_0^{2q-2}}\right)$. We have that when $t \geq T_0$, it satisfies $\Lambda_{i,l}^{(t)} \geq \Theta\left(\frac{1}{\text{polylog}(k)}\right)$.*

*Proof of Claim E.6.* Using the result in E.5 and beginning from $\Lambda_{i,l}^{(0)} = \tilde{\Theta}(\sigma_0)$, we have that

$$\Lambda_{i,l}^{(t)} \approx \Lambda_{i,l}^{(0)}\left(1 + \tilde{\Theta}\left(\frac{\eta}{k}\right)\frac{\sigma_0^{2q-2}}{\varrho^{2q-2}}\right)^t. \tag{7}$$

Thus, when $T_0 = \tilde{\Theta}\left(\frac{k}{\eta\sigma_0^{2q-2}}\right)$, we have

$$\Lambda_{i,l}^{(t)} \approx \tilde{\Theta}(\sigma_0)e^{\text{polylog}(k)},$$

which means

$$\Lambda_{i,l}^{(t)} = \Theta\left(\frac{1}{\text{polylog}(k)}\right).$$

$\square$

# F PROOF OF THEOREM C.4

Before we formally show Theorem C.4, we need some lemmas. First, we need to prove that for every feature $v_{i,l} \in \mathcal{V}$. at least one of "diagonal" correlations $\langle w_r^{(t)}, v_{i,l}\rangle, r \in \mathcal{M}_{i,l}^{(0)}$ grows and the "off-diagonal" correlations $\langle w_r^{(t)}, v_{j,l'}\rangle, v_{j,l'} \neq v_{i,l}$ decreases. To show these, we provide three lemmas about the lower and upper bound on $\langle w_r^{(t)}, v_{i,l}\rangle, r \in \mathcal{M}_{i,l}^{(0)}$ and upper bound on $\langle w_r^{(t)}, v_{j,l'}\rangle, v_{j,l'} \neq v_{i,l}, r \in \mathcal{M}_{i,l}^{(0)}$.

## F.1 DIAGONAL CORRELATIONS

The first lemma is used to obtain upper bound on $\Lambda_{i,l}^{(t)}$.

**Lemma F.1.** *Suppose Assumption C.2 holds and Induction Hypothesis C.3 holds for all iterations $< t$. We have*

$$\forall v_{i,l} \in \mathcal{V}: \quad \Lambda_{i,l}^{(t)} \leq \tilde{O}(1).$$

*Proof of Lemma F.1.* Based on Claim E.4, we have that for every $r \in \mathcal{M}_{i,l}^{(0)}$,

$$\langle w_r^{(t+1)}, v_{i,l}\rangle = \langle w_r^{(t)}, v_{i,l}\rangle + \eta\mathbb{E}_{(X,y)\sim\mathcal{Z}}\Bigg[V_{r,i,l}(X)\Delta_{r,i,l}(X) - W_{r,i,l}(X) + \Delta_{r,i,l}(X)(\mathcal{E}_1 + \mathcal{E}_2)$$

$$\pm (V_{r,i,l}(X) + \hat{V}_{r,i,l}(X)(\gamma + \sigma_p))(\mathcal{E}_5 + \mathcal{E}_6) \pm (\mathcal{E}_5 + \mathcal{E}_6)(\mathcal{E}_1 + \mathcal{E}_2) - \mathcal{E}_3 - \mathcal{E}_4\Bigg].$$

When taking the positive part, we know there exists $\delta_{r,i,l}^{(t)} \in [0,1]$ such that

$$[\langle w_r^{(t+1)}, v_{i,l}\rangle]^+ = [\langle w_r^{(t)}, v_{i,l}\rangle]^+ + \eta\delta_{r,i,l}^{(t)}\mathbb{E}_{(X,y)\sim\mathcal{Z}}\Bigg[V_{r,i,l}(X)\Delta_{r,i,l}(X) - W_{r,i,l}(X) + \Delta_{r,i,l}(X)(\mathcal{E}_1 + \mathcal{E}_2)$$

$$\pm (V_{r,i,l}(X) + \hat{V}_{r,i,l}(X)(\gamma + \sigma_p))(\mathcal{E}_5 + \mathcal{E}_6) \pm (\mathcal{E}_5 + \mathcal{E}_6)(\mathcal{E}_1 + \mathcal{E}_2) - \mathcal{E}_3 - \mathcal{E}_4\Bigg].$$

Suppose we are now at some iteration $t > T_0$. In this stage, $\Lambda_{i,l}^{(t)} \geq 1/\text{polylog}(k)$. As $T_0 = \tilde{\Theta}\left(\frac{k}{\eta\sigma_0^{2q-2}}\right)$ and $\eta \leq \frac{1}{\text{poly}(k)}$, we have

$$\Delta_{r,i,l}(X)V_{r,i,l}(X) - W_{r,i,l}(X) = \sum_{p\in\mathcal{P}_{v_{i,l}}(X)} z_p \left(\sum_{p'\in\mathcal{P}_{v_{i,l}}(X)}(\tau-1)z_{p'} - \left(\frac{1}{\theta}-1\right)z_p\right)$$
$$\times \mathbb{I}_{\{v_{i,l}\in\mathcal{V}(X)\}}\overline{\text{ReLU}}(\langle w_r^{(t)}, v_{i,l}\rangle)\overline{\text{ReLU}}'(\langle w_r^{(t)}, v_{i,l}\rangle)$$
$$= O\left(\frac{1}{t^{1/q}}\right)\cdot\mathbb{I}_{\{v_{i,l}\in\mathcal{V}(X)\}}\overline{\text{ReLU}}(\langle w_r^{(t)}, v_{i,l}\rangle)\overline{\text{ReLU}}'(\langle w_r^{(t)}, v_{i,l}\rangle).$$

Using Claim E.5 and we also keep the main increasing term, we have

$$[\langle w_r^{(t+1)}, v_{i,l}\rangle]^+ \leq [\langle w_r^{(t)}, v_{i,l}\rangle]^+ + \tilde{O}\left(\frac{\eta}{kT_0^{1/q}}\right)\cdot\overline{\text{ReLU}}(\langle w_r^{(t)}, v_{i,l}\rangle)\overline{\text{ReLU}}'(\langle w_r^{(t)}, v_{i,l}\rangle)$$

$$\leq [\langle w_r^{(t)}, v_{i,l}\rangle]^+ + \tilde{O}\left(\frac{\eta}{kT_0^{1/q}}\right)\cdot\overline{\text{ReLU}}(\langle w_r^{(t)}, v_{i,l}\rangle).$$

Taking the maximum on both side and as we are at $t > T_0$, we have

$$\max_{r\in\mathcal{M}_{i,l}^{(0)}}[\langle w_r^{(t+1)}, v_{i,l}\rangle]^+ \leq \max_{r\in\mathcal{M}_{i,l}^{(0)}}[\langle w_r^{(t)}, v_{i,l}\rangle]^+\left(1+\tilde{O}\left(\frac{\eta}{kT_0^{1/q}}\right)\right).$$

When $t \leq T = T_0 + \tilde{O}\left(\frac{kT_0^{1/q}}{\eta}\right)$, we have

$$\Lambda_{i,l}^{(t)} \leq \tilde{O}(1).$$

$\square$

The second lemma is used to lower bound on $\langle w_r^{(t)}, v_{i,l}\rangle, r\in\mathcal{M}_{i,l}^{(0)}$ and indicates that the diagonal correlations are nearly non-negative.

**Lemma F.2.** *Suppose Assumption C.2 holds and Induction Hypothesis C.3 holds for all iterations $< t$. We have*

$$\forall v_{i,l}\in\mathcal{V}, \forall r\in\mathcal{M}_{i,l}^{(0)}: \quad \langle w_r^{(t)}, v_{i,l}\rangle \geq -\tilde{O}(\sigma_0).$$

*Proof of Lemma F.2.* We start with any iteration $t$ that is $\langle w_r^{(t)}, v_{i,l}\rangle \leq -\tilde{\Omega}(\sigma_0)$ to see how negative the next iteration will be. Without loss of generality, we consider the case when $\langle w_r^{(t')}, v_{i,l}\rangle \leq -\tilde{\Omega}(\sigma_0)$ holds for every $t' \geq t$. Now based on Claim E.4, we have

$$\langle w_r^{(t+1)}, v_{i,l}\rangle = \langle w_r^{(t)}, v_{i,l}\rangle + \eta\mathbb{E}_{(X,y)\sim\mathcal{Z}}\bigg[V_{r,i,l}(X)\Delta_{r,i,l}(X) - W_{r,i,l}(X) + \Delta_{r,i,l}(X)(\mathcal{E}_1+\mathcal{E}_2) - \mathcal{E}_3 - \mathcal{E}_4$$

$$\pm(V_{r,i,l}(X)+\hat{V}_{r,i,l}(X)(\gamma+\sigma_p))(\mathcal{E}_5+\mathcal{E}_6)\pm(\mathcal{E}_5+\mathcal{E}_6)(\mathcal{E}_1+\mathcal{E}_2)\bigg]$$

$$\overset{(a)}{\geq}\langle w_r^{(t)}, v_{i,l}\rangle + \eta\mathbb{E}_{(X,y)\sim\mathcal{Z}}\bigg[-\mathcal{E}_3 - \mathcal{E}_4 - (\mathcal{E}_5+\mathcal{E}_6)(\mathcal{E}_1+\mathcal{E}_2)\bigg]$$

$$\geq\langle w_r^{(t)}, v_{i,l}\rangle - \eta\bigg[\mathcal{E}_3 + \mathcal{E}_4 + (\mathcal{E}_5+\mathcal{E}_6)(\mathcal{E}_1+\mathcal{E}_2)\bigg]$$

where (a) is because that as we assume $\langle w_r^{(t)}, v_{i,l}\rangle \leq -\tilde{\Omega}(\sigma_0)$, we have

$$W_{r,i,l}(X) = \left(\frac{1}{\theta}-1\right)\mathbb{I}_{\{v_{i,l}\in\mathcal{V}(X)\}}\sum_{p\in\mathcal{P}_{v_{i,l}}(X)}\overline{\text{ReLU}}(\langle w_r, x_p\rangle)\overline{\text{ReLU}}'(\langle w_r, x_p\rangle)z_p$$

$$= \left(\frac{1}{\theta}-1\right)\mathbb{I}_{\{v_{i,l}\in\mathcal{V}(X)\}}\sum_{p\in\mathcal{P}_{v_{i,l}}(X)}\overline{\text{ReLU}}(\langle w_r, v_{i,l}\rangle z_p\pm\tilde{o}(\sigma_0))\overline{\text{ReLU}}'(\langle w_r, v_{i,l}\rangle z_p\pm\tilde{o}(\sigma_0))z_p$$

$$= 0,$$

and similar results also hold for $\Delta_{r,i,l}, V_{r,i,l}, \hat{V}_{r,i,l}$. This shows that when $t \leq T_0$,

$$
\begin{aligned}
\langle w_r^{(t+1)}, v_{i,l} \rangle &\geq \langle w_r^{(t)}, v_{i,l} \rangle - \eta \tilde{O}(\sigma_0^{(2q-1)}) \cdot (\gamma + \sigma_p) \cdot s^2 - \eta \tilde{O}((\sigma_0 \gamma k)^{2q-1}) \cdot (\gamma + \sigma_p) P^2 \\
&\quad - \eta \tilde{O}(\sigma_0^{2q-1}(\gamma k)^{q-1}) \cdot (\gamma + \sigma_p) Ps \\
&\geq -\tilde{O}(\sigma_0) - \eta T_0 \tilde{O}(\sigma_0^{(2q-1)}) \cdot (\gamma + \sigma_p) \cdot s^2 - \eta T_0 \tilde{O}((\sigma_0 \gamma k)^{2q-1}) \cdot (\gamma + \sigma_p) P^2 \\
&\quad - \eta T_0 \tilde{O}(\sigma_0^{2q-1}(\gamma k)^{q-1}) \cdot (\gamma + \sigma_p) Ps \\
&\geq -\tilde{O}(\sigma_0) - \tilde{O}\big(\sigma_0^2 + \frac{k\sigma_0}{\sqrt{d}}\big) - \tilde{O}\big(\sigma_0^2 + \frac{k\sigma_0}{\sqrt{d}}\big)(\gamma k)^{2q-1} \cdot P^2 \\
&\geq -\tilde{O}(\sigma_0).
\end{aligned}
$$

When $t \in [T_0, T]$, we have

$$
\begin{aligned}
\langle w_r^{(t)}, v_{i,l} \rangle &\geq \langle w_r^{(T_0)}, v_{i,l} \rangle - \eta(T - T_0)\tilde{O}(\sigma_0^{(2q-1)}) \cdot (\gamma + \sigma_p) \cdot s^2 - \eta(T - T_0)\tilde{O}((\sigma_0 \gamma k)^{2q-1}) \cdot (\gamma + \sigma_p) P^2 \\
&\quad - \eta(T - T_0)\tilde{O}(\sigma_0^{2q-1}(\gamma k)^{q-1}) \cdot (\gamma + \sigma_p) Ps \\
&\geq -\tilde{O}(\sigma_0).
\end{aligned}
$$

$\square$

### F.2 OFF-DIAGONAL CORRELATIONS

**Lemma F.3.** *Suppose Assumption C.2 holds and Induction Hypothesis C.3 holds for all iterations $< t$. Then*

$$
\forall v_{i,l} \in \mathcal{V}, \forall r \in \mathcal{M}_{i,l}^{(0)}, \text{ for } v_{j,l'} \neq v_{i,l}: \quad |\langle w_r^{(t)}, v_{j,l'} \rangle| \leq \tilde{O}(\sigma_0).
$$

*Proof of Lemma F.3.* For every $r \in \mathcal{M}_{i,l}^{(0)}$, using Claim E.4, we have

$$
\begin{aligned}
|\langle w_r^{(t+1)}, v_{j,l'} \rangle| &\leq |\langle w_r^{(t)}, v_{j,l'} \rangle| + \eta \mathbb{E}_{(X,y) \sim Z} \bigg[ \Big( \hat{V}_{r,i,l}(X) \Delta_{r,i,l}(X) - \hat{W}_{r,i,l}(X) \Big)(\gamma + \sigma_p) \\
&\quad + \hat{V}_{r,i,l}(X)(\mathcal{E}_5 + \mathcal{E}_6)(\gamma + \sigma_p) + U_{r,j,l'}(X) + \Delta_{r,i,l}(X)(\mathcal{E}_1 + \mathcal{E}_2) \\
&\quad + (\mathcal{E}_5 + \mathcal{E}_6)(\mathcal{E}_1 + \mathcal{E}_2) - \mathcal{E}_3 - \mathcal{E}_4 \bigg]
\end{aligned}
$$

**Stage I.** We first consider the stage when $t \leq T_0$. In this stage, similar to the analysis in the proof of Claim E.5, we have that

$$
\begin{aligned}
&\mathbb{E}_{(X,y) \sim Z} \bigg[ \Big( \hat{V}_{r,i,l}(X) \Delta_{r,i,l}(X) - \hat{W}_{r,i,l}(X) \Big)(\gamma + \sigma_p) \bigg] \\
&\leq \tilde{O}\left(\frac{1}{k}\right) \cdot (\gamma + \sigma_p)\overline{\text{ReLU}}(\langle w_r^{(t)}, v_{i,l} \rangle)\overline{\text{ReLU}}'(\langle w_r^{(t)}, v_{i,l} \rangle),
\end{aligned}
$$

where $\tilde{O}\left(\frac{1}{k}\right)$ is the probability of $v_{i,l} \in \mathcal{V}(X)$, and

$$
\mathbb{E}_{(X,y) \sim Z} \bigg[ \hat{V}_{r,i,l}(X)(\mathcal{E}_5 + \mathcal{E}_6)(\gamma + \sigma_p) \bigg] \leq \tilde{O}\left(\frac{1}{k}\right) \cdot (\mathcal{E}_1 + \mathcal{E}_2)\overline{\text{ReLU}}'(\langle w_r^{(t)}, v_{i,l} \rangle),
$$

and

$$
\mathbb{E}_{(X,y) \sim Z} \bigg[ U_{r,j,l'}(X) \bigg] \leq \tilde{O}\left(\frac{1}{k}\sigma_0^{2q-1}\right),
$$

where $\tilde{O}\left(\frac{1}{k}\right)$ is the probability of $v_{j,l'} \in \mathcal{V}(X)$. Thus, when $t \leq T_0$, we also keep the main increasin term and obtain that

$$
|\langle w_r^{(t)}, v_{j,l'} \rangle| \leq |\langle w_r^{(0)}, v_{j,l'} \rangle| + \tilde{O}\left(\frac{\eta}{k}\right) \cdot (\gamma + \sigma_p) \sum_{t=0}^{T_0} \overline{\text{ReLU}}(\Lambda_{i,l}^{(t)})\overline{\text{ReLU}}'(\Lambda_{i,l}^{(t)}) \tag{8}
$$

From Claim E.5, we have that

$$\tilde{\Theta}\left(\frac{\eta}{k}\right)\sum_{t=0}^{T_0-1}\overline{\text{ReLU}}(\Lambda_{i,l}^{(t)})\overline{\text{ReLU}}'(\Lambda_{i,l}^{(t)}) = \sum_{t=0}^{T_0-1}\Lambda_{i,l}^{(t+1)} - \sum_{t=0}^{T_0-1}\Lambda_{i,l}^{(t)}$$

$$= \Lambda_{i,l}^{(T_0)} - \Lambda_{i,l}^{(0)} \leq \frac{1}{\text{polylog}(k)}. \tag{9}$$

Putting (9) into (8), we have that for every $t \leq T_0$,

$$|\langle w_r^{(t)}, v_{j,l'}\rangle| \leq |\langle w_r^{(0)}, v_{j,l'}\rangle| + \tilde{O}\left(\frac{\sigma_0}{k} + \frac{1}{\sqrt{d}}\right) + \tilde{O}\left(\frac{\eta}{k}\right)\cdot(\gamma+\sigma_p)\overline{\text{ReLU}}(\Lambda_{i,l}^{(T_0)})\overline{\text{ReLU}}'(\Lambda_{i,l}^{(T_0)})$$

$$\leq \tilde{O}(\sigma_0).$$

**Stage II.** In the second stage, when $t \geq T_0$, we have

$$\mathbb{E}_{(X,y)\sim\mathcal{Z}}\left[\left(\hat{V}_{r,i,l}(X)\Delta_{r,i,l}(X) - \hat{W}_{r,i,l}(X)\right)(\gamma+\sigma_p)\right]$$

$$\leq \tilde{O}\left(\frac{1}{kT_0^{1/q}}\right)\cdot(\gamma+\sigma_p)\overline{\text{ReLU}}(\langle w_r^{(t)}, v_{i,l}\rangle)\overline{\text{ReLU}}'(\langle w_r^{(t)}, v_{i,l}\rangle)$$

$$\leq \tilde{O}\left(\frac{1}{kT_0^{1/q}}\right)\cdot(\gamma+\sigma_p),$$

where the first inequality is from Lemma F.1. Thus, when $t \in [T_0, T]$

$$|\langle w_r^{(t)}, v_{j,l'}\rangle| \leq |\langle w_r^{(T_0)}, v_{j,l'}\rangle| + \tilde{O}\left(\frac{\eta(T-T_0)}{kT_0^{1/q}}\right)\cdot(\gamma+\sigma_p)$$

$$\leq |\langle w_r^{(T_0)}, v_{j,l'}\rangle| + \tilde{O}(\sigma_0/k) + \tilde{O}(1/\sqrt{d})$$

$$\leq \tilde{O}(\sigma_0)$$

Combining all above results, we complete our proof. □

## F.3 LOTTERY WINING: KERNELS INSIDE $\mathcal{M}_{i,l}^{(0)}$

In this subsection, we prove that the feature $v_{i,l}$ captured by kernels not in $\mathcal{M}_{i,l}^{(0)}$ is negligible. To prove this result, we first need a lemma from (Allen-Zhu & Li, 2020, Lemma C.19) that compare the growth speed of two sequences of updates of the form $x_{t+1} \leftarrow x_t + \eta C_t x_t^{q-1}$.

**Lemma F.4.** *Let $q \geq 3$ be a constant and $x_0, y_0 = o(1)$. Let $\{x_t, y_t\}_{t\geq 0}$ be two positive sequences updated as*

- *$x_{t+1} \geq x_t + \eta C_t x_t^{q-1}$ for some $C_t = \Theta(1)$,*

- *$y_{t+1} \leq y_t + \eta S C_t y_t^{q-1}$ for some constant $S = \Theta(1)$.*

*Suppose $x_0 \geq y_0 S^{1/(q-2)}\left(1 + \frac{1}{\text{polylog}(k)}\right)$, then we must have for every $A = O(1)$, let $T_x$ be the first iteration such that $x_t \geq A$, then*

$$yT_x \leq O(y_0 \cdot \text{polylog}(k)).$$

Now we begin to prove our result.

**Lemma F.5.** *Suppose Assumption C.2 holds and Induction Hypothesis C.3 holds for all iterations $< t$. Then*

$$\forall v_{i,l} \in \mathcal{V}, \forall r \notin \mathcal{M}_{i,l}^{(0)}: \quad \langle w_r^{(t)}, v_{i,l}\rangle \leq \tilde{O}(\sigma_0).$$

*Proof of Lemma F.5.* When $r \in \mathcal{M}_{j,l'}^{(0)}, (v_{j,l'} \neq v_{i,l})$, we have prove that $\langle w_r^{(t)}, v_{i,l}\rangle \leq \tilde{O}(\sigma_0)$ in Lemma F.3. So we only prove the case when $r \notin \cup_{i\in[k],l\in[2]}\mathcal{M}_{i,l}^{(0)}$.

We assume that there exists an $w_{r'} \notin \cup_{i \in [k], l \in [2]} \mathcal{M}_{i,l}^{(0)}$ such that induction hypothesis C.3 (a)-(c) holds for every $(X, y) \in \mathcal{Z}$. We want to see if the sequence $\langle w_{r'}^{(t)}, v_{i,l} \rangle$ will increase more quickly than $\max_{r \in \mathcal{M}_{i,l}^{(0)}} \langle w_r^{(t)}, v_{i,l} \rangle$. Under this assumption, we have that (here we also only keep the main increasing term),

$$\langle w_{r'}^{(t+1)}, v_{i,l} \rangle = \langle w_{r'}^{(t)}, v_{i,l} \rangle + \eta \mathbb{E}_{(X,y) \sim \mathcal{Z}} \left[ V_{r,i,l}(X) \Delta_{r,i,l}(X) - W_{r,i,l}(X) \right].$$

**Stage I** We first consider when $t \leq T_0$. In this stage, $\Lambda_{i,l}^{(t)} \leq \varrho$. We define two sequences. First, we take $w_{r^*} = \operatorname{argmax}_{r \in \mathcal{M}_{i,l}^{(0)}} \langle w_r^{(0)}, v_{i,l} \rangle$ and define $x_t := \langle w_{r^*}^{(t)}, v_{i,l} \rangle \cdot \left( \frac{s}{qk} \right)^{1/2q} \frac{1}{\varrho^{(2q-1)/2q}}$. We also define $y_t = \max \{ \langle w_{r'}^{(t)}, v_{i,l} \rangle \cdot \left( \frac{s}{qk} \right)^{1/2q} \frac{1}{\varrho^{(2q-1)/2q}}, \sigma_0 \}$. From Claim E.5, when $t \leq T_0$, we have that

$$\langle w_{r^*}^{(t+1)}, v_{i,l} \rangle = \langle w_{r^*}^{(t)}, v_{i,l} \rangle + \Theta \left( \frac{s\eta}{k} \right) \overline{\text{ReLU}}(\langle w_{r^*}^{(t)}, v_{i,l} \rangle) \overline{\text{ReLU}}'(\langle w_{r^*}^{(t)}, v_{i,l} \rangle)$$

$$\geq \langle w_{r^*}^{(t)}, v_{i,l} \rangle + \Theta \left( \frac{s\eta}{k} \right) \frac{1}{q\varrho^{2q-1}} ([\langle w_{r^*}, v_{i,l} \rangle]^+)^{2q-1}.$$

Let $S = \left( \frac{1 + C/(\log(k) - C)}{1 + 1/\log(k)} \right)^{q-2}, C > 1$. We have

$$\langle w_{r'}^{(t+1)}, v_{i,l} \rangle = \langle w_{r'}^{(t)}, v_{i,l} \rangle + \Theta \left( \frac{s\eta}{k} \right) \overline{\text{ReLU}}(\langle w_{r'}^{(t)}, v_{i,l} \rangle) \overline{\text{ReLU}}'(\langle w_{r'}^{(t)}, v_{i,l} \rangle)$$

$$\leq \langle w_{r'}^{(t)}, v_{i,l} \rangle + \Theta \left( \frac{s\eta}{k} \right) \frac{1}{q\varrho^{2q-1}} ([\langle w_{r'}^{(t)}, v_{i,l} \rangle]^+)^{2q-1} S.$$

Set $C_t = 1$. Then we have that

$$x_{t+1} \geq x_t + \eta C_t x_t^{2q-1},$$
$$y_{t+1} \leq y_t + \eta S C_t y_t^{2q-1}.$$

Besides, $x_0 = \Lambda_{i,l}^{(0)}$ and $y_0 \leq \Lambda_{i,l}^{(0)}(1 - O(1/\log(k)))$ based on the definition of $\mathcal{M}_{i,l}^{(0)}$. Here we assume $y_0 \leq \Lambda_{i,l}^{(0)}(1 - C/\log(k))$. Thus, we have

$$x_0 \geq y_0 \left( 1 + \frac{C}{\log(k) - C} \right) = y_0 S^{\frac{1}{q-2}} \left( 1 + \frac{1}{\log(k)} \right).$$

So using the result from Lemma F.4, when $\langle w_{r^*}^{(t+1)}, v_{i,l} \rangle$ reaches $\tilde{\Omega}(1)$, which necessarily is an iteration $t \geq T_0$, we still have that

$$y_t \leq \tilde{O}(y_0) \implies \langle w_{r'}^{(t)}, v_{i,l} \rangle \leq \tilde{O}(\sigma_0).$$

**Stage II** We now consider when $t \in [T_0, T]$. In this stage, using the induction hypothesis C.3 (d) and (e), we have that

$$\mathbb{E}_{(X,y) \sim \mathcal{Z}} \left[ \langle \nabla_{w_r} L(H; X), v_{i,l} \rangle \right] \leq \tilde{O} \left( \frac{1}{k} \sigma_0^{2q-1} \right).$$

Thus,

$$\langle w_{r'}^{(t+1)}, v_{i,l} \rangle \leq \langle w_{r'}^{(t)}, v_{i,l} \rangle + \tilde{O} \left( \frac{\eta}{k} \sigma_0^{2q-1} \right)$$

$$\leq \langle w_{r'}^{(T_0)}, v_{i,l} \rangle + \tilde{O} \left( \frac{\eta(T - T_0)}{k} \sigma_0^{2q-1} \right)$$

$$\leq \tilde{O}(\sigma_0).$$

$\square$

### F.4 Noise Correlation

In this subsection, we prove that the kernels correlate small with the random noise.

**Lemma F.6.** *Suppose Assumption C.2 holds and Induction Hypothesis C.3 holds for all iterations $< t$. For every $v_{i,l} \in \mathcal{V}$, for every $r \in \mathcal{M}_{i,l}^{(0)}$, for every $(X,y) \in \mathcal{Z}$, we have*

(a) *For every $p \in \mathcal{P}_{v_{i,l}}(X)$, $|\langle w_r^{(t)}, \xi_p \rangle| \leq \tilde{o}(\sigma_0)$.*

(b) *For every $p \in \mathcal{P}(X) \setminus \mathcal{P}_{v_{i,l}}(X)$, $|\langle w_r^{(t)}, \xi_p \rangle| \leq \tilde{O}(\sigma_0)$.*

(c) *For every $p \in [P] \setminus \mathcal{P}(X)$, $|\langle w_r^{(t)}, \xi_p \rangle| \leq \tilde{O}(\sigma_0 \gamma k)$.*

*Moreover, for every $r \notin \cup_{i \in [k], l \in [2]} \mathcal{M}_{i,l}^{(0)}$, for every $(X,y) \in \mathcal{Z}$, we have*

(d) *for every $p \in \mathcal{P}(X)$, $|\langle w_r^{(t)}, \xi_p \rangle| \leq \tilde{O}(\sigma_0)$.*

(e) *for every $p \in [P] \setminus \mathcal{P}(X)$, $|\langle w_r^{(t)}, \xi_p \rangle| \leq \tilde{O}(\sigma_0 \gamma k)$.*

*Proof of Lemma F.6.* For every $r \in [km]$, for every $(X^*, y^*) \in \mathcal{Z}$ and every $p^* \in [P]$, we have that

$$\langle -\nabla_{w_r} L(X), \xi_{p^*} \rangle = \sum_{p \in [P]} \left( \Phi_r(X) - \left( \frac{1}{\theta} - 1 \right) \overline{\text{ReLU}}(\langle w_r, x_p \rangle) \right) \overline{\text{ReLU}}'(\langle w_r, x_p \rangle) \langle x_p, \xi_{p^*} \rangle.$$

When $X \neq X^*$, we have $|\langle x_p, \xi_{p^*} \rangle| \leq \tilde{O}(\sigma_p) \leq o(1/\sqrt{d})$; and when $X = X^*$ but $p \neq p^*$, we have $|\langle x_p, \xi_{p^*} \rangle| \leq \tilde{O}(\sigma_p) \leq o(1/\sqrt{d})$. Therefore, we have

$$\mathbb{E}_{(X,y) \sim \mathcal{Z}} \left[ \langle -\nabla_{w_r} L(X), \xi_{p^*} \rangle \right] = \mathbb{E}_{(X,y) \in \mathcal{Z}} \left[ \mathbb{I}_{X=X^*} \langle -\nabla_{w_r} L(X), \xi_{p^*} \rangle + \mathbb{I}_{X \neq X^*} \langle -\nabla_{w_r} L(X), \xi_{p^*} \rangle \right].$$

For the first term,

$$\mathbb{E}_{(X,y) \sim \mathcal{Z}} \left[ \mathbb{I}_{X=X^*} \langle -\nabla_{w_r} L(X), \xi_{p^*} \rangle \right]$$

$$= \frac{1}{N} \mathbb{E}_{(X^*,y^*) \sim \mathcal{Z}} \left[ \Phi_r(X^*) \overline{\text{ReLU}}'(\langle w_r, x_{p^*} \rangle) \langle x_{p^*}, \xi_{p^*} \rangle \right.$$

$$\left. - \left( \frac{1}{\theta} - 1 \right) \overline{\text{ReLU}}(\langle w_r, x_{p^*} \rangle) \overline{\text{ReLU}}'(\langle w_r, x_{p^*} \rangle) \langle x_{p^*}, \xi_{p^*} \rangle \pm o \left( \frac{1}{\sqrt{d}} \right) \right]$$

$$\stackrel{(a)}{=} \tilde{\Theta} \left( \frac{1}{N} \right) \mathbb{E}_{(X^*,y^*) \sim \mathcal{Z}} \left[ \Phi_r(X^*) \overline{\text{ReLU}}'(\langle w_r, x_{p^*} \rangle) \right.$$

$$\left. - \left( \frac{1}{\theta} - 1 \right) \overline{\text{ReLU}}(\langle w_r, x_{p^*} \rangle) \overline{\text{ReLU}}'(\langle w_r, x_{p^*} \rangle) \pm o \left( \frac{1}{\sqrt{d}} \right) \right]$$

$$= \tilde{\Theta} \left( \frac{1}{N} \right) \mathbb{E}_{(X^*,y^*) \sim \mathcal{Z}} \left[ \mathbb{I}_{v_{i,l} \in \mathcal{V}(X^*)} \sum_{p \in \mathcal{P}_{v_{i,l}}(X)} \left[ \overline{\text{ReLU}}(\langle \hat{w}_r, x_p \rangle) - \overline{\text{ReLU}}(\langle w_r, x_p \rangle) \right] \right.$$

$$\left. \times \overline{\text{ReLU}}'(\langle w_r, x_{p^*} \rangle) - \left( \frac{1}{\theta} - 1 \right) \overline{\text{ReLU}}(\langle w_r, x_{p^*} \rangle) \overline{\text{ReLU}}'(\langle w_r, x_{p^*} \rangle) \pm o \left( \frac{1}{\sqrt{d}} \right) \right]$$

where (a) is because $\|\xi_{p^*}\|_2^2 = \tilde{\Theta}(1)$. For the second term,

$$\mathbb{E}_{(X,y) \sim \mathcal{Z}} \left[ \mathbb{I}_{X \neq X^*} \langle -\nabla_{w_r} L(X), \xi_{p^*} \rangle \right] = \pm o \left( \frac{1}{\sqrt{d}} \right)$$

Now we begin to prove (a). For every $v_{i,l} \in \mathcal{V}$, for every $r \in \mathcal{M}_{i,l}^{(0)}$, for every $p^* \in \mathcal{P}_{v_{i,l}}(X^*)$, using the induction hypothesis C.3, when $t \in [0, T_0]$, we have

$$
\mathbb{E}_{(X,y)\sim\mathcal{Z}}\left[\langle -\nabla_{w_r} L(X), \xi_{p^*}\rangle\right]
$$
$$
= \tilde{\Theta}\left(\frac{1}{N}\right)\mathbb{E}_{(X,y)\sim\mathcal{Z}}\left[\left(\mathbb{I}_{v_{i,l}\in\mathcal{V}(X^*)}\sum_{p\in\mathcal{P}_{v_{i,l}}(X^*)}z_{p^*}^{q-1}\left(\sum_{p'\in\mathcal{P}_{v_{i,l}}(X^*)}(\tau^q-1)z_{p'}^q-\left(\frac{1}{\theta}-1\right)z_{p^*}^q\right)\right)\right.
$$
$$
\left.\times \overline{\mathrm{ReLU}}(\langle w_r, v_{i,l}\rangle)\overline{\mathrm{ReLU}}'(\langle w_r, v_{i,l}\rangle)\right] \pm o\left(\frac{1}{\sqrt{d}}\right).
$$

Thus, we have

$$
\langle w_r^{(t+1)}, \xi_{p^*}\rangle \le \langle w_r^{(t)}, \xi_{p^*}\rangle + \tilde{O}\left(\frac{\eta}{N}\right)\overline{\mathrm{ReLU}}(\langle w_r, v_{i,l}\rangle)\overline{\mathrm{ReLU}}'(\langle w_r, v_{i,l}\rangle) + o\left(\frac{\eta}{\sqrt{d}}\right),
$$

Now we use the results from Lemma F.1, when $t \le T_0$,

$$
\langle w_r^{(t)}, \xi_{p^*}\rangle \le \langle w_r^{(0)}, \xi_{p^*}\rangle + \tilde{O}\left(\frac{\eta T_0}{N}\right) + o\left(\frac{\eta T_0}{\sqrt{d}}\right)
$$

So when $N \ge \tilde{\omega}\left(\frac{k}{\sigma_0^{2q-1}}\right)$ and $\sqrt{d} \ge \tilde{\omega}(k/\sigma_0^{2q-1})$, we have $\langle w_r^{(t)}, \xi_{p^*}\rangle \le \tilde{o}(\sigma_0)$. Then when $t \in [T_0, T]$, we have

$$
\mathbb{E}_{(X,y)\in\mathcal{Z}}\left[\mathbb{I}_{X=X^*}\langle -\nabla_{w_r} L(X), \xi_{p^*}\rangle\right]
$$
$$
= \tilde{\Theta}\left(\frac{1}{NT_0^{1/q}}\right)\mathbb{E}_{(X,y)\sim\mathcal{Z}}\left[\mathbb{I}_{v_{i,l}\in\mathcal{V}(X^*)}\overline{\mathrm{ReLU}}(\langle w_r, v_{i,l}\rangle)\overline{\mathrm{ReLU}}'(\langle w_r, v_{i,l}\rangle)\right] \pm o\left(\frac{1}{N\sqrt{d}}\right).
$$

Therefore, for $t \in [T_0, T]$, we have

$$
\langle w_r^{(t)}, \xi_{p^*}\rangle \le \langle w_r^{(T_0)}, \xi_{p^*}\rangle + \tilde{O}\left(\frac{\eta(t-T_0)}{NT_0^{1/q}}\right) + o\left(\frac{\eta(t-T_0)}{\sqrt{d}}\right) \le \tilde{o}(\sigma_0),
$$

when $\sqrt{d} \ge \tilde{\omega}(k^{5/2}/\eta^{1/q})$.

Now we begin to prove (b). For every $p^* \in \mathcal{P}(X^*) \setminus \mathcal{P}_{v_{i,l}}(X^*)$, using the induction hypothesis C.3, when $t \in [0, T_0]$, we have

$$
\mathbb{E}_{(X,y)\in\mathcal{Z}}\left[\mathbb{I}_{X=X^*}\langle -\nabla_{w_r} L(X), \xi_{p^*}\rangle\right]
$$
$$
\le \tilde{O}\left(\frac{1}{N}\right)\mathbb{E}_{(X^*,y^*)\sim\mathcal{Z}}\left[\mathbb{I}_{v_{i,l}\in\mathcal{V}(X^*)}\sum_{p\in\mathcal{P}_{v_{i,l}}(X)}\left[\overline{\mathrm{ReLU}}(\langle \hat{w}_r, x_p\rangle) - \overline{\mathrm{ReLU}}(\langle w_r, x_p\rangle)\right]\tilde{O}(\sigma_0^{(q-1)}) \pm o\left(\frac{1}{\sqrt{d}}\right)\right]
$$
$$
\le \tilde{O}\left(\frac{1}{N}\right)\mathbb{E}_{(X^*,y^*)\sim\mathcal{Z}}\left[\tilde{O}(\sigma_0^{(q-1)}) \pm o\left(\frac{1}{\sqrt{d}}\right)\right]
$$

Thus, when $t \le T_0$, we have

$$
\langle w_r^{(t)}, \xi_{p^*}\rangle \le \langle w_r^{(0)}, \xi_{p^*}\rangle + \tilde{O}\left(\frac{\eta T_0}{N}\sigma_0^{(q-1)}\right) + o\left(\frac{\eta T_0}{\sqrt{d}}\right) \le \tilde{O}(\sigma_0),
$$

when $N \ge \frac{k}{\sigma_0^q}$ and $\sqrt{d} \ge k/\sigma_0^{2q-1}$. Then when $t \in [T_0, T]$, we have

$$
\langle w_r^{(t)}, \xi_{p^*}\rangle \le \langle w_r^{(T_0)}, \xi_{p^*}\rangle + \tilde{O}\left(\frac{\eta(t-T_0)}{N}\sigma_0^{(q-1)}\right) + o\left(\frac{\eta(t-T_0)}{\sqrt{d}}\right) \le \tilde{O}(\sigma_0).
$$

We begin to prove (c). For every $p \in [P] \setminus \mathcal{P}(X)$, using the induction hypothesis C.3, when $t \in [0, T_0]$, we have

$$
\mathbb{E}_{(X,y)\in\mathcal{Z}}\left[\mathbb{I}_{X=X^*}\langle -\nabla_{w_r} L(X), \xi_{p^*}\rangle\right] \le \tilde{O}\left(\frac{1}{N}\right)\mathbb{E}_{(X^*,y)\sim\mathcal{Z}}\left[\tilde{O}((\sigma_0\gamma k)^{(q-1)}) \pm o\left(\frac{1}{\sqrt{d}}\right)\right].
$$

Then the process to prove (c) is similar to the proof of (b).

To prove (d) and (e), for every $p^* \in \mathcal{P}(X^*)$, using the induction hypothesis C.3, we have

$$\mathbb{E}_{(X,y) \in \mathcal{Z}}\left[\mathbb{I}_{X=X^*}\langle -\nabla_{w_r} L(X), \xi_{p^*}\rangle\right] \leq \tilde{O}\left(\frac{1}{N}\right)\left[\tilde{O}(\sigma_0^{(2q-1)}) \pm o\left(\frac{1}{\sqrt{d}}\right)\right].$$

and for every $p^* \in [P] \setminus \mathcal{P}(X^*)$, using the induction hypothesis C.3, we have

$$\mathbb{E}_{(X,y) \in \mathcal{Z}}\left[\mathbb{I}_{X=X^*}\langle -\nabla_{w_r} L(X), \xi_{p^*}\rangle\right] \leq \tilde{O}\left(\frac{1}{N}\right)\left[\tilde{O}((\sigma_0\gamma k)^{(2q-1)}) \pm o\left(\frac{1}{\sqrt{d}}\right)\right].$$

Following the similar process, we could also prove (d) and (e). □

### F.5 PROOF OF THEOREM C.4

In this subsection, we will combine all lemmas and begin to prove Theorem C.4.

*Proof of Theorem C.4.* At iteration $t$, according to the data structure we defined in Definition 1, we have

$$\forall p \in \mathcal{P}_{v_{i,l}}(X): \quad \langle w_r^{(t)}, x_p\rangle = \langle w_r^{(t)}, v_{i,l}\rangle z_p + \sum_{v' \in \mathcal{V}} \alpha_{p,v'}\langle w_r^{(t)}, v'\rangle + \langle w_r^{(t)}, \xi_p\rangle, \quad (10)$$

$$\forall p \in [P] \setminus \mathcal{P}(X): \quad \langle w_r^{(t)}, x_p\rangle = \sum_{v' \in \mathcal{V}} \alpha_{p,v'}\langle w_r^{(t)}, v'\rangle + \langle w_r^{(t)}, \xi_p\rangle. \quad (11)$$

It is easy to verify the induction hypothesis C.3 holds at iteration $t = 0$. Suppose induction hypothesis C.3 holds for all iteration $< t$. We have established several lemmas:

$$\text{Lemma F.3} \implies \forall v_{i,l} \in \mathcal{V}, \forall r \in \mathcal{M}_{i,l}^{(0)}, \text{ for } v_{j,l'} \neq v_{i,l}: |\langle w_r^{(t)}, v_{j,l'}\rangle| \leq \tilde{O}(\sigma_0) \tag{12}$$

$$\text{Lemma F.2 and Lemma F.1} \implies \forall v_{i,l} \in \mathcal{V}, \forall r \in \mathcal{M}_{i,l}^{(0)}: \langle w_r^{(t)}, v_{i,l}\rangle \in [-\tilde{O}(\sigma_0), \tilde{O}(1)] \tag{13}$$

$$\text{Lemma F.5} \implies \forall v_{i,l} \in \mathcal{V}, \forall r \notin \mathcal{M}_{i,l}^{(0)}: \langle w_r^{(t)}, v_{i,l}\rangle \leq \tilde{O}(\sigma_0). \tag{14}$$

- To prove Induction Hypothesis C.3(a), we plug (12) and (13) into (10), and use $\alpha_{p,v'} =\in [0, \gamma]$, $|\mathcal{V}| = 2k$ and $|\langle w_r^{(t)}, \xi_p\rangle| \leq \tilde{o}(\sigma_0)$ from Lemma F.6(a).

- To prove Induction Hypothesis C.3(b), we plug (12) and (13) into (10), and use $\alpha_{p,v'} =\in [0, \gamma]$, $|\mathcal{V}| = 2k$ and $|\langle w_r^{(t)}, \xi_p\rangle| \leq \tilde{O}(\sigma_0)$ from Lemma F.6(b).

- To prove Induction Hypothesis C.3(c), we plug (12) and (13) into (11), and use $\alpha_{p,v'} =\in [0, \gamma]$, $|\mathcal{V}| = 2k$ and $|\langle w_r^{(t)}, \xi_p\rangle| \leq \tilde{O}(\sigma_0\gamma k)$ from Lemma F.6(c).

- To prove Induction Hypothesis C.3(d), we plug (14) into (10), and use $\alpha_{p,v'} =\in [0, \gamma]$, $|\mathcal{V}| = 2k$ and $|\langle w_r^{(t)}, \xi_p\rangle| \leq \tilde{O}(\sigma_0)$ from Lemma F.6(d).

- To prove Induction Hypothesis C.3(e), we plug (14) into (11), and use $\alpha_{p,v'} =\in [0, \gamma]$, $|\mathcal{V}| = 2k$ and $|\langle w_r^{(t)}, \xi_p\rangle| \leq \tilde{O}(\sigma_0\gamma k)$ from Lemma F.6(e).

- Induction Hypothesis C.3 (f), (g) and (h) are easily obtained from Lemma F.2, Lemma F.1 and Lemma F.5.

□

## G   TEST PERFORMANCE ON DOWNSTREAM CLASSIFICATION TASKS

In this section, we analyze the performance of mask-reconstruction pretraining on downstream classification tasks to show its superiority over supervised training.

## G.1 Main Results

We add an extra linear layer on the pretrained encoder. We collect labeled data points $\mathcal{Z}_{\text{down}} = \{(X_i, y_i)\}_{i=1}^{N_2} \sim \mathcal{D}$ and use these labeled data points to update the weights $u_{i,r}, i \in [k], r \in [km]$ of the extra linear layer and fine-tune the kernels of the pretrained encoder $w_r, r \in [km]$. The output of linear layer is denoted as $F_i(X) = \sum_{r \in [km]} u_{i,r} h_r(X)$. The loss function on downstream tasks is

$$L_{\text{down}}(F) = \frac{1}{N_2} \sum_{i \in [N_2]} L_{\text{down}}(F; X_i, y_i),$$

where $L_{\text{down}}(F; X, y) = -\log \frac{e^{F_y(X)}}{\sum_{j \in [k]} e^{F_j(X)}}$. We define $\text{logit}_i(F; X) = \frac{e^{F_y(X)}}{\sum_{j \in [k]} e^{F_j(X)}}$. The gradient of $L_{\text{down}}(F; X, y)$ is

$$-\nabla_{u_{i,r}} L_{\text{down}}(F; X, y) = (\mathbb{I}_{i=y} - \text{logit}_i(F; X)) h_r(X).$$

We initialize $u_{i,r}^{(0)} = 0, i \in [k], r \in [km]$ and the initialization of $w_r^{(0)}, r \in [km]$ is $w_r^{(T)}$, i.e., kernels of the pretrained encoder. We update the weights using gradient descent:

$$u_{i,r}^{(t+1)} = u_{i,r}^{(t)} - \eta_2 \nabla_{u_{i,r}} L_{\text{down}}(F; X, y),$$
$$w_r^{(t+1)} = w_r^{(t)} - \eta_1 \nabla_{w_r} L_{\text{down}}(F; X, y). \tag{15}$$

We set $\eta_1$ to be much smaller than $\eta_2$.

The following lemma states that the induction hypothesis C.3 still holds in the training of classification tasks.

**Lemma G.1.** *For $N_2 \geq k$ many samples, setting the learning rate $\eta_2 = \Theta(k)$ and $\eta_1 \leq \tilde{\Theta}(k)$, after $T_{\text{down}} \geq \frac{\text{poly}(k)}{\eta_1 \eta_2}$ many iterations, for sufficiently large $k > 0$, Induction Hypothesis C.3 holds for all iterations with high probability.*

Then we have the following theorem showing the performance of downstream classification test.

**Theorem G.2** (Performance on downstream classification tasks). *For $N_2 \geq k$ many samples, setting the learning rate $\eta_2 = \Theta(k)$ and $\eta_1 \leq \tilde{\Theta}(k)$, after $T_{\text{down}} \geq \frac{\text{poly}(k)}{\eta_1 \eta_2}$ many iterations, with high probability, we have*

(a) *(training loss is small) for every $(X, y) \in \mathcal{Z}_{\text{down}}$, i.e.,*

$$L_{\text{down}}(F) = \mathbb{E}_{(X,y) \sim \mathcal{Z}_{\text{down}}} [L_{\text{down}}(F; X, y)] \leq \frac{1}{\text{poly}(k)}.$$

(b) *(test performance is good) for new data point $(X, y) \sim \mathcal{D}$, the test performance is*

$$\Pr_{(X,y) \in \mathcal{D}} \left[ F_y(X) \geq \max_{j \neq y} F_j(X) + \tilde{O}(1) \right] \geq 1 - e^{-\Omega(\log^2 k)}.$$

## G.2 Finetuning of downstream classification models

In this subsection, we fine-tune the weights $w_r, r \in [km]$ of pretrained encoder of Student network and update the weights of the linear layer $u_{i,r}, i \in [k], r \in [km]$.

### G.2.1 Updates of $u_{i,r}$

We first define several terms which will be used frequently.

**Definition G.3.**

$$Z_{i,l}(X) = \mathbb{I}_{v_{i,l} \in \mathcal{V}(X)} \sum_{p \in \mathcal{P}_{v_{i,l}}(X)} z_p,$$

$$\psi_{r,i,l} = [\langle w_r, v_{i,l} \rangle]^+, \quad \Psi_{i,l} = \sum_{r \in \mathcal{M}_{i,l}^{(0)}} \psi_{r,i,l}^2, \quad \Psi_i = \sum_{l \in [2]} \Psi_{i,l}.$$

When $r \in \mathcal{M}_{i,l}^{(0)}$, at $t = 0$, using the induction hypothesis C.3, we have

$$h_r(X) = \mathbb{I}_{v_{i,l} \in \mathcal{V}(X)} \sum_{p \in \mathcal{P}_{v_{i,l}}(X)} \overline{\mathrm{ReLU}}\left(\langle w_r, v_{i,l}\rangle z_p + \tilde{o}(\sigma_0)\right) + \tilde{O}(\sigma_0^q) \cdot (s+1) + \tilde{O}((\sigma_0 \gamma k)^q) \cdot P$$

$$= [\langle w_r, v_{i,l}\rangle]^+ \cdot \mathbb{I}_{v_{i,l} \in \mathcal{V}(X)} \sum_{p \in \mathcal{P}_{v_{i,l}}(X)} z_p + \tilde{O}(\sigma_0^q) \cdot s + \tilde{O}((\sigma_0 \gamma k)^q) \cdot P$$

$$= \psi_{r,i,l} \cdot Z_{i,l}(X) + \mathcal{E}_5 + \mathcal{E}_6.$$

When $r \notin \cup_{i \in [k], l \in [2]} \mathcal{M}_{i,l}^{(0)}$, at $t = 0$, using the induction hypothesis C.3, we have

$$h_r(X) = \tilde{O}(\sigma_0^q)(s+2) + \tilde{O}((\sigma_0 \gamma k)^q) \cdot P.$$

The gradients with respect to the output $F_i(X)$ include three types.

**(1) Near zero gradients**   For $u_{i,r}$, when $r \notin \cup_{i \in [k], l \in [2]} \mathcal{M}_{i,l}^{(0)}$,

$$h_r(X) = \tilde{O}(\sigma_0^q) \cdot (s+2) + \tilde{O}((\sigma_0 \gamma k)^q) \cdot P.$$

which is very small. Thus, there is nearly no updates on those weights and they keep near zero, i.e.,

$$u_{i,r}^{(t)} \approx 0 \quad \text{when } r \notin \cup_{i \in [k], l \in [2]} \mathcal{M}_{i,l}^{(0)}.$$

**(2) Negative gradients**   For $u_{i,r}$, when $r \in \mathcal{M}_{j,l}^{(0)}, j \neq i$, we now show the gradients $-\nabla_{u_{i,r}} L(F; X, y)$ for different type of data points:

    (a)  when $y = i$, for every $(X, y) \sim \mathcal{Z}_{\mathrm{down}}$,

$$-\nabla_{u_{i,r}} L(F; X, y) = (1 - \mathrm{logit}_i(F; X)) h_r(X), \quad \sum_{p \in \mathcal{P}_{v_{j,l}}(X)} z_p \in [\Omega(1), 0.4]$$

    (b)  when $y \neq i$ but $y = j$, for every $(X, y) \in \mathcal{Z}_{\mathrm{down},m}$ or $(X, y) \in \mathcal{Z}_{\mathrm{down},s}, \hat{l} = l$,

$$-\nabla_{u_{i,r}} L(F; X, y) = -\mathrm{logit}_i(F; X) h_r(X), \quad \sum_{p \in \mathcal{P}_{v_{j,l}}(X)} z_p \in [1, O(1)]$$

    (c)  when $y \neq i$ but $y = j$, for every $(X, y) \in \mathcal{Z}_{\mathrm{down},s}, \hat{l} = 3 - l$,

$$-\nabla_{u_{i,r}} L(F; X, y) = -\mathrm{logit}_i(F; X) h_r(X), \quad \sum_{p \in \mathcal{P}_{v_{j,l}}(X)} z_p \in [\rho, O(\rho)]$$

    (d)  when $y \neq i$ and $y \neq j$, for every $(X, y) \in \mathcal{Z}_{\mathrm{down}}$,

$$-\nabla_{u_{i,r}} L(F; X, y) = -\mathrm{logit}_i(F; X) h_r(X), \quad \sum_{p \in \mathcal{P}_{v_{j,l}}(X)} z_p \in [\Omega(1), 0.4]$$

**(3) Positive gradients**   For $u_{i,r}$, we now show the gradients $-\nabla_{u_{i,r}} L(F; X, y)$ when $r \in \mathcal{M}_{i,l}^{(0)}$ for different type of data points:

    (a)  when $y = i$, for every $(X, y) \sim \mathcal{Z}_{\mathrm{down},m}$ or $(X, y) \in \mathcal{Z}_{\mathrm{down},s}, \hat{l} = l$

$$-\nabla_{u_{i,r}} L(F; X, y) = (1 - \mathrm{logit}_i(F; X)) h_r(X), \quad \sum_{p \in \mathcal{P}_{v_{i,l}}(X)} z_p \in [1, O(1)].$$

    (b)  when $y = i$, for every $(X, y) \sim \mathcal{Z}_{\mathrm{down},s}, \hat{l} = 3 - l$,

$$-\nabla_{u_{i,r}} L(F; X, y) = (1 - \mathrm{logit}_i(F; X)) h_r(X), \quad \sum_{p \in \mathcal{P}_{v_{i,l}}(X)} z_p \in [\rho, O(\rho)].$$

(c) when $y \neq i$, for every $(X, y) \in \mathcal{Z}_{\text{down}}$,

$$-\nabla_{u_{i,r}} L(F; X, y) = -\text{logit}_i(F; X) h_r(X), \quad \sum_{p \in \mathcal{P}_{v_{i,l}}(X)} z_p \in [\Omega(1), 0.4].$$

Now we begin to show the full gradients. As we assume the ratio of single-view data is $\mu = \frac{1}{\text{poly}(k)}$, it has little influence on the update of weights. So we ignore single-view data and only focus on $(X, y) \in \mathcal{Z}_{\text{down},m}$. Then when $r \in \mathcal{M}_{j,l}^{(0)}, j \neq i$, we have

$$\mathbb{E}_{(X,y)\sim\mathcal{Z}_{\text{down}}}[-\nabla_{u_{i,r}} L(F; X, y)] = \mathbb{E}_{(X,y)\sim\mathcal{Z}_{\text{down}}}\left[\mathbb{I}_{\{y=i\}}(1 - \text{logit}_i(F; X))\left[\frac{0.4s}{k} \cdot \psi_{r,j,l} + \mathcal{E}_5 + \mathcal{E}_6\right]\right]$$

$$- \mathbb{E}_{(X,y)\sim\mathcal{Z}_{\text{down}}}\left[\mathbb{I}_{\{y=j\}}\text{logit}_i(F; X)\left[O(1) \cdot \psi_{r,j,l} + \mathcal{E}_5 + \mathcal{E}_6\right]\right]$$

$$- \mathbb{E}_{(X,y)\sim\mathcal{Z}_{\text{down}}}\left[\mathbb{I}_{\{y\neq i, y\neq j\}}\text{logit}_i(F; X)\left[\frac{0.4s}{k} \cdot \psi_{r,j,l} + \mathcal{E}_5 + \mathcal{E}_6\right]\right]$$

$$= \frac{k-1}{k^2} \cdot \left[\frac{0.4s}{k} \cdot \psi_{r,j,l} + \mathcal{E}_5 + \mathcal{E}_6\right] - \frac{1}{k^2}\left[\psi_{r,j,l} \cdot O(1) + \mathcal{E}_5 + \mathcal{E}_6\right]$$

$$- \frac{k-2}{k^2}\left[\frac{0.4s}{k} \cdot \psi_{r,j,l} + \mathcal{E}_5 + \mathcal{E}_6\right],$$

where $\frac{1}{k}, \frac{1}{k}, \frac{k-2}{k}$ is the ratios for each type of data and at $t = 0$, we have $\text{logit}_i(F; X) = \frac{1}{k}, i \in [k]$ because we initialize $u_{i,r} = 0$. Therefore, if we ignore the small term, at $t = 1$, we have

$$u_{i,r}^{(1)} \approx \eta_2 \left(\frac{0.4(k-1)s}{k^3} - \frac{O(1)}{k^2} - \frac{0.4(k-2)s}{k^3}\right)\psi_{r,j,l} + \frac{\eta_2(\mathcal{E}_5 + \mathcal{E}_6)}{k}$$

$$\approx \eta_2 \left(\frac{0.4s}{k^3} - \frac{O(1)}{k^2}\right)\psi_{r,j,l} + \frac{\eta_2(\mathcal{E}_5 + \mathcal{E}_6)}{k} < 0. \tag{16}$$

Using this weight, we could also obtain the bounds of loss function after the update of $w_r, r \in [km]$ (we will show the following inequality in (24) after we update $w_r$):

$$0 \leq 1 - \text{logit}_y(F; X) \leq \tilde{O}\left(\frac{1}{k}\right),$$

$$0 \leq \text{logit}_i(F; X) \leq \tilde{O}\left(\frac{1}{k}\right), \quad \forall i \in [k] \setminus y.$$

Thus, at $t = 2$, we have

$$u_{i,r}^{(2)} \geq \eta_2 \left(\frac{0.4s}{k^3} - \frac{O(1)}{k^2}\right)\psi_{r,j,l} + \frac{\eta_2(\mathcal{E}_5 + \mathcal{E}_6)}{k} - \tilde{O}\left(\frac{\eta_2}{k^2}\right)\psi_{r,j,l} - \frac{\eta_2(\mathcal{E}_5 + \mathcal{E}_6)}{k^2},$$

$$u_{i,r}^{(2)} \leq \eta_2 \left(\frac{0.4s}{k^3} - \frac{O(1)}{k^2}\right)\psi_{r,j,l} + \frac{\eta_2(\mathcal{E}_5 + \mathcal{E}_6)}{k} + \tilde{O}\left(\frac{\eta_2}{k^3}\right)\psi_{r,j,l} + \frac{\eta_2(\mathcal{E}_5 + \mathcal{E}_6)}{k^2}.$$

So the approximation of $u_{i,r}^{(2)}$ is

$$u_{i,r}^{(2)} \approx -\tilde{O}\left(\frac{\eta_2}{k^2}\right)\psi_{r,j,l} + \frac{\eta_2(\mathcal{E}_5 + \mathcal{E}_6)}{k}.$$

Then for $t > 2$, as we continue to train to minimize the loss function, $1 - \text{logit}_y(F; X)$ will become smaller and so as $\text{logit}_i(F; X), i \in [k] \setminus y$. So the main term in $u_{i,r}, i \in [k], r \in [km]$ is the term of the first two updates and there is nearly no order changes on values of weights after the first two step of gradient descent. Thus, for simplicity of analysis, we could take

$$u_{i,r}^{(t)} \approx -\tilde{O}\left(\frac{\eta_2}{k^2}\right)\psi_{r,j,l} + \frac{\eta_2(\mathcal{E}_5 + \mathcal{E}_6)}{k}, \quad \text{for } t \geq 2. \tag{17}$$

Similar to the former case, when $r \in \mathcal{M}_{i,l}^{(0)}$, we have

$$\mathbb{E}_{(X,y)\sim\mathcal{Z}_{\text{down}}}[-\nabla_{u_{i,r}}L(F;X,y)] = \frac{k-1}{k^2}[\psi_{r,i,l} \cdot O(1) + \mathcal{E}_5 + \mathcal{E}_6]$$

$$-\frac{k-1}{k^2}\left[\frac{0.4s}{k} \cdot \psi_{r,i,l} + \mathcal{E}_5 + \mathcal{E}_6\right]$$

Then if we ignore the small term, at $t = 1$, we have

$$u_{i,r}^{(1)} \approx \eta_2 \left(\frac{O(1) \cdot (k-1)}{k^2} - \frac{0.4s(k-1)}{k^3}\right)\psi_{r,i,l} + \frac{\eta_2(\mathcal{E}_5 + \mathcal{E}_6)}{k} > 0. \tag{18}$$

Similar to the former analysis, for simplicity of analysis, we also take that

$$u_{i,r}^{(t)} \approx \tilde{O}\left(\frac{\eta_2}{k}\right)\psi_{r,i,l} + \frac{\eta_2(\mathcal{E}_5 + \mathcal{E}_6)}{k}, \quad \text{for } t \geq 2. \tag{19}$$

### G.2.2 FINETUNING OF $w_r$ AND PROOF OF LEMMA G.1

After the update of $u_{i,r}$, we then finetune $w_r$. We have the gradients:

$$-\nabla_{w_r}L(F;X,y) = (1 - \text{logit}_y(F;X))u_{y,r} \sum_{p\in[P]} \overline{\text{ReLU}}'(\langle w_r, x_p\rangle)x_p$$

$$- \sum_{i\in[k]\backslash y} \text{logit}_i(F;X)u_{i,r} \sum_{p\in[P]} \overline{\text{ReLU}}'(\langle w_r, x_p\rangle)x_p$$

$$= \left[(1 - \text{logit}_y(F;X))u_{y,r} - \sum_{j\in[k]\backslash y} \text{logit}_j(F;X)u_{j,r}\right] \sum_{p\in[P]} \overline{\text{ReLU}}'(\langle w_r, x_p\rangle)x_p$$

**Diagonal correlations.** For $r \in \mathcal{M}_{i,l}^{(0)}$, as we initialize $w_r$ by the pretrained encoder, we have

$$\sum_{p\in[P]} \overline{\text{ReLU}}'(\langle w_r, x_p\rangle)\langle x_p, v_{i,l}\rangle = \mathbb{I}_{\{v_{i,l}\in\mathcal{V}(X)\}} \sum_{p\in\mathcal{P}_{v_{i,l}}(X)} \overline{\text{ReLU}}'(\langle w_r, x_p\rangle)(z_p + \gamma + \sigma_p)$$

$$+ \tilde{O}(\sigma_0^{q-1}) \cdot (\gamma + \sigma_p) \cdot (s+1)$$

$$+ \tilde{O}((\sigma_0\gamma k)^{q-1}) \cdot (\gamma + \sigma_p) \cdot P.$$

Thus,

$$\langle -\nabla_{w_r}L(F;X,y), v_{i,l}\rangle = (V_{r,i,l} + \mathcal{E}_1 + \mathcal{E}_2)\left[(1 - \text{logit}_y(F;X))u_{y,r} - \sum_{j\in[k]\backslash y} \text{logit}_j(F;X)u_{j,r}\right]. \tag{20}$$

At $t = 1$, for every $(X,y) \sim \mathcal{Z}_{\text{down}}$, when $i = y$, put (16) and (18) into (20), we have

$$\langle -\nabla_{w_r}L(F;X,y), v_{i,l}\rangle = \eta_2\left(\frac{O(1)}{k} - \frac{0.4s}{k^2}\right)(V_{r,i,l} + \mathcal{E}_1 + \mathcal{E}_2)(1 - \text{logit}_y(F;X))\psi_{r,i,l}$$

$$+ (V_{r,i,l} + \mathcal{E}_1 + \mathcal{E}_2) \cdot \frac{\eta_2(\mathcal{E}_5 + \mathcal{E}_6)}{k}$$

Similarly, when $y \neq i$,

$$\langle -\nabla_{w_r}L(F;X,y), v_{i,l}\rangle = \eta_2\left(-\frac{O(1)}{k} + \frac{0.4s}{k^2}\right)(V_{r,i,l} + \mathcal{E}_1 + \mathcal{E}_2)\text{logit}_i(F;X)\psi_{r,i,l}$$

$$+ (V_{r,i,l} + \mathcal{E}_1 + \mathcal{E}_2) \cdot \frac{\eta_2(\mathcal{E}_5 + \mathcal{E}_6)}{k}$$

Denote $S_{i,l} = \sum_{p\in\mathcal{P}_{v_{i,l}}(X)} z_p$. We have

$$\mathbb{E}_{(X,y)\sim\mathcal{Z}_{\text{down}}}[\langle -\nabla_{w_r}L(F;X,y), v_{i,l}\rangle] = \frac{1}{k}\eta_2\left(\frac{O(1)}{k} - \frac{0.4s}{k^2}\right)S_{i,l}\frac{k-1}{k}\psi_{r,i,l}$$

$$+ \frac{k-1}{k}\eta_2\left(-\frac{O(1)}{k} + \frac{0.4s}{k^2}\right)S_{i,l}\frac{s}{k^2}\psi_{r,i,l} + \frac{\eta_2(\mathcal{E}_5 + \mathcal{E}_6)}{k}$$

$$= \left(\frac{k-1}{k^2} - \frac{(k-1)s}{k^3}\right)\eta_2\left(\frac{O(1)}{k} - \frac{0.4s}{k^2}\right)S_{i,l}\psi_{r,i,l} + \frac{\eta_2(\mathcal{E}_5 + \mathcal{E}_6)}{k}.$$

Thus, at $t = 1$, we have

$$\langle w_r^{(1)}, v_{i,l} \rangle = \langle w_r^{(0)}, v_{i,l} \rangle + O\left(\frac{\eta_1 \eta_2}{k^2}\right) \psi_{r,i,l} + \frac{\eta_1 \eta_2 (\mathcal{E}_5 + \mathcal{E}_6)}{k} \tag{21}$$

$$\leq \Lambda_{i,l}^{(T)} + O\left(\frac{\eta_1 \eta_2}{k^2}\right) \Lambda_{i,l}^{(T)} + \frac{\eta_1 \eta_2 (\mathcal{E}_5 + \mathcal{E}_6)}{k} \leq \tilde{O}(1),$$

when $\eta_1 \eta_2 \leq \tilde{O}(k^2)$. The lower bound on $\langle w_r^{(1)}, v_{i,l} \rangle$ can be easily obtained by similar methods.

Besides, for $t > 1$, for every $(X, y) \sim \mathcal{Z}_{\text{down}}$, when $i = y$, putting (17) and (19) into (20) and keeping the main term, we have

$$\langle -\nabla_{w_r} L(F; X, y), v_{i,l} \rangle \approx \tilde{O}\left(\frac{\eta_2}{k}\right) (V_{r,i,l} + \mathcal{E}_1 + \mathcal{E}_2)(1 - \text{logit}_y(F; X)) \psi_{r,i,l}$$

$$+ (V_{r,i,l} + \mathcal{E}_1 + \mathcal{E}_2) \cdot \frac{\eta_2 (\mathcal{E}_5 + \mathcal{E}_6)}{k}$$

Similarly, when $y \neq i$,

$$\langle -\nabla_{w_r} L(F; X, y), v_{i,l} \rangle \approx -\tilde{O}\left(\frac{\eta_2}{k}\right) (V_{r,i,l} + \mathcal{E}_1 + \mathcal{E}_2) \text{logit}_i(F; X) \psi_{r,i,l}$$

$$+ (V_{r,i,l} + \mathcal{E}_1 + \mathcal{E}_2) \cdot \frac{\eta_2 (\mathcal{E}_5 + \mathcal{E}_6)}{k}$$

Suppose induction hypothesis C.3 holds at time $t$. Now we have that

$$\mathbb{E}_{(X,y) \sim \mathcal{Z}_{\text{down},m}}[\langle -\nabla_{w_r} L(F; X, y), v_{i,l} \rangle]$$

$$\overset{(a)}{\geq} \tilde{O}\left(\frac{\eta_2}{k}\right) \psi_{r,i,l} \mathbb{E}_{(X,y) \sim \mathcal{Z}_{\text{down},m}}\left[\mathbb{I}_{\{y=i\}}(1 - \text{logit}_y(F; X))\right]$$

$$- 0.4 \mathbb{I}_{\{y \neq i\}} \text{logit}_i(F; X)\Big] + \frac{\eta_2 (\mathcal{E}_5 + \mathcal{E}_6)}{k}.$$

where (a) is because for $y \neq i$, $V_{r,i,l} = 0.4 \mathbb{I}_{\{v_{i,l} \in \mathcal{V}(X)\}} \leq 0.4$, and

$$\mathbb{E}_{(X,y) \sim \mathcal{Z}_{\text{down},s}}[\langle -\nabla_{w_r} L(F; X, y), v_{i,\hat{l}} \rangle]$$

$$\overset{(a)}{\geq} \tilde{O}\left(\frac{\eta_2}{k}\right) \psi_{r,i,\hat{l}} \mathbb{E}_{(X,y) \sim \mathcal{Z}_{\text{down},s}}\left[\mathbb{I}_{\{y=i\}}(1 - \text{logit}_y(F; X))\right]$$

$$- 0.4 \mathbb{I}_{\{y \neq i\}} \text{logit}_i(F; X)\Big] + \frac{\eta_2 (\mathcal{E}_5 + \mathcal{E}_6)}{k}.$$

Thus, using the result $\psi_{r,i,l} \geq \frac{1}{\text{polylog}(k)}$, we obtain

$$\sum_{i \in [k]} \langle w_r^{(t+1)}, v_{i,l} \rangle \geq \sum_{i \in [k]} \langle w_r^{(t)}, v_{i,l} \rangle + \tilde{\Omega}\left(\frac{\eta_1 \eta_2}{k}\right) \mathbb{E}_{(X,y) \sim \mathcal{Z}_{\text{down}}}\left[(1 - \text{logit}_y(F; X))\right] + \eta_1 \eta_2 (\mathcal{E}_5 + \mathcal{E}_6).$$

As the induction hypothesis C.3 still holds in the training process, we have $\Lambda_{i,l}^{(t)} \leq \tilde{O}(1)$. Thus,

$$\sum_{t=1}^{T_{\text{down}}} \mathbb{E}_{(X,y) \sim \mathcal{Z}_{\text{down}}}\left[(1 - \text{logit}_y(F; X))\right] + \tilde{O}\left(k T_{\text{down}}(\mathcal{E}_5 + \mathcal{E}_6)\right) \leq \tilde{O}\left(\frac{k^2}{\eta_1 \eta_2}\right). \tag{22}$$

So, if we assume induction hypothesis C.3 holds for all iteration $< t$, then

$$\langle w_r^{(t)}, v_{i,l} \rangle \leq \langle w_r^{(1)}, v_{i,l} \rangle + \tilde{O}\left(\frac{\eta_1 \eta_2}{k^2}\right) \sum_{t=1}^{t} \mathbb{E}_{(X,y) \sim \mathcal{Z}_{\text{down}}}\left[(1 - \text{logit}_y(F; X))\right]$$

$$+ \frac{t \eta_1 \eta_2 (\mathcal{E}_5 + \mathcal{E}_6)}{k} \leq \tilde{O}(1).$$

**Off-diagonal correlations.** For $r \in \mathcal{M}_{i,l}^{(0)}$, as we initialize $w_r$ by the pretrained encoder, we have

$$\sum_{p \in [P]} \overline{\mathrm{ReLU}}'(\langle w_r, x_p \rangle) \langle x_p, v_{j,l'} \rangle = \hat{V}_{r,i,l}(X)(\gamma + \sigma_p) + \mathbb{I}_{\{v_{j,l'} \in \mathcal{V}(X)\}} \tilde{O}(\sigma_0^{q-1}) + \mathcal{E}_1 + \mathcal{E}_2.$$

Thus,

$$\langle -\nabla_{w_r} L(F; X, y), v_{j,l'} \rangle = (\hat{V}_{r,i,l}(X)(\gamma + \sigma_p) + \mathbb{I}_{\{v_{j,l'} \in \mathcal{V}(X)\}} \tilde{O}(\sigma_0^{q-1}) + \mathcal{E}_1 + \mathcal{E}_2)$$
$$\times \left[ (1 - \mathrm{logit}_y(F; X)) u_{y,r} - \sum_{j \in [k] \backslash y} \mathrm{logit}_j(F; X) u_{j,r} \right]. \quad (23)$$

At $t = 1$, for every $(X, y) \sim \mathcal{Z}_{\mathrm{down}}$, when $i = y$, put (16) and (18) into (23), we have

$$\langle -\nabla_{w_r} L(F; X, y), v_{j,l'} \rangle = ((\gamma + \sigma_p) + \mathbb{I}_{\{v_{j,l'} \in \mathcal{V}(X)\}} \tilde{O}(\sigma_0^{q-1}))$$
$$\times \left( \eta_2 \left( \frac{O(1)}{k} - \frac{0.4s}{k^2} \right) (1 - \mathrm{logit}_y(F; X)) \psi_{r,i,l} + \frac{\eta_2(\mathcal{E}_5 + \mathcal{E}_6)}{k} \right)$$

Similarly, when $y \neq i$ but $y = j$,

$$\langle -\nabla_{w_r} L(F; X, y), v_{j,l} \rangle = \left( \mathbb{I}_{\{v_{i,l} \in \mathcal{V}(X)\}} (\gamma + \sigma_p) + \tilde{O}(\sigma_0^{q-1}) + \mathcal{E}_1 + \mathcal{E}_2 \right)$$
$$\times \left( \eta_2 \left( -\frac{O(1)}{k} + \frac{0.4s}{k^2} \right) \mathrm{logit}_i(F; X) \psi_{r,i,l} + \frac{\eta_2(\mathcal{E}_5 + \mathcal{E}_6)}{k} \right).$$

When $y \neq i$ and $y \neq j$,

$$\langle -\nabla_{w_r} L(F; X, y), v_{j,l} \rangle = \left( \mathbb{I}_{\{v_{i,l} \in \mathcal{V}(X)\}} (\gamma + \sigma_p) + \mathbb{I}_{\{v_{j,l'} \in \mathcal{V}(X)\}} \tilde{O}(\sigma_0^{q-1}) + \mathcal{E}_1 + \mathcal{E}_2 \right)$$
$$\times \left( \eta_2 \left( -\frac{O(1)}{k} + \frac{0.4s}{k^2} \right) \mathrm{logit}_i(F; X) \psi_{r,i,l} + \frac{\eta_2(\mathcal{E}_5 + \mathcal{E}_6)}{k} \right).$$

Therefore,

$$\mathbb{E}_{(X,y) \sim \mathcal{Z}_{\mathrm{down}}} [\langle -\nabla_{w_r} L(F; X, y), v_{j,l'} \rangle]$$
$$= \frac{1}{k}((\gamma + \sigma_p) + \tilde{O}(s\sigma_0^{q-1}/k)) \eta_2 \left( \frac{O(1)}{k} - \frac{0.4s}{k^2} \right) \frac{k-1}{k} \psi_{r,i,l}$$
$$+ \frac{1}{k} \left( \frac{s}{k}(\gamma + \sigma_p) + \tilde{O}(\sigma_0^{q-1}) \right) \eta_2 \left( -\frac{O(1)}{k} + \frac{0.4s}{k^2} \right) \frac{1}{k} \psi_{r,i,l}$$
$$+ \frac{k-2}{k} \left( \frac{s}{k}(\gamma + \sigma_p) + \tilde{O}(s\sigma_0^{q-1}/k) \right) \eta_2 \left( -\frac{O(1)}{k} + \frac{0.4s}{k^2} \right) \frac{1}{k} \psi_{r,i,l}$$
$$+ \frac{\eta_2(\gamma + \sigma_p)(\mathcal{E}_5 + \mathcal{E}_6)}{k}$$
$$= -\frac{s}{k}((\gamma + \sigma_p) + \tilde{O}(\sigma_0^{q-1})) \eta_2 \left( \frac{O(1)}{k} - \frac{0.4s}{k^2} \right) \psi_{r,i,l} + \frac{\eta_2(\gamma + \sigma_p)(\mathcal{E}_5 + \mathcal{E}_6)}{k}.$$

Thus, at $t = 1$, we have

$$\langle w_r^{(1)}, v_{j,l'} \rangle \leq \langle w_r^{(0)}, v_{j,l'} \rangle + \tilde{O} \left( \frac{\eta_1 \eta_2}{k^2}(\gamma + \sigma_p) \right) \psi_{r,i,l} + \frac{\eta_1 \eta_2(\gamma + \sigma_p)(\mathcal{E}_5 + \mathcal{E}_6)}{k} \leq \tilde{O}(\sigma_0),$$

when $\eta_1 \eta_2 \leq \tilde{O}(k^2)$. Suppose induction hypothesis C.3 holds for all iterations $< t$. We have

$$\langle w_r^{(t)}, v_{j,l'} \rangle \leq \langle w_r^{(1)}, v_{j,l'} \rangle + \tilde{O} \left( \frac{\eta_1 \eta_2}{k^2}(\gamma + \sigma_p) \right) \sum_{t=1}^{T_{\mathrm{down}}} \mathbb{E}_{(X,y) \sim \mathcal{Z}_{\mathrm{down}}} \left[ (1 - \mathrm{logit}_y(F; X)) \right]$$
$$+ \frac{T_{\mathrm{down}} \eta_1 \eta_2(\gamma + \sigma_p)(\mathcal{E}_5 + \mathcal{E}_6)}{k} \leq \tilde{O}(\sigma_0)$$

**Kernels outside** $\cup_{i\in[k],l\in[2]}\mathcal{M}_{i,l}^{(0)}$**.** For $r \notin \mathcal{M}_{i,l}^{(0)}$, as we initialize $w_r$ by the pretrained encoder, we have

$$\sum_{p\in[P]} \overline{\mathrm{ReLU}}'(\langle w_r, x_p\rangle)\langle x_p, v_{i,l}\rangle = \tilde{O}(\sigma_0^{q-1}) + \mathcal{E}_1 + \mathcal{E}_2,$$

which is very small and there is nearly no increase on $\langle w_r, v_{i,l}\rangle$. Thus, when induction hypothesis C.3 holds for all iterations $< t$, for $r \notin \mathcal{M}_{i,l}^{(0)}$, we have $\langle w_r^{(t)}, v_{i,l}\rangle \leq \tilde{O}(\sigma_0)$.

**Noise correlations.** For every $r \in [km]$, for every $(X^*, y^*) \in \mathcal{Z}$ and every $p^* \in [P]$, we have that

$$\mathbb{E}_{(X,y)\sim\mathcal{Z}}\left[\mathbb{I}_{X=X^*}\langle-\nabla_{w_r}L(F;X,y),\xi_{p^*}\rangle\right] = \tilde{\Theta}\left(\frac{1}{N_2}\right)\mathbb{E}_{(X,y)\sim\mathcal{Z}}\left[(\overline{\mathrm{ReLU}}'(\langle w_r, x_{p^*}\rangle) \pm o(1/\sqrt{d}))\right.$$
$$\left.\times\left[(1-\mathrm{logit}_y(F;X^*))u_{y,r} - \sum_{j\in[k]\backslash y}\mathrm{logit}_j(F;X^*)u_{j,r}\right]\right],$$

and

$$\mathbb{E}_{(X,y)\sim\mathcal{Z}}\left[\mathbb{I}_{X\neq X^*}\langle-\nabla_{w_r}L(F;X,y),\xi_{p^*}\rangle\right] = \pm o(1/\sqrt{d}).$$

For every $v_{i,l} \in \mathcal{V}$, for every $r \in \mathcal{M}_{i,l}^{(0)}$, for every $p^* \in \mathcal{P}_{v_{i,l}}(X^*)$, when $i = y$, we have

$$\mathbb{E}_{(X,y)\sim\mathcal{Z}}\left[\mathbb{I}_{\{i=y\}}\langle-\nabla_{w_r}L(F;X,y),\xi_{p^*}\rangle\right]$$
$$= \tilde{\Theta}\left(\frac{\eta_2}{N_2}\right)\overline{\mathrm{ReLU}}'(\langle w_r, x_{p^*}\rangle)\left(\frac{O(1)}{k} - \frac{0.4s}{k^2}\right)(1-\mathrm{logit}_y(F;X^*))\psi_{r,i,l} \pm o(1/\sqrt{d})$$
$$\stackrel{(a)}{=} \tilde{\Theta}\left(\frac{\eta_2}{N_2}\right)\left(\frac{O(1)}{k} - \frac{0.4s}{k^2}\right)\psi_{r,i,l} + \frac{\eta_2(\mathcal{E}_5+\mathcal{E}_6)}{N_2 k} \pm o(\eta_2/\sqrt{d}),$$

where $(a)$ is because $1 - \mathrm{logit}_y(F;X^*) = \frac{k-1}{k}$ at $t = 0$. When $i \neq y$, we have

$$\mathbb{E}_{(X,y)\sim\mathcal{Z}}\left[\mathbb{I}_{\{i\neq y\}}\langle-\nabla_{w_r}L(F;X,y),\xi_{p^*}\rangle\right]$$
$$= \tilde{\Theta}\left(\frac{1}{(k-1)N_2}\right)\eta_2\left(-\frac{O(1)}{k} + \frac{0.4s}{k^2}\right)\psi_{r,i,l} + \frac{\eta_2(\mathcal{E}_5+\mathcal{E}_6)}{N_2 k(k-1)} \pm o(\eta_2/\sqrt{d}),$$

Thus, we have

$$\mathbb{E}_{(X,y)\sim\mathcal{Z}}\left[\langle-\nabla_{w_r}L(F;X,y),\xi_{p^*}\rangle\right] = \frac{\eta_2(\mathcal{E}_5+\mathcal{E}_6)}{N_2 k^2} \pm o(\eta_2/\sqrt{d}),$$

and

$$\langle w_r^{(1)}, \xi_p\rangle = \langle w_r^{(0)}, \xi_p\rangle + \frac{\eta_1\eta_2(\mathcal{E}_5+\mathcal{E}_6)}{N_2 k^2} \pm o(\eta_1\eta_2/\sqrt{d}) \leq \tilde{o}(\sigma_0).$$

Thus, when induction hypothesis C.3 holds for all iterations $< t$, we have

$$\langle w_r^{(t)}, \xi_p\rangle \leq \langle w_r^{(1)}, \xi_p\rangle + \frac{T_{\mathrm{down}}\eta_1\eta_2(\mathcal{E}_5+\mathcal{E}_6)}{N_2 k^2} \pm o(T_{\mathrm{down}}\eta_1\eta_2/\sqrt{d}) \leq \tilde{o}(\sigma_0).$$

Similarly, following the similar step as in the proof of Lemma F.6, we can also prove other claims about the noise correlations in the downstream tasks. We skip the similar steps here.

Combining all above results, we can prove the Lemma G.1.

### G.2.3 TRAINING LOSS AND PROOF OF THEOREM G.2 (A)

We set $\eta_2$ to be $O(k)$. The reason why we set the step size to $O(k)$ is in the first step, the weights of negative parts $(< 0)$ and positive parts $(> 0)$ is well separated. Thus, by setting a suitable step length $\eta_2 = O(k)$, we can obtain a small loss in the first update of (15). We will show that the training loss is small in the following.

After one-step training, at $t = 1$, for $(X, y) \in \mathcal{Z}_{\text{down},m}$, we have

$$
F_j(X) - F_y(X)
$$

$$
= \sum_{l=1}^{2} \sum_{r \in \mathcal{M}_{j,l}^{(0)}} (u_{j,r} - u_{y,r})\Big(\psi_{r,j,l} \cdot Z_{j,l}(X) + \mathcal{E}_5 + \mathcal{E}_6\Big) + \sum_{l=1}^{2} \sum_{r \in \mathcal{M}_{y,l}^{(0)}} (u_{j,r} - u_{y,r})\Big(\psi_{r,y,l} \cdot Z_{y,l}(X) + \mathcal{E}_5 + \mathcal{E}_6\Big)
$$

$$
+ \sum_{i \in [k] \backslash \{j,y\}, l \in [2]} \sum_{r \in \mathcal{M}_{i,l}^{(0)}} (u_{j,r} - u_{y,r})\Big(\psi_{r,v'} \cdot Z_{v'}(X) + \mathcal{E}_5 + \mathcal{E}_6\Big)
$$

$$
\overset{(a)}{=} \sum_{l=1}^{2} \sum_{r \in \mathcal{M}_{j,l}^{(0)}} (u_{j,r} - u_{y,r})\Big(\psi_{r,j,l} \cdot Z_{j,l}(X) + \mathcal{E}_5 + \mathcal{E}_6\Big) + \sum_{l=1}^{2} \sum_{r \in \mathcal{M}_{y,l}^{(0)}} (u_{j,r} - u_{y,r})\Big(\psi_{r,y,l} \cdot Z_{y,l}(X) + \mathcal{E}_5 + \mathcal{E}_6\Big)
$$

$$
+ \eta_2 m_0 (\mathcal{E}_5 + \mathcal{E}_6)
$$

$$
= \eta_2 \sum_{l=1}^{2} \sum_{r \in \mathcal{M}_{j,l}^{(0)}} \left( \frac{O(1) \cdot (k-1)}{k^2} - \frac{0.4s(k-1)}{k^3} - \frac{0.4s}{k^3} + \frac{O(1)}{k^2} \right) \Big(\psi_{r,j,l}^2 \cdot Z_{j,l}(X)\Big)
$$

$$
+ \eta_2 \sum_{l=1}^{2} \sum_{r \in \mathcal{M}_{y,l}^{(0)}} \left( \frac{0.4s}{k^3} - \frac{O(1)}{k^2} - \frac{O(1) \cdot (k-1)}{k^2} + \frac{0.4s(k-1)}{k^3} \right) \Big(\psi_{r,y,l}^2 \cdot Z_{y,l}(X)\Big) + \eta_2 m_0 (\mathcal{E}_5 + \mathcal{E}_6)
$$

$$
= \eta_2 \left( \frac{O(1)}{k} - \frac{0.4s}{k^2} \right) \left( \sum_{l=1}^{2} \sum_{r \in \mathcal{M}_{j,l}^{(0)}} \psi_{r,j,l}^2 \cdot Z_{j,l}(X) - \sum_{l=1}^{2} \sum_{r \in \mathcal{M}_{y,l}^{(0)}} \psi_{r,y,l}^2 \cdot Z_{y,l}(X) \right) + \eta_2 m_0 (\mathcal{E}_5 + \mathcal{E}_6)
$$

$$
= \eta_2 \left( \frac{O(1)}{k} - \frac{0.4s}{k^2} \right) \left( 0.4 \sum_{l=1}^{2} \sum_{r \in \mathcal{M}_{j,l}^{(0)}} \mathbb{I}_{\{v_{j,l} \in \mathcal{V}(X)\}} \psi_{r,j,l}^2 - \sum_{l=1}^{2} \sum_{r \in \mathcal{M}_{y,l}^{(0)}} \psi_{r,y,l}^2 \right) + \eta_2 m_0 (\mathcal{E}_5 + \mathcal{E}_6),
$$

where (a) is because the third term is nearly zero. We could show the similar result for single-view data. At $t = 1$, for $(X, y) \in \mathcal{Z}_{\text{down},s}$, we have

$$
F_j(X) - F_y(X)
$$

$$
= \eta_2 \left( \frac{O(1)}{k} - \frac{0.4s}{k^2} \right) \left( 0.4 \sum_{l=1}^{2} \sum_{r \in \mathcal{M}_{j,l}^{(0)}} \mathbb{I}_{\{v_{j,l} \in \mathcal{V}(X)\}} \psi_{r,j,l}^2 - \sum_{r \in \mathcal{M}_{y,\hat{l}}^{(0)}} \psi_{r,y,\hat{l}}^2 \right.
$$

$$
\left. - \rho \sum_{r \in \mathcal{M}_{y,3-\hat{l}}^{(0)}} \psi_{r,y,3-\hat{l}}^2 \right) + \eta_2 m_0 (\mathcal{E}_5 + \mathcal{E}_6).
$$

Thus, at $t = 1$, we have

$$
\mathbb{E}_{(X,y) \sim \mathcal{Z}_{\text{down},m}} \left[ \text{logit}_y(F; X) \right] \approx \left[ \left( \frac{2s}{k} - \frac{2s^2}{k^2} \right) \frac{1}{1 + \sum_{i \in [k] \backslash y} e^{0.4\Psi_{i,l} - \Psi_y}} + \frac{s^2}{k^2} \frac{1}{1 + \sum_{i \in [k] \backslash y} e^{0.4\Psi_i - \Psi_y}} \right.
$$

$$
\left. + \left( 1 - \frac{s}{k} \right)^2 \frac{1}{1 + \sum_{i \in [k] \backslash y} e^{0.4s/k - \Psi_y}} \right]
$$

$$
\geq 1 - \tilde{O}\left( \frac{1}{k} \right), \tag{24}
$$

where the last inequality using the result that $\psi_{r,i,l} \geq \frac{1}{\text{polylog}(k)}$ and $\psi_{r,i,l} \leq \tilde{O}(1)$ from Lemma F.1 at initialization, $|\mathcal{M}_{i,l}^0| \leq O(\log^5 k)$ from Lemma C.1. We could obtain the similar results for single-view data.

Finally, if we set $T_{\text{down}} \geq \frac{\text{poly}(k)}{\eta_1 \eta_2}$, according to (22), it is easy to verify that

$$\frac{1}{T_{\text{down}}} \sum_{t=1}^{T_{\text{down}}} \mathbb{E}_{(X,y) \sim \mathcal{Z}_{\text{down}}} \left[ -\log \frac{e^{F_y(X)}}{\sum_{j \in [k]} e^{F_j(X)}} \right] \leq \frac{1}{T_{\text{down}}} \sum_{t_1=1}^{T_{\text{down}}} \mathbb{E}_{(X,y) \sim \mathcal{Z}_{\text{down}}} \left[ 1 - \text{logit}_y(F; X) \right]$$

$$\leq \frac{1}{\text{poly}(k)}.$$

This implies that the training loss is small and so we prove Theorem G.2 (a).

### G.2.4 PROOF OF THEOREM G.2 (B)

In this subsection, we prove Theorem G.2 (b). For $(X, y) \sim \mathcal{D}_m$, due to our definition of data structure in Definition 1, with probability at least $1 - e^{-\Omega(\log^2 k)}$, it satisfies that for every $j \in [k] \setminus y$,

$$F_j(X) - F_y(X) \approx O(1) \cdot \left( 0.4 \sum_{l=1}^{2} \mathbb{I}_{\{v_{j,l} \in \mathcal{V}(X)\}} \Psi_{j,l} - \sum_{l=1}^{2} \Psi_{y,l} \right). \tag{25}$$

and for $(X, y) \sim \mathcal{D}_s$,

$$F_j(X) - F_y(X) \approx O(1) \cdot \left( 0.4 \sum_{l=1}^{2} \mathbb{I}_{\{v_{j,l} \in \mathcal{V}(X)\}} \Psi_{j,l} - \rho \Psi_{y,3-\hat{\imath}} - \Psi_{y,\hat{\imath}} \right). \tag{26}$$

To prove Theorem G.2 (b), we need a lemma:

**Lemma G.4.** *For every* $(X, y) \in \mathcal{Z}_{\text{down}}$,

$$1 - \text{logit}_y(F; X) \leq \tilde{O}\left( \frac{k^4}{s^2} \right) \cdot \mathbb{E}_{(X,y) \sim \mathcal{Z}_{\text{down}}}[1 - \text{logit}_y(F; X)].$$

*(The same also hold with probability* $\geq 1 - e^{-\Omega(\log^2 k)}$ *for every* $(X, y) \sim \mathcal{D}$ *on the left hand side.)*

*Furthermore, if* $\mathbb{E}_{(X,y) \sim \mathcal{Z}_{\text{down}}}[1 - \text{logit}_y(F; X)] \leq \frac{1}{k^5}$ *is sufficiently small, we have for every* $j \in [k] \setminus y$,

$$F_j(X) - F_y(X) \leq -\tilde{O}(1).$$

*Proof of Lemma G.4.* The proof of Lemma G.4 for multi-view data has been shown in (Allen-Zhu & Li, 2020, Claim C.16). Now we prove this lemma also holds for single-view data.

For a data point $(X, y) \in \mathcal{Z}_{\text{down},s}$, let us denote by $\mathcal{H}(X)$ be the set of all $i \in [k] \setminus \{y\}$ such that

$$\sum_{l \in [2]} \sum_{p \in \mathcal{P}_{v_{i,l}}(X)} z_p \geq 0.8 - \frac{1}{100 \log k}, \quad \sum_{l \in [2]} \sum_{p \in \mathcal{P}_{v_{y,l}}(X)} z_p \leq 1 + \rho + \frac{1}{100 \log k}.$$

Now suppose $1 - \text{logit}_y(F; X) = \zeta(X)$, then using $\min\{1, \beta\} \leq 2\left(1 - \frac{1}{1+\beta}\right)$, we have

$$\min\left\{1, \sum_{i \in [k] \setminus \{y\}} e^{F_i(X) - F_y(X)}\right\} \leq 2\zeta(X).$$

By (26) and our definition of $\mathcal{H}(X)$, this implies that

$$\min\left\{1, \sum_{i \in \mathcal{H}(X)} e^{O(1) \cdot (0.4 \Psi_i - \rho \Psi_{y,3-\hat{\imath}} - \Psi_{y,\hat{\imath}})}\right\} \leq 4\zeta(X)$$

Now we define $\phi = \mathbb{E}_{(X,y) \sim \mathcal{Z}_{\text{down},s}}[1 - \text{logit}_y(F; X)]$, then

$$\mathbb{E}_{(X,y) \sim \mathcal{Z}_{\text{down},s}} \left[ \min\left\{1, \sum_{i \in \mathcal{H}(X)} e^{O(1) \cdot (0.4 \Psi_i - \rho \Psi_{y,3-\hat{\imath}} - \Psi_{y,\hat{\imath}})}\right\} \right] \leq 4\phi$$

$$\Longrightarrow \mathbb{E}_{(X,y) \sim \mathcal{Z}_{\text{down},s}} \left[ \sum_{i \in \mathcal{H}(X)} \min\left\{\frac{1}{k}, e^{O(1) \cdot (0.4 \Psi_i - \rho \Psi_{y,3-\hat{\imath}} - \Psi_{y,\hat{\imath}})}\right\} \right] \leq 4\phi.$$

It equals to

$$\sum_{j\in[k]}\sum_{i\in[k]}\mathbb{I}_{\{i\neq j\}}\mathbb{E}_{(X,y)\sim\mathcal{Z}_{\text{down},s}}[\mathbb{I}_{\{j=y\}}\mathbb{I}_{\{i\in\mathcal{H}(X)\}}]\min\left\{\frac{1}{k},e^{O(1)\cdot(0.4\Psi_i-\rho\Psi_{y,3-\hat{i}}-\Psi_{y,\hat{i}})}\right\}\leq 4\phi.$$

Note that for every $i\neq j\in[k]$, the probability of choosing a single-view sample $(X,y)$ from $\mathcal{Z}_{\text{down},s}$ with $y=j$ and $i\in\mathcal{H}(X)$ is at least $\Omega\left(\frac{1}{k}\cdot\frac{s^2}{k^2}\right)$. This implies

$$\sum_{j\in[k]}\sum_{i\in[k]\setminus j}\min\left\{\frac{1}{k},e^{O(1)\cdot(0.4\Psi_i-\rho\Psi_{y,3-\hat{i}}-\Psi_{y,\hat{i}})}\right\}\leq\tilde{O}\left(\frac{k^3}{s^2}\phi\right).$$

Finally, using $1-\frac{1}{1+\beta}\leq\min\{1,\beta\}$, for every $(X,y)\sim\mathcal{Z}_{\text{down},s}$, we have

$$1-\text{logit}_y(F;X)\leq\min\left\{1,\sum_{i\in[k]\setminus y}2e^{O(1)\cdot(0.4\Psi_i-\rho\Psi_{y,3-\hat{i}}-\Psi_{y,\hat{i}})}\right\}$$

$$\leq k\cdot\sum_{i\in[k]\setminus y}\min\left\{\frac{1}{k},e^{O(1)\cdot(0.4\Psi_i-\rho\Psi_{y,3-\hat{i}}-\Psi_{y,\hat{i}})}\right\}\leq\tilde{O}\left(\frac{k^4}{s^2}\phi\right).$$

This implies that when $\mathbb{E}_{(X,y)\sim\mathcal{Z}_{\text{down},s}}\left[(1-\text{logit}_y(F;X))\right]\leq\frac{1}{k^5}$, we have

$$0.4\Psi_i-\rho\Psi_{y,3-\hat{i}}-\Psi_{y,\hat{i}}\leq-\tilde{O}(1).$$

$\square$

As we have proved in Section G.2.3 that

$$\mathbb{E}_{(X,y)\sim\mathcal{Z}_{\text{down}}}\left[(1-\text{logit}_y(F;X))\right]\leq\frac{1}{\text{poly}(k)},$$

We could set $T_{\text{down}}\geq\tilde{O}\left(\frac{k^7}{\eta_1\eta_2}\right)$ and then based on Lemma G.4, we have

$$\Pr_{(X,y)\in\mathcal{D}}\left[F_y(X)\geq\max_{j\neq y}F_j(X)+\tilde{O}(1)\right]\geq 1-e^{-\Omega(\log^2 k)}.$$

# H EXTENSIONS ON OTHER MRP METHODS

We have prove that Theorem C.4 holds in the above sections, which means that under the Teacher-Student Framework, the pretraining phase can capture all features. In this section, we extend our proof methods to other popular mask-reconstruction pretraining methods. Here we mainly consider the masked autoencoder (MAE) structure He et al. (2021). For simplicity of analysis, we set the weights of the decoder as the copy of encoder weights and add a linear layer with $b_i=c(\theta),i\in[P]$ to finally obtain the recovered patches. The explicit framework is shown in Fig. 6. Denote the position encoding of patch $p$ as $\mathbf{e}_p\in\mathbb{R}^P$, where at position $p$ the element equal to 1, otherwise the element equal to 0. Recall that $\epsilon X=(\epsilon_1 x_1,\epsilon_2 x_2,\ldots,\epsilon_P x_P)$. Under this framework, the loss function is

$$L(H;X,\epsilon)=\frac{1}{2}\sum_{p\in[P]}\left\|x_p-c(\theta)\sum_{r\in[km]}w_r\overline{\text{ReLU}}(\langle w_r,\mathbf{e}_p^T\epsilon X\rangle)\right\|_2^2$$

$$=\frac{1}{2}\sum_{p\in[P]}\left\|x_p-c(\theta)\sum_{r\in[km]}w_r\overline{\text{ReLU}}(\langle w_r,\epsilon_p x_p\rangle)\right\|_2^2$$

and

$$L(H;X)=\mathbb{E}_\epsilon[L(H;X,\epsilon)]=\frac{1}{2}\sum_{p\in[P]}\left\|x_p-\sum_{r\in[km]}w_r\overline{\text{ReLU}}(\langle w_r,x_p\rangle)\right\|_2^2$$

$$+\frac{1}{2}\left(\frac{1-\theta}{\theta}\right)\sum_{p\in[P]}\left\|\sum_{r\in[km]}w_r\overline{\text{ReLU}}(\langle w_r,x_p\rangle)\right\|_2^2.$$

Denote

$$A_{r,p}(X)=\overline{\text{ReLU}}(\langle w_r,x_p\rangle)+\overline{\text{ReLU}}'(\langle w_r,x_p\rangle)[\langle w_r,x_p\rangle]^+.$$

We have that

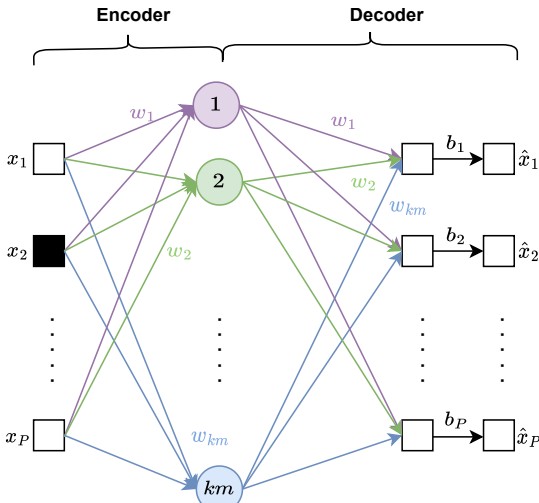

Figure 6: Masked Autoencoder

**Fact 2.2.** *Given the data point* $(X, y) \in \mathcal{D}$, *for every* $w_r, r \in [km]$,

$$-\nabla_{w_r} L(X) = \sum_{p \in [P]} A_{r,p} \left( x_p - \frac{1}{\theta} \sum_{r' \in [km]} w_{r'} \overline{\text{ReLU}}(\langle w_{r'}, x_p \rangle) \right).$$

To prove that under MAE framework, the pretraining can also capture all features, we have the same induction hypothesis as Induction Hypothesis C.3 but now the parameter assumption is a little different. Our new assumptions are shown as follows:

**Assumption H.1** (Parameter Assumption: MAE framework). *The parameters introduced in the paper need to satisfy the following conditions:*

- $\varrho$ *is the threshold for the smoothed ReLU activation. We assume* $\varrho = \frac{1}{\text{polylog}(k)}$.

- $q \geq 4$ *and* $\sigma_0^{q-2} \leq \frac{1}{k}$.

- $\gamma$ *controls feature noise.* $\gamma \leq \tilde{O}\left(\frac{\sigma_0}{k}\right)$.

- $s$ *controls feature sparsity.* $s = \Theta(\text{polylog}(k))$.

- $N \geq \tilde{\omega}\left(\frac{k}{\sigma_0^{q-1}}\right), \sqrt{d} \geq \tilde{\omega}(k/\sigma_0^{q-1})$, *and* $P \leq \sigma_0^{-q+1/2}$.

- $\text{polylog}(k) \leq m \leq \sqrt{k}$.

- $\eta \geq \frac{1}{k^{q(q-2)}}$ *and* $\eta \leq \frac{1}{\text{poly}(k)}$.

- $c(\theta) = \frac{1}{\theta}$.

Now we have the following result on the feature learning process of MAE.

**Theorem H.2** (Feature learning process of MAE). *Suppose Assumption H.1 holds. By running the gradient descent step based on gradient Fact. 2.2 with learning rate* $\eta \leq \frac{1}{\text{poly}(k)}$, *after* $T = \frac{\text{poly}(k)}{\eta}$ *iterations, for sufficiently large* $k > 0$, *Induction Hypothesis C.3 holds for all iterations* $t = 0, 1, \ldots, T$ *with high probability.*

See its proof in Appendix H.2.5. Similarly, we also have the result about the performance on downstream classification tasks shown as follows.

**Theorem H.3** (Performance on downstream classification tasks under MAE pretraining). *For* $N_2 \geq k$ *many samples, setting the learning rate* $\eta_2 = \Theta(k)$ *and* $\eta_1 \leq \tilde{\Theta}(k)$, *after* $T_{\text{down}} \geq \frac{\text{poly}(k)}{\eta_1 \eta_2}$ *many iterations, with high probability, we have*

(a) *(training loss is small) for every $(X, y) \in \mathcal{Z}_{\text{down}}$, i.e.,*

$$L_{\text{down}}(F) = \mathbb{E}_{(X,y) \sim \mathcal{Z}_{\text{down}}}[L_{\text{down}}(F; X, y)] \leq \frac{1}{\text{poly}(k)}.$$

(b) *(test performance is good) for new data point $(X, y) \sim \mathcal{D}$, the test performance is*

$$\Pr_{(X,y) \in \mathcal{D}} \left[ F_y(X) \geq \max_{j \neq y} F_j(X) + \tilde{O}(1) \right] \geq 1 - e^{-\Omega(\log^2 k)}.$$

See its proof in Appendix H.2.6. Theorem H.2 guarantees that under MAE pretraining, the pretrained convolution kernels can capture all discriminative features in the data and each convolution kernel only grab at most one discriminative feature. Such a result accords with the result of MRP in Theorem 1 of the manuscript. Please refer to more detailed discussion and analysis of Theorem 1 in manuscript. *We also note that the assumptions under MAE framework are more strict than the assumptions of Teacher-Student framework. In the assumptions of MAE, we need $q \geq 4$, which means we need to let the low-magnitude feature noises to be compressed much much smaller in order that we can separate the true feature from feature noises.* Then Theorem H.3 shows that as we have captured all features in the MAE pretraining phase, we can also obtain very high accuracy with high probability in the downstream classification tasks. Therefore, compared with supervised learning, MAE also shows better performance on classification downstream task. This result shows the generality of our analysis framework.

To prove Theorem H.2 and Theorem H.3, we mainly follow the similar framework used to prove Theorems 1 and Theorem 2 in the manuscript. To begin with, we first prove some auxiliary theories based on which one can easily prove the desired results.

## H.1    SOME RESULTS FROM INDUCTION HYPOTHESIS C.3 UNDER MAE

We first introduce some claims about the terms in the gradients.

**Claim H.4.** *Suppose Assumption H.1 and Induction Hypothesis C.3 holds at iterations $t$. Then for every $r \in \mathcal{M}_{i,l}^{(0)}$, we have*

- *if $p \in \mathcal{P}_{v_{i,l}}(X)$,*

$$A_{r,p}(X) = \mathbb{I}_{v_{i,l} \in \mathcal{V}(X)} \overline{\text{ReLU}}(\langle w_r, x_p \rangle) + \mathbb{I}_{v_{i,l} \in \mathcal{V}(X)} \overline{\text{ReLU}}'(\langle w_r, x_p \rangle)[\langle w_r, x_p \rangle]^+.$$

- *if $p \in \mathcal{P}(X) \setminus \mathcal{P}_{v_{i,l}}(X)$,*

$$A_{r,p}(X) \approx \tilde{O}(\sigma_0^q).$$

- *if $p \in [P] \setminus \mathcal{P}(X)$,*

$$A_{r,p}(X) \approx \tilde{O}((\sigma_0 \gamma k)^q).$$

We also denote

$$\Delta_p(X) = x_p - \frac{1}{\theta} \sum_{r' \in [km]} w_{r'} \overline{\text{ReLU}}(\langle w_{r'}, x_p \rangle).$$

**Claim H.5.** *Suppose Assumption H.1 and Induction Hypothesis C.3 holds at iterations $t$,*

- *When $p \in \mathcal{P}_{v_{i,l}}(X)$, we have*

$$\langle \Delta_p(X), v_{i,l} \rangle = z_p \mathbb{I}_{v_{i,l} \in \mathcal{V}(X)} - \frac{1}{\theta} \sum_{r' \in \mathcal{M}_{i,l}^{(0)}} \langle w_{r'}, v_{i,l} \rangle \mathbb{I}_{v_{i,l} \in \mathcal{V}(X)} \overline{\text{ReLU}}(\langle w_{r'}, x_p \rangle) - \sum_{r' \notin \mathcal{M}_{i,l}^{(0)}} \tilde{O}(\sigma_0) \tilde{O}(\sigma_0^q)$$

$$= z_p \mathbb{I}_{v_{i,l} \in \mathcal{V}(X)} - \frac{1}{\theta} \sum_{r' \in \mathcal{M}_{i,l}^{(0)}} \langle w_{r'}, v_{i,l} \rangle \mathbb{I}_{v_{i,l} \in \mathcal{V}(X)} \overline{\text{ReLU}}(\langle w_{r'}, x_p \rangle) \pm \tilde{O}(\sigma_0^{q-\frac{1}{2}}).$$

- *When $p \notin \mathcal{P}_{v_{i,l}}(X)$ but $p \in \mathcal{P}_{v_{j,l'}}(X)$ for $v_{j,l'} \neq v_{i,l}$, we have*

$$\langle \Delta_p(X), v_{i,l} \rangle = \gamma \pm \tilde{O}(\sigma_0^{q-\frac{1}{2}}) \pm \sum_{r' \in \mathcal{M}_{i,l}^{(0)}} \langle w_{r'}, v_{i,l} \rangle \tilde{O}(\sigma_0^q) \pm \sum_{r' \in \mathcal{M}_{j,l'}^{(0)}} \mathbb{I}_{v_{j,l'} \in \mathcal{V}(X)} \tilde{O}(\sigma_0) \overline{\mathrm{ReLU}}(\langle w_{r'}, x_p \rangle).$$

- *When $p \in [P] \setminus \mathcal{P}(X)$, we have*

$$\langle \Delta_p(X), v_{i,l} \rangle = \gamma \pm \tilde{O}(\sigma_0^{q+1}(\gamma k)^q) \pm \sum_{r' \in \mathcal{M}_{i,l}^{(0)}} \langle w_{r'}, v_{i,l} \rangle \tilde{O}((\sigma_0 \gamma k)^q).$$

Now we have some claims for the gradients. The proof is just based on the result from Claim H.4 and Claim H.5.

**Claim H.6.** *Suppose Assumption H.1 and Induction Hypothesis C.3 holds at iterations $t$. Then for every $v_{i,l} \in \mathcal{V}$, for every $r \in \mathcal{M}_{i,l}^{(0)}$, for every $(X, y) \in \mathcal{Z}$, we have*

*(a)*

$$-\langle \nabla_{w_r} L(X), v_{i,l} \rangle$$

$$= \sum_{p \in \mathcal{P}_{v_{i,l}}(X)} A_{r,p} \left( z_p \mathbb{I}_{v_{i,l} \in \mathcal{V}(X)} - \frac{1}{\theta} \sum_{r' \in \mathcal{M}_{i,l}^{(0)}} \langle w_{r'}, v_{i,l} \rangle \mathbb{I}_{v_{i,l} \in \mathcal{V}(X)} \overline{\mathrm{ReLU}}(\langle w_{r'}, x_p \rangle) \right)$$

$$\pm \gamma \tilde{O}(\sigma_0^q) \pm \tilde{O}(\sigma_0^{2q-1/2}) \pm \tilde{O}((\sigma_0^{2q+1})(\gamma k)^{2q}) \cdot P$$

*(b) for $v_{j,l'} \neq v_{i,l}$,*

$$-\langle \nabla_{w_r} L(X), v_{j,l'} \rangle$$

$$= \pm \sum_{p \in \mathcal{P}_{v_{i,l}}(X)} A_{r,p}(\gamma + \tilde{O}(\sigma_0^{q-\frac{1}{2}})) \pm \gamma \tilde{O}(\sigma_0^q) \pm \tilde{O}(\sigma_0^{2q-1/2}) \pm \tilde{O}((\sigma_0^{2q+1})(\gamma k)^{2q}) \cdot P$$

$$+ \sum_{p \in \mathcal{P}_{v_{j,l'}}(X)} \tilde{O}(\sigma_0^q) \left( z_p \mathbb{I}_{v_{j,l'} \in \mathcal{V}(X)} - \frac{1}{\theta} \sum_{r' \in \mathcal{M}_{j,l'}^{(0)}} \langle w_{r'}, v_{j,l'} \rangle \mathbb{I}_{v_{j,l'} \in \mathcal{V}(X)} \overline{\mathrm{ReLU}}(\langle w_{r'}, x_p \rangle) \right).$$

**Intuitions on Claim H.6.** From Claim H.6, we can find that the positive-correlation gradient $-\langle \nabla_{w_r} L(X), v_{i,l} \rangle$ has a non-small term that drive the correlation between $w_r$ and $v_{i,l}$ when $r \in \mathcal{M}_{i,l}^{(0)}$ to increase during the training courses. How the correlation increase will be shown in Claim H.7 in the following. On the other hand, the negative correlations will keep small as the negative-correlation gradients $-\langle \nabla_{w_r} L(X), v_{j,l'} \rangle$ always have small terms. These intuitions are same as the intuitions under Teacher-Student Framework and thus we could also prove that MAE pretraining could also capture all features.

Now we have the following claim shows about at which iteration $\Lambda_{i,l}^{(t)}$ will be greater than $\varrho$.

**Claim H.7.** *Suppose Assumption C.2 holds and induction hypothesis C.3 holds at iteration $t$. For every $v_{i,l}$, suppose $\Lambda_{i,l}^{(t)} \leq \varrho$. Then we have*

$$\Lambda_{i,l}^{(t+1)} \approx \Lambda_{i,l}^{(t)} + \tilde{\Theta}\left(\frac{\eta}{k}\right) \overline{\mathrm{ReLU}}(\langle w_r, v_{i,l} \rangle).$$

*Proof of Claim H.7.* Recall that $\Lambda_{i,l}^{(t)} := \max_{r \in [km]} [\langle w_r^{(t)}, v_{i,l} \rangle]^+$. We choose any $r \in [km]$ that makes $\langle w_r^{(t)}, v_{i,l} \rangle \geq \tilde{\Omega}(\sigma_0)$. Now we show the updates. We know that

$$\langle w_r^{(t+1)}, v_{i,l} \rangle = \langle w_r^{(t)}, v_{i,l} \rangle + \eta \mathbb{E}_{(X,y)\sim\mathcal{Z}} \left[ \langle -\nabla_{w_r} L(X), v_{i,l} \rangle \right]$$

Using Claim H.6 and following the similar method in the proof of Claim E.4, we have

$$-\langle \nabla_{w_r} L(X), v_{i,l} \rangle = \sum_{p \in \mathcal{P}_{v_{i,l}}(X)} A_{r,p} \left( z_p \mathbb{I}_{v_{i,l} \in \mathcal{V}(X)} - \frac{1}{\theta} \sum_{r' \in \mathcal{M}_{i,l}^{(0)}} \langle w_{r'}, v_{i,l} \rangle \mathbb{I}_{v_{i,l} \in \mathcal{V}(X)} \overline{\mathrm{ReLU}}(\langle w_{r'}, x_p \rangle) \right)$$

$$= \mathbb{I}_{v_{i,l} \in \mathcal{V}(X)} \overline{\mathrm{ReLU}}(\langle w_r, v_{i,l} \rangle) \left( 1 - (1/\theta) \sum_{r' \in \mathcal{M}_{i,l}^{(0)}} \overline{\mathrm{ReLU}}(\langle w_{r'}, v_{i,l} \rangle) \langle w_{r'}, v_{i,l} \rangle \right)$$

$$\times \sum_{p \in \mathcal{P}_{v_{i,l}}(X)} (1+q) z_p^{q+1}.$$

As the term $(1/\theta) \sum_{r' \in \mathcal{M}_{i,l}^{(0)}} \overline{\mathrm{ReLU}}(\langle w_{r'}, v_{i,l} \rangle) \langle w_{r'}, v_{i,l} \rangle$ is small at the intial stage compared with the constant 1, we have

$$\Lambda_{i,l}^{(t+1)} \approx \Lambda_{i,l}^{(t)} + \tilde{\Theta} \left( \frac{\eta}{k} \right) \overline{\mathrm{ReLU}}(\langle w_r, v_{i,l} \rangle).$$

$\square$

Using Claim E.5, and $\tilde{\Omega}(\sigma_0) \leq \Lambda_{i,l}^{(0)} \leq \tilde{O}(\sigma_0)$, we have the following result:

**Claim H.8.** *Suppose Assumption H.1 holds and Induction Hypothesis C.3 holds for every iteration. Define $T_0 := \tilde{\Theta} \left( \frac{k}{\eta \sigma_0^{q-1}} \right)$. We have that when $t \geq T_0$, it satisfies $\Lambda_{i,l}^{(t)} \geq \Theta \left( \frac{1}{\mathrm{polylog}(k)} \right)$.*

### H.2 PROOF OF THEOREM H.2

#### H.2.1 DIAGONAL CORRELATIONS

**Lemma H.9.** *Suppose Assumption H.1 holds and Induction Hypothesis C.3 holds for all iterations $< t$. We have*

$$\forall v_{i,l} \in \mathcal{V}: \quad \Lambda_{i,l}^{(t)} \leq \min \left\{ \sqrt{\frac{\theta}{|\mathcal{M}_{i,l}^{(0)}|}}, \tilde{O}(1) \right\}.$$

*Proof.* Suppose we are now at some iteration $t > T_0$. In this stage, $\Lambda_{i,l}^{(t)} \geq 1/\mathrm{polylog}(k)$. As $T_0 = \tilde{\Theta} \left( \frac{k}{\eta \sigma_0^{q-1}} \right)$ and $\eta \leq \frac{1}{\mathrm{poly}(k)}$, we have

$$-\langle \nabla_{w_r} L(X), v_{i,l} \rangle = \sum_{p \in \mathcal{P}_{v_{i,l}}(X)} A_{r,p} \left( z_p \mathbb{I}_{v_{i,l} \in \mathcal{V}(X)} - \frac{1}{\theta} \sum_{r' \in \mathcal{M}_{i,l}^{(0)}} \langle w_{r'}, v_{i,l} \rangle \mathbb{I}_{v_{i,l} \in \mathcal{V}(X)} \overline{\mathrm{ReLU}}(\langle w_{r'}, x_p \rangle) \right)$$

$$= \mathbb{I}_{v_{i,l} \in \mathcal{V}(X)} [\langle w_r, v_{i,l} \rangle]^+ \left( 1 - (1/\theta) \sum_{r' \in \mathcal{M}_{i,l}^{(0)}} \langle w_{r'}, v_{i,l} \rangle^2 \right) \sum_{p \in \mathcal{P}_{v_{i,l}}(X)} 2 z_p^2.$$

Then we have

$$[\langle w_r^{(t+1)}, v_{i,l} \rangle]^+ \leq [\langle w_r^{(t)}, v_{i,l} \rangle]^+ + \tilde{O} \left( \frac{\eta}{k} \right) [\langle w_r^{(t)}, v_{i,l} \rangle]^+$$

Taking the maximum on both side and as we are at $t > T_0$, we have

$$\max_{r \in \mathcal{M}_{i,l}^{(0)}} [\langle w_r^{(t+1)}, v_{i,l} \rangle]^+ \leq \max_{r \in \mathcal{M}_{i,l}^{(0)}} [\langle w_r^{(t)}, v_{i,l} \rangle]^+ \left( 1 + \tilde{O} \left( \frac{\eta}{k} \right) \right).$$

When $t \leq T = T_0 + \tilde{O}\left( \frac{k}{\eta} \right)$, we have

$$\Lambda_{i,l}^{(t)} \leq \tilde{O}(1).$$

Besides, we also need

$$1 - (1/\theta) \sum_{r' \in \mathcal{M}_{i,l}^{(0)}} \langle w_{r'}, v_{i,l} \rangle^2 \geq 0,$$

which means

$$\Lambda_{i,l}^{(t)} \leq \sqrt{\frac{\theta}{|\mathcal{M}_{i,l}^{(0)}|}}.$$

This condition shows that when $\Lambda_{i,l}^{(t)} \to \sqrt{\frac{\theta}{|\mathcal{M}_{i,l}^{(0)}|}}$, the increase on the positive correlations tends to zero and the training process becomes to converge. $\square$

**Lemma H.10.** *Suppose Assumption H.1 holds and Induction Hypothesis C.3 holds for all iterations $< t$. We have*

$$\forall v_{i,l} \in \mathcal{V}, \forall r \in \mathcal{M}_{i,l}^{(0)} : \quad \langle w_r^{(t)}, v_{i,l} \rangle \geq -\tilde{O}(\sigma_0).$$

*Proof.* We start with any iteration $t$ that is $\langle w_r^{(t)}, v_{i,l} \rangle \leq -\tilde{\Omega}(\sigma_0)$ to see how negative the next iteration will be. Without loss of generality, we consider the case when $\langle w_r^{(t')}, v_{i,l} \rangle \leq -\tilde{\Omega}(\sigma_0)$ holds for every $t' \geq t$. Then based on Claim H.6 and when we assum $\langle w_r^{(t')}, v_{i,l} \rangle \leq -\tilde{\Omega}(\sigma_0)$, $A_{r,p} = 0$, we have

$$\langle w_r^{(t+1)}, v_{i,l} \rangle \geq \langle w_r^{(t)}, v_{i,l} \rangle - \gamma \tilde{O}(\sigma_0^q) - \tilde{O}(\sigma_0^{2q-1/2}) - \tilde{O}((\sigma_0^{2q+1})(\gamma k)^{2q}) \cdot P.$$

When $t \leq T_0$, we have

$$\begin{aligned}
\langle w_r^{(t+1)}, v_{i,l} \rangle &\geq \langle w_r^{(t)}, v_{i,l} \rangle - \gamma \tilde{O}(\sigma_0^q) - \tilde{O}(\sigma_0^{2q-1/2}) - \tilde{O}((\sigma_0^{2q+1})(\gamma k)^{2q}) \cdot P \\
&\geq -\tilde{O}(\sigma_0) - \eta T_0(\gamma \tilde{O}(\sigma_0^q) + \tilde{O}(\sigma_0^{2q-1/2}) + \tilde{O}((\sigma_0^{2q+1})(\gamma k)^{2q}) \cdot P) \\
&\geq -\tilde{O}(\sigma_0).
\end{aligned}$$

When $t \in [T_0, T]$, we have

$$\begin{aligned}
\langle w_r^{(t+1)}, v_{i,l} \rangle &\geq \langle w_r^{(T_0)}, v_{i,l} \rangle - \eta(\gamma \tilde{O}(\sigma_0^q) + \tilde{O}(\sigma_0^{2q-1/2}) + \tilde{O}((\sigma_0^{2q+1})(\gamma k)^{2q}) \cdot P) \\
&\geq -\tilde{O}(\sigma_0) - \eta(T - T_0)(\gamma \tilde{O}(\sigma_0^q) + \tilde{O}(\sigma_0^{2q-1/2}) + \tilde{O}((\sigma_0^{2q+1})(\gamma k)^{2q}) \cdot P) \\
&\geq -\tilde{O}(\sigma_0).
\end{aligned}$$

$\square$

### H.2.2 OFF-DIAGONAL CORRELATIONS

**Lemma H.11.** *Suppose Assumption H.1 holds and Induction Hypothesis C.3 holds for all iterations $< t$. Then*

$$\forall v_{i,l} \in \mathcal{V}, \forall r \in \mathcal{M}_{i,l}^{(0)}, \text{ for } v_{j,l'} \neq v_{i,l} : \quad |\langle w_r^{(t)}, v_{j,l'} \rangle| \leq \tilde{O}(\sigma_0).$$

*Proof.* **Stage I.** We first consider the stage when $t \leq T_0$. For every $r \in \mathcal{M}_{i,l}^{(0)}$, using Claim H.6, we have

$$\mathbb{E}_{(X,y) \sim \mathcal{Z}}[-\langle \nabla_{w_r} L(X), v_{j,l'} \rangle]$$

$$\leq \mathbb{E}_{(X,y) \sim \mathcal{Z}} \left[ \sum_{p \in \mathcal{P}_{v_{i,l}}(X)} A_{r,p} \right] (\gamma + \tilde{O}(\sigma_0^{q-\frac{1}{2}})) + \gamma \tilde{O}(\sigma_0^q) + \tilde{O}(\sigma_0^{2q-1/2}) + \tilde{O}((\sigma_0^{2q+1})(\gamma k)^{2q}) \cdot P + \tilde{O}(\sigma_0^q)$$

$$\times \mathbb{E}_{(X,y) \sim \mathcal{Z}} \left[ \sum_{p \in \mathcal{P}_{v_{j,l'}}(X)} \left( z_p \mathbb{I}_{v_{j,l'} \in \mathcal{V}(X)} - \frac{1}{\theta} \sum_{r' \in \mathcal{M}_{j,l'}^{(0)}} \langle w_{r'}, v_{j,l'} \rangle \mathbb{I}_{v_{j,l'} \in \mathcal{V}(X)} \overline{\text{ReLU}}(\langle w_{r'}, x_p \rangle) \right) \right]$$

$$\leq \tilde{\Theta}\left(\frac{1}{k}\right) \overline{\text{ReLU}}(\langle w_r, v_{i,l} \rangle)(\gamma + \tilde{O}(\sigma_0^{q-\frac{1}{2}})) + \tilde{O}\left(\frac{\sigma_0^q}{k}\right).$$

From Claim H.7, we have that

$$\tilde{\Theta}\left(\frac{\eta}{k}\right) \sum_{t=0}^{T_0-1} \overline{\text{ReLU}}(\langle w_r^{(t)}, v_{i,l}\rangle) = \Lambda_{i,l}^{(T_0)} - \Lambda_{i,l}^{(0)} \leq \frac{1}{\text{polylog}(k)}.$$

Thus, when $t \leq T_0$,

$$|\langle w_r^{(t)}, v_{j,l'}\rangle| \leq |\langle w_r^{(0)}, v_{j,l'}\rangle| + \gamma + \tilde{O}(\sigma_0^{q-\frac{1}{2}}) + T_0 \tilde{O}\left(\frac{\eta \sigma_0^q}{k}\right) \leq \tilde{O}(\sigma_0).$$

**Stage II.** When $t \in [T_0, T]$, we have

$$|\langle w_r^{(t)}, v_{j,l'}\rangle| \leq |\langle w_r^{(T_0)}, v_{j,l'}\rangle| + \tilde{O}\left(\frac{\eta(T-T_0)}{k}\right) \cdot (\overline{\text{ReLU}}(\langle w_r, v_{i,l}\rangle)(\gamma + \tilde{O}(\sigma_0^{q-\frac{1}{2}})) + \tilde{O}(\sigma_0^q))$$

$$\leq \tilde{O}(\sigma_0).$$

$\square$

### H.2.3   LOTTERY WINNING: KERNELS INSIDE $\mathcal{M}_{i,l}^{(0)}$

**Lemma H.12.** *Suppose Assumption H.1 holds and Induction Hypothesis C.3 holds for all iterations $< t$. Then*

$$\forall v_{i,l} \in \mathcal{V}, \forall r \notin \mathcal{M}_{i,l}^{(0)}: \quad \langle w_r^{(t)}, v_{i,l}\rangle \leq \tilde{O}(\sigma_0).$$

*Proof.* When $r \in \mathcal{M}_{j,l'}^{(0)}, (v_{j,l'} \neq v_{i,l})$, we have prove that $\langle w_r^{(t)}, v_{i,l}\rangle \leq \tilde{O}(\sigma_0)$ in Lemma H.11. So we only prove the case when $r \notin \cup_{i\in[k],l\in[2]}\mathcal{M}_{i,l}^{(0)}$.

We assume that there exists an $w_{r'} \notin \cup_{i\in[k],l\in[2]}\mathcal{M}_{i,l}^{(0)}$ such that induction hypothesis C.3 (a)-(c) holds for every $(X,y) \in \mathcal{Z}$. We want to see if the sequence $\langle w_{r'}^{(t)}, v_{i,l}\rangle$ will increase more quickly than $\max_{r\in\mathcal{M}_{i,l}^{(0)}} \langle w_r^{(t)}, v_{i,l}\rangle$.

**Stage I.** We first consider when $t \leq T_0$. In this stage, $\Lambda_{i,l}^{(t)} \leq \varrho$. We define two sequences. First, we take $w_{r^*} = \text{argmax}_{r\in\mathcal{M}_{i,l}^{(0)}} \langle w_r^{(0)}, v_{i,l}\rangle$ and define $x_t := \langle w_{r^*}^{(t)}, v_{i,l}\rangle \cdot \left(\frac{s}{qk}\right)^{1/(q-1)} \frac{1}{\varrho}$. We also define $y_t = \max\{\langle w_{r'}^{(t)}, v_{i,l}\rangle \cdot \left(\frac{s}{qk}\right)^{1/(q-1)} \frac{1}{\varrho}, \sigma_0\}$. From Claim H.7, when $t \leq T_0$, we have that

$$\langle w_{r^*}^{(t+1)}, v_{i,l}\rangle = \langle w_{r^*}^{(t)}, v_{i,l}\rangle + \Theta\left(\frac{s\eta}{k}\right)\overline{\text{ReLU}}(\langle w_{r^*}^{(t)}, v_{i,l}\rangle)$$

$$\geq \langle w_{r^*}^{(t)}, v_{i,l}\rangle + \Theta\left(\frac{s\eta}{k}\right)\frac{1}{q\varrho^{q-1}}([\langle w_{r^*}, v_{i,l}\rangle]^+)^q.$$

Let $S = \left(\frac{1+C/(\log(k)-C)}{1+1/\log(k)}\right)^{q-2}, C > 1$. We have

$$\langle w_{r'}^{(t+1)}, v_{i,l}\rangle = \langle w_{r'}^{(t)}, v_{i,l}\rangle + \Theta\left(\frac{s\eta}{k}\right)\overline{\text{ReLU}}(\langle w_{r'}^{(t)}, v_{i,l}\rangle)$$

$$\leq \langle w_{r'}^{(t)}, v_{i,l}\rangle + \Theta\left(\frac{s\eta}{k}\right)\frac{1}{q\varrho^{q-1}}([\langle w_{r'}^{(t)}, v_{i,l}\rangle]^+)^q S.$$

Then following the same process as the proof of Lemma F.5, we have

$$\langle w_{r'}^{(t)}, v_{i,l}\rangle \leq \tilde{O}(\sigma_0).$$

The proof of Stage II is also similar to Lemma F.5. $\square$

### H.2.4 NOISE CORRELATION

As our noise correlation result is similar to Lemma F.6, we don't repeat it here but just prove it holds under MAE framework and under our new parameter assumptions.

*Proof of Lemma F.6 under MAE framework.* For every $r \in [km]$, for every $(X^*, y^*) \in \mathcal{Z}$ and every $p^* \in [P]$, we have that

$$\langle -\nabla_{w_r} L(X), \xi_{p^*} \rangle = \sum_{p \in [P]} A_{r,p} \left( \langle x_p, \xi_{p^*} \rangle - \frac{1}{\theta} \sum_{r' \in [km]} \langle w_{r'}, \xi_{p^*} \rangle \overline{\text{ReLU}}(\langle w_{r'}, x_p \rangle) \right).$$

When $X \neq X^*$, we have $|\langle x_p, \xi_{p^*} \rangle| \leq \tilde{O}(\sigma_p) \leq o(1/\sqrt{d})$; and when $X = X^*$ but $p \neq p^*$, we have $|\langle x_p, \xi_{p^*} \rangle| \leq \tilde{O}(\sigma_p) \leq o(1/\sqrt{d})$. Therefore, we have

$$\mathbb{E}_{(X,y) \sim \mathcal{Z}} \left[ \langle -\nabla_{w_r} L(X), \xi_{p^*} \rangle \right] = \mathbb{E}_{(X,y) \in \mathcal{Z}} \left[ \mathbb{I}_{X=X^*} \langle -\nabla_{w_r} L(X), \xi_{p^*} \rangle + \mathbb{I}_{X \neq X^*} \langle -\nabla_{w_r} L(X), \xi_{p^*} \rangle \right].$$

Now we begin to prove (a). For every $v_{i,l} \in \mathcal{V}$, for every $r \in \mathcal{M}_{i,l}^{(0)}$, for every $p^* \in \mathcal{P}_{v_{i,l}}(X^*)$, using the induction hypothesis C.3, when $t \in [0, T_0]$, we have that for the first term,

$$\mathbb{E}_{(X,y) \sim \mathcal{Z}} \left[ \mathbb{I}_{X=X^*} \langle -\nabla_{w_r} L(X), \xi_{p^*} \rangle \right]$$

$$= \frac{1}{N} \mathbb{E}_{(X^*,y^*) \sim \mathcal{Z}} \left[ A_{r,p^*} \left( \langle x_{p^*}, \xi_{p^*} \rangle - \frac{1}{\theta} \sum_{r' \in [km]} \langle w_{r'}, \xi_{p^*} \rangle \overline{\text{ReLU}}(\langle w_{r'}, x_{p^*} \rangle) \right) \pm o \left( \frac{1}{\sqrt{d}} \right) \pm \tilde{o}(\sigma_0) \overline{\text{ReLU}}(\Lambda_{i,l}^{(t)}) \right]$$

For the second term,

$$\mathbb{E}_{(X,y) \sim \mathcal{Z}} \left[ \mathbb{I}_{X \neq X^*} \langle -\nabla_{w_r} L(X), \xi_{p^*} \rangle \right] = \pm o \left( \frac{1}{\sqrt{d}} \right) \pm \tilde{o}(\sigma_0) \overline{\text{ReLU}}(\Lambda_{i,l}^{(t)})$$

Thus, we have

$$\langle w_r^{(t+1)}, \xi_{p^*} \rangle \leq \langle w_r^{(t)}, \xi_{p^*} \rangle + \tilde{O} \left( \frac{\eta}{N} \right) \tilde{o}(\sigma_0) \overline{\text{ReLU}}(\Lambda_{i,l}^{(t)}) + o \left( \frac{\eta}{\sqrt{d}} \right) + \tilde{o}(\eta \sigma_0) \overline{\text{ReLU}}(\Lambda_{i,l}^{(t)}),$$

Now we use the results from Lemma H.9, when $t \leq T_0$,

$$\langle w_r^{(t)}, \xi_{p^*} \rangle \leq \langle w_r^{(t-1)}, \xi_{p^*} \rangle + \tilde{o}(\eta \sigma_0) \overline{\text{ReLU}}(\Lambda_{i,l}^{(t)}) + o \left( \frac{\eta}{\sqrt{d}} \right)$$

$$\leq \langle w_r^{(0)}, \xi_{p^*} \rangle + \tilde{o}(\eta \sigma_0) \sum_{t=0}^{T_0-1} \overline{\text{ReLU}}(\Lambda_{i,l}^{(t)}) + o \left( \frac{\eta T_0}{\sqrt{d}} \right)$$

$$\leq \tilde{o}(\sigma_0).$$

So when $N \geq \tilde{\omega} \left( \frac{k}{\sigma_0^{q-1}} \right)$ and $\sqrt{d} \geq \tilde{\omega}(k/\sigma_0^{q-1})$, we have $\langle w_r^{(t)}, \xi_{p^*} \rangle \leq \tilde{o}(\sigma_0)$. Therefore, for $t \in [T_0, T]$, we have

$$\langle w_r^{(t)}, \xi_{p^*} \rangle \leq \langle w_r^{(T_0)}, \xi_{p^*} \rangle + \tilde{O} \left( \frac{\eta(t - T_0)}{N} \right) + o \left( \frac{\eta(t - T_0)}{\sqrt{d}} \right) \leq \tilde{o}(\sigma_0),$$

when $\sqrt{d} \geq \tilde{\omega}(k)$. Following the similar process, we could also prove (b)-(e). □

### H.2.5 PROOF OF THEOREM H.2

Theorem H.2 can be easily obtained following the similar steps in the proof of Theorem C.4 when we have Lemma H.10-Lemma H.12.

### H.2.6 PROOF OF THEOREM H.3

Theorem H.3 can be easily obtained following the same steps in the proof of Theorem G.2 in Section G.

### H.3 DISCUSSION ON BEiT METHODS

We have proved that the pretraining phase can capture all features both under Teacher-Student framework and MAE framework. Now we have a discussion on BEiT framework (Bao et al., 2021). For BEiT, if we regard the pretrained encoder of BEiT as a fixed teacher and stuck an additional layer to map the patch token feature of BEiT encoder to discrete pseudo label, then this setting becomes very similar to our setting. The only different is the BEiT encoder (teacher) is fixed, while our teacher encoder is learned online (updated along with the weights of the student). Since there are a lot of similarities between these two frameworks, our proof methods can naturally extend to BEiT with the suitable choices of additional layers.

