# OpenReview forum: "Towards Understanding Why Mask Reconstruction Pretraining Helps in Downstream Tasks"
_ICLR.cc/2023/Conference — ICLR 2023 poster_

### Official Review · Reviewer_TnqJ · 2022-10-24

**Confidence:** 4
**Clarity, Quality, Novelty And Reproducibility:** Check the weakness part for more deta…
**Correctness:** 3
**Technical Novelty And Significance:** 3
**Empirical Novelty And Significance:** 4
**Recommendation:** 6

**Strength And Weaknesses:**

- Strength:
  - The authors provide a very detailed theoretical analysis based on the proposed teacher-student MRP framework.
  - The idea of modeling "semantics" as several attribute vectors associated with classes is interesting.
  - After theoretical analysis, the authors further conduct implementations to support their analysis.

- Weakness:
  - In the MRP architecture proposed in Sec. 3.2, are there any requirements of the teacher model parameters, like they should be the EMA of the parameters of the student model?
  - The analysis heavily bases on the assumption that the "semantics" of the downstream dataset should be a subset of the pre-training dataset. Does it suggest that the downstream dataset should have similar distribution with the pre-training one, which is actually a very strong and unrealistic assumption? If not, what is the difference?
  - The authors claim that the MRP pre-trained models would contain all the semantics of the downstream tasks, each at least corresponding to one kernel. Does it suggest to do transfer learning, if I can find the specific wining ticket of a given downstream task, pruning can actually replace fully fine-tuning?
  - I wonder whether the MRP pre-trained models would be worse when the pre-training dataset does not contain all the semantics in the downstream datasets compared with supervised pre-training, since the latter replies on partial semantics matching while MRP might response to more semantics according to Fig. 3.
- Could you further explain what the role of masking \epsilon is, since it seems not to appear in your analysis in Sec. 4?



**Summary Of The Paper:**

This paper is overall a quite interesting paper. Based on an abstract teacher-student framework with simple network architecture, the authors explore how a masked reconstruction pre-training architecture performs feature learning and becomes beneficial for downstream tasks by modeling "semantics" as attribute vectors associated with different annotation labels. With the above analysis, the authors further conduct a MRP pre-training with ResNet and demonstrate more superior performance compared with supervised pre-training.



**Summary Of The Review:**

This is overall an interesting paper solid theoretical analysis, which will provide good insights for future researches and algorithm development.

---

> ### Author Response · Authors · 2022-11-15
> **#Reply to Reviewer TnqJ - Part II**
>
> **Q4. I wonder whether the MRP pre-trained models would be worse when the pre-training dataset does not contain all the semantics in the downstream datasets...**
>
> **Reply:**
> Following the similar analysis to the reply of Q2, we cannot conclude that MRP pretrained models will be worse if the semantic sets of pretraining and downstream dataset are separated. The main difference between the two pre-training models on the downstream tasks is the initialized weights. We set the initial weights of downstream tasks as the pretrained weights by MRP, while in supervised pre-training, the initial weights are initialized by random Gaussian.
> As our training networks are over-parameterized, i.e., $m\in[polylog k, \sqrt{k}]$ and the size of the lottery ticket winning set is smaller than $O(polylog k)$ at the end of pre-training, the number of kernel weights that finally capture the features is much smaller than the total number of kernels. For the remaining kernels, after pretraining, they still can capture new features in the downstream fine-tuning phase. Thus, we cannot conclude that MRP pretrained models will be worse.
>
> **Q5. Could you further explain what the role of masking $\epsilon$ is, since it seems not to appear in your analysis in Sec. 4?**
>
> **Reply:** As we make an expectation on the pretraining loss function $L(H;\epsilon)$ over $\epsilon$ when doing gradient descent, the expectation of $\epsilon$ ( here $\mathbb{E}[\epsilon]=\theta$) really matters in the analysis. More specifically, we have
> $$L(H;X)=\mathbb{E}[L(H;X,\epsilon)]=\frac{1}{2}\sum _ {r\in[km]}\left(\sum _ {p}\overline{ReLu}(\langle \hat{w} _ r,x _ p\rangle)-\sum _ {p}\overline{ReLu}(\langle w _ r,x _ p\rangle)\right)^2+\frac{1}{2}\left(\frac{1}{\theta}-1\right)\sum _ {r\in[km]} \sum _ {p} \overline{ReLu}(\langle w _ r,x _ p\rangle),$$ which is shown in Eq.(3).
> From the definition of $\theta$,  when $\theta\to 1$, this means $P(\epsilon_p=1)\to 1$, i.e., there is nearly no mask. Then according to our choice of the teacher’s network parameters, when $\theta\to 1$, $\hat{w}_r\approx w_r$,  which means that the loss keeps small and so there is nearly no update of parameters of student and teacher models. In this case, the student model cannot learn the useful semantic features in the data. When $\theta\to 0$ (i.e., we mask all the data), $\hat{w}_r\to\infty$ and $L(H;X)\to\infty$. In this case, the student model also cannot learn useful semantic features in the data. Overall, there is a tradeoff on the mask ratio $\theta$. The analysis also holds under the MAE model. When $\theta\to 0$, $L(H;X)\to\infty$. When $\theta\to 1$, there is a trivial answer to make $L(H;X)\to 0$. In both above cases, no semantic features will be learned. But it is difficult to determine the exact optimal mask ratio based on our analysis, because we analyze the change of correlation scores through approximations (by $O, \Omega,\Theta$), which hide the influence of constants. We have added the above analysis to   Appendix E.1.

---

> ### Author Response · Authors · 2022-11-15
> **Reply to Reviewer TnqJ - Part I**
>
> Thanks a lot for your meticulous reading as well as many useful, detailed comments. We have carefully addressed your comments in the following.
>
> **Q1. ...are there any requirements of the teacher model parameters...?**
>
> **Reply:** Wang et al. [1] have shown that for MRP model, the asymmetry between student and teacher is sufficient to achieve good performance, which is indeed also observed in other MRP works, e.g. [2, 3]. Here asymmetry means that student and teacher have different parameters. So following this asymmetry spirit, we set the parameters of the teacher model to be $\hat{w}_r=\tau w_r, r\in[km]$, where $w_r, r \in [km]$ are parameters of the student models and $\tau>1$. This ensures the asymmetry between student and teacher.  We did not update the teacher model parameters by the EMA of the student model, since EMA contains all historical parameters which indeed couples all historical learning process together and greatly increases the difficulty for analysis.
>
> [1] Wang, X., et al. On the Importance of Asymmetry for Siamese Representation Learning. CVPR 2022.
>
> [2] Xingbin Liu, et al, Exploring Target Representations for Masked Autoencoders
>
> [3] Chen Wei, et al, Masked Feature Prediction for Self-Supervised Visual Pre-Training, CVPR 2022.
>
> **Q2. ...have similar distribution with the pre-training one, which is actually a very strong and unrealistic assumption?...**
>
> **Reply:** Besides the case that the pre-training and downstream tasks share the same data distribution, our analysis can be extended to more general transfer learning cases. Denote the feature set of pretraining dataset as $\mathcal{V}$ and the feature set of transfer learning downstream dataset as $\mathcal{V}’$. We discuss different cases in Sec. 4.2, which include 1) the pretraining and downstream tasks share the same feature set (i.e., $\mathcal{V}=\mathcal{V}’$) but have different data distributions (e.g. different ratio of single-view data and multi-view data); 2) the feature set of pretraining dataset cover the feature set of downstream dataset, i.e., $\mathcal{V}\subset \mathcal{V}’$; 3) the pretraining and downstream tasks share the partial feature set, i.e., $\mathcal{V}\cap \mathcal{V}’\neq \emptyset$. For case (1) and (2), following the similar proof process of Theorem 2, the fine-tuned model can also obtain high classification accuracy on downstream tasks. For case (3), since we analyze the modern networks which are usually over-parameterized,  only a small percentage of kernels are used to capture all features in the pretraining data. Specifically, the size of the lottery ticket winning set will be $O(polylog k)$ at the end of training. As we set  $m\in [ploylog k, \sqrt{k}]$, this means that for sufficiently large $k$, the remaining kernels that do not capture any features are of large numbers. Thus, for the remaining kernels, they can randomly capture new features in $\mathcal{V}’$ of the downstream task, and share the similar feature learning process with conventional supervised learning. In this way, MRP still improves the overall transfer learning performance.
>
> Besides, the experimental results in Sec. 5.2 also support our above analysis. Table 1 shows that for the transfer learning classification on VOC07 dataset, MRP enjoys better performance than supervised training. Consider the fact that  the pretraining dataset, i.e. ImageNet, and the downstream dataset, VOC07, share many different categories and thus have different semantics; the above empirical results validate our above analysis. We have added the above analysis to Sec. 4.2.
>
> **Q3. ...pruning can actually replace fully fine-tuning?**
>
> **Reply:**  Based on our theoretical results, we can find the specific winning tickets by using iterative pruning methods. However, even after pruning, there still needs fine-tuning in practice. This is because 1) pruning only finds specific winning tickets which only have relatively larger correlation scores with the semantics, 2) finding more suitable correlation scores between those tickets and the semantics still needs a labeled dataset for fine-tuning so that these correlation scores are more suitable for a downstream task. Indeed, given a self-supervised pre-training model, Chen et al. [4] already explored the iterative pruning method to find the specific winning ticket for a downstream task, and achieved promising results. Please refer to [4] for more details.
>
> [4] Chen, T., et al. (2021, CVPR). The lottery tickets hypothesis for supervised and self-supervised pre-training in computer vision models.

---

### Official Review · Reviewer_GkZ3 · 2022-10-24

**Confidence:** 4
**Correctness:** 4
**Technical Novelty And Significance:** 4
**Empirical Novelty And Significance:** 2
**Recommendation:** 8

**Clarity, Quality, Novelty And Reproducibility:**

The paper is well written and the definitions, assumptions, and theorems are cleanly stated and make sense to the reader. Note that I did not read the proofs and can not speak of their correctness.

**Strength And Weaknesses:**

Strength:

1) The paper provides a strong theoretical analysis, supporting the benefit of MRP for SSL.

2) It is really hard to pick a minimal set of assumptions about the data and model that are just sufficient to prove the advantage of MRP in SSL. I like how the paper mathematically defines multi/sing-view distributions and picks a simple network architecture and manages to prove the benefit of MRP over SL.


Weaknesses:

1) There are several limiting assumptions in the multi-view data distribution and network architectures that do not apply in practice. However, I understand that these restrictions are required for a sound theoretical analysis.

2) The theoretical analysis in the paper does not result in a new practical approach, instead it supports the success of the existing methods in a limited setting.

**Summary Of The Paper:**

The paper provides a theoretical analysis that explains the benefit of MRP in SSL. Utilizing the multi-view data distribution and certain assumptions about the model architecture, the paper shows that the encoder trained with MRP can capture *all* the discriminative semantics of each semantic class in the pretraining dataset. This gives an advantage to the SSL+SL compared the SL only model since in SL training the model only captures *some* semantics due to lottery ticket hypothesis.

**Summary Of The Review:**

Recent works have shown practically that supervised fine-tuning of the encoder learned via MRP remarkably surpasses the SL training from scratch but I am not aware of any theoretical work that supports the benefit of MRP. The paper provides new theoretical justification for the success of MRP in practice. Overall, the theoretical contributions of the paper are significant and novel.

---

> ### Author Response · Authors · 2022-11-15
> **#Reply to Reviewer GkZ3**
>
> Thanks a lot for your meticulous reading as well as many useful, detailed comments. We have carefully addressed your comments in the following.
>
> **Q1. There are several limiting assumptions in the multi-view data distribution and network architectures that do not apply in practice. However, I understand that these restrictions are required for a sound theoretical analysis.**
>
> **Reply:** Thanks for your understanding.  We admit that the data assumptions are complicated, and the neural networks do not cover all kinds of popular networks. As you mentioned, all these simplifications are necessary for rigorous theoretical analysis. Indeed, to remedy these issues, we also use experiments to verify our data assumptions and theoretical implications on different networks.
>
> For the multi-view data assumptions,  we have also empirically investigated and validated this assumption on real data, e.g. the widely used ImageNet dataset, by using both ResNet and vision transformers. Indeed, Allen-Zhu et al. also did many experiments to justify it (see [1, Appendix F.3]), and further used this assumption for network performance analysis.
>
> For network architectures, we also investigate our theoretical implications on two widely used network architectures, including CNN (ResNet50) and also vision transformer (ViT-small). On these two classical networks, we well observe that MRP can capture more semantics than the conventional supervised learning across several downstream tasks, e.g. classification, transfer learning, object detection.
>
>
> [1] Allen-Zhu, Z., & Li, Y. (2020). Towards understanding ensemble, knowledge distillation and self-distillation in deep learning. arXiv preprint arXiv:2012.09816.
>
> **Q2. The theoretical analysis in the paper does not result in a new practical approach, instead it supports the success of the existing methods in a limited setting.**
>
> **Reply:** Main purpose of our paper is to theoretically explain why features obtained by mask-reconstruction pre-training can help in downstream tasks. We theoretically prove that MRP can capture more features compared with supervised cases, which justify why MRP helps in downstream tasks.  Experimental results also align with our theoretical implications. Thus, this theoretical work provides an explanation to the superiority of MRP and deepens our understanding of mask-reconstruction based self-supervised methods.
>
> Besides, this theoretical work may also motivate one practical approach in the downstream fine-tuning. As we have proved that features are captured by some part of kernels in the pretrained network. We can extract the subnetworks that capture the features by pruning and reduce the model size in the downstream tasks, which is more computationally efficient. Similar experimental works have been done in [2]. We also have the discussion about pruning in Reply to Reviewer TnqJ. Q3.
>
>
> [2] Chen, T., Frankle, J., Chang, S., Liu, S., Zhang, Y., Carbin, M., & Wang, Z. (2021). The lottery tickets hypothesis for supervised and self-supervised pre-training in computer vision models. In Proceedings of the IEEE/CVF Conference on Computer Vision and Pattern Recognition (pp. 16306-16316).

---

### Official Review · Reviewer_J85J · 2022-10-25

**Confidence:** 3
**Correctness:** 4
**Technical Novelty And Significance:** 3
**Empirical Novelty And Significance:** 2
**Recommendation:** 6

**Clarity, Quality, Novelty And Reproducibility:**

Clarity

There are parts where the work is not clear and has typos. I've listed two glaring examples here, but I'm sure there are more throughout the text that should be re-checked.

Top of page 8: "in the pretraiing"
The use of "semantic" as a singular noun is somewhat strange. I haven't seen this before and the authors should consider a different word choice.

Quality and Novelty

While some data assumptions come from a previous work, the problem setting is unique and the analysis is valuable.

**Strength And Weaknesses:**

Strengths
- The problem is clearly an important one to study, and hasn't been adequately addressed by the existing literature.
- The use of the multi-view data assumption from  (Allen-Zhu & Li, 2020) is clever.
- The empirical results match the theoretical findings, from multi-view data assumption to test performance on downstream tasks.

Weaknesses
- The assumption that pre-training and downstream datasets follow the same distribution is in some ways limited. While the authors discuss how this framework is useful, it would be nice to see more analysis of "transfer learning", since this is much of the appeal of pre-training approaches.
- Dataset assumptions are fairly strict, though the authors reason about how these assumptions are justified.

**Summary Of The Paper:**

The authors study the mask-reconstruction pretraining (MRP) approach and its ability to generalize well for downstream tasks. They show that under certain dataset and task assumptions, MRP can perform well for downstream classification tasks whereas supervised learning cannot.

**Summary Of The Review:**

The theory is both correct, novel, and matches the empirical results of MRP, which is an important problem to study. While there are important results, the dataset assumptions cast some doubt on how generally applicable the results are.

---

> ### Author Response · Authors · 2022-11-15
> **#Reply to Reviewer J85J**
>
> Thanks a lot for your meticulous reading as well as many useful, detailed comments. We have carefully addressed your comments in the following.
>
> **Q1. The assumption that pre-training and downstream datasets follow the same distribution is in some ways limited. While the authors discuss how this framework is useful, it would be nice to see more analysis of "transfer learning", since this is much of the appeal of pre-training approaches.**
>
> **Reply:** Besides the case that the pre-training and downstream tasks share the same data distribution, our analysis can be extended to more general transfer learning cases. Denote the feature set of pretraining dataset as $\mathcal{V}$ and the feature set of transfer learning downstream dataset as $\mathcal{V}’$. We discuss different cases in Sec. 4.2, which include 1) the pretraining and downstream tasks share the same feature set (i.e., $\mathcal{V}=\mathcal{V}’$) but have different data distributions (e.g. different ratio of single-view data and multi-view data); 2) the feature set of pretraining dataset cover the feature set of downstream dataset, i.e., $\mathcal{V}\subset \mathcal{V}’$; 3) the pretraining and downstream tasks share the partial feature set, i.e., $\mathcal{V}\cap \mathcal{V}’\neq \emptyset$. For case (1) and (2), following the similar proof process of Theorem 2, the fine-tuned model can also obtain high classification accuracy on downstream tasks. For case (3), since we analyze the modern networks which are usually over-parameterized,  only a small percentage of kernels are used to capture all features in the pretraining data. Specifically, the size of the lottery ticket winning set will be $O(polylog k)$ at the end of training. As we set  $m\in [ploylog k, \sqrt{k}]$, this means that for sufficiently large $k$, the remaining kernels that do not capture any features are of large numbers. Thus, for the remaining kernels, they can randomly capture new features in $\mathcal{V}’$ of the downstream task, and share the similar feature learning process with conventional supervised learning. In this way, MRP still improves the overall transfer learning performance.
>
> Besides, the experimental results in Sec. 5.2 also support our above analysis. Table 1 shows that for the transfer learning classification on VOC07 dataset, MRP enjoys better performance than supervised training. Consider the fact that  the pretraining dataset, i.e. ImageNet, and the downstream dataset, VOC07, share many different categories and thus have different semantics; the above empirical results validate our above analysis. We have added the above analysis to Sec. 4.2.
>
> **Q2. Dataset assumptions are fairly strict, though the authors reason about how these assumptions are justified.**
>
> **Reply:**  Although the assumption of the dataset seems to be complicated, the assumption is actually intuitive. That is, one dataset contains both multi-view data and single-view data, where multi-view data consists of several independent discriminative semantics which can identify the semantic class of this multi-view data, while single-view data only contains a discriminative semantic. Moreover, we have also empirically investigated and validated this assumption on real data, e.g. the widely used ImageNet dataset. Indeed, Allen-Zhu et al. also did many experiments to justify it (see [1, Appendix F.3]), and further used this assumption for network performance analysis.
>
> [1] Allen-Zhu, Z., & Li, Y. (2020). Towards understanding ensemble, knowledge distillation and self-distillation in deep learning. arXiv preprint arXiv:2012.09816.
>
> **Q3. There are parts where the work is not clear and has typos. I've listed two glaring examples here, but I'm sure there are more throughout the text that should be re-checked. Top of page 8: "in the pretraining" The use of "semantic" as a singular noun is somewhat strange. I haven't seen this before and the authors should consider a different word choice.**
>
> **Reply:** Thanks for pointing it out. We have checked through the paper and corrected them accordingly. We also changed the use of semantics to ‘‘features’’.

---

### Official Review · Reviewer_1jPb · 2022-10-27

**Confidence:** 2
**Correctness:** 3
**Technical Novelty And Significance:** 3
**Empirical Novelty And Significance:** 3
**Recommendation:** 6

**Clarity, Quality, Novelty And Reproducibility:**

The paper writing is clear. The theoretical proof is novel and inspirational to the community.

**Strength And Weaknesses:**

Strengths:
1. The problem of why MRP works so well and is much better than supervised learning is quite interesting and important for the community. This paper is a pioneer in studying how semantic features are learned under the above problem.
2. The theoretical analysis covers both multi-view data and single-view data. It also covers both MAE-like methods, Teacher-Student MRP methods, and supervised learning methods. It also discusses different downstream tasks.

Weaknesses:
1. In this paper, it assumes the encoder in the student network is just a two-layer convolution + Smoothed ReLU network. I understand that this setting is easier to analyze. But in real-world applications, people usually use deeper models. And we know that some conclusions found in a simple model don't necessarily generalize to deeper models. So I wonder if it's possible to generalize the analysis to deeper models?
2. I am a bit confused about the definition of whether one semantic is captured by the kernel. In 4.1, If I understand correctly, when the correlation score is increased to `1/polylog(k)`, it means matched. Why would the threshold be set as that?
3. The experiments mainly cover visualization and accuracy. But those two proxies cannot indicate whether all discriminative semantics are captured in MAE/MRP. Is there any way we can quantitatively measure how many semantics in each class are captured by kernels?
4. As you mentioned in section 4.2, Transformer is not discussed because of its correlated manipulations and highly nonlinear attention. Is that possible that we remove the nonlinear part and make it analyzable?
5. Some typos such as `The pretraining dataset Z has` in Assumption1.

**Summary Of The Paper:**

This paper introduces a theoretical analysis to understand why mask reconstruction pre-training (MRP) works well in downstream tasks. It's found MRP can capture all discriminative semantics of each class in the pre-training dataset and thus utilize them to help downstream tasks. It provides solid proof, straightforward visualization and experiment result.

**Summary Of The Review:**

Overall, this paper targets an intriguing problem and provides solid proof. I'd lean to accept it.

---

> ### Author Response · Authors · 2022-11-15
> **#Reply to Reviewer 1jPb**
>
> Thanks a lot for your meticulous reading as well as many useful and insightful comments. We have carefully addressed your comments in the following.
>
> **Q1. In this paper, it assumes...So I wonder if it's possible to generalize the analysis to deeper models?**
>
> **Reply:** Thanks. We first would like to emphasize that although we theoretically analyze two-layered convolutional neural networks, we empirically investigate our theoretical implications on deep networks, e.g. ResNet50 in Sec. 5, and vision transformers (ViT)  in Appendix B, and find the good alignment between our theoretical implications on shallow networks and the empirical observations on deep networks.
>
> Moreover, this analysis method could be extended to deeper models under proper assumptions. At present, for our assumed multi-view data structure, the two-layered convolution network is already able to capture all semantics in the data, which also aligns with the experimental visualization and performance advantages of MRP. Similar to the discussions in [1], if we extend the multi-view data structure to a ''hierarchical'' multi-view data structure (i.e., there are multiple levels of features), then the theoretical analysis should naturally and directly be extended to the deeper networks. Intuitively speaking, adding more layers should expect to fit more complex data. But this extension needs more time and effort to theoretically justify the function of each layer, and cannot be finished in a few days.  We will try our best to explore more rigorous theoretical extensions.
>
> [1] Wen, Z., & Li, Y. (2021, July). Toward understanding the feature learning process of self-supervised contrastive learning. In International Conference on Machine Learning.
>
> **Q2. I am a bit confused about the definition of... Why would the threshold be set as that?**
>
> **Reply:** This threshold comes from the definition of the smoothed ReLU function. In the smoothed ReLU function, we set the threshold $\varrho$ to be $O(1/polylog k)$. Some correlation scores will increase and will be larger than the threshold $\varrho$, while some keep small in $O(1/poly k)$.  This process corresponds to the semantic learning process, and has been shown in Claim E.5 and Claim E.6 in our Appendix. It indicates that among those winning lottery tickets at initialization, some convolution kernels will win out and their correlation score will be larger than the threshold $\varrho$ after certain training iterations.
>
>
> For the question of why we set $\varrho$ to be $O(1/polylog k)$, this is mainly because of the size of $\mathcal{M} _ {i,l}$. At initialization, the size of the lottery ticket winning set $\mathcal{M} _ {i,l}$ is $O(polylog k)$ (see Lemma C.1 in Appendix). After pre-training, some kernels in $\mathcal{M} _ {i,l}$ will win out. Then in the downstream classification tasks, we add an additional linear layer in the pretrained encoder network. The linear combination of kernel weights which capture the features of $i$-th class will be approximately $\Theta(|\mathcal{M} _ {i,l}|\cdot \Lambda _ {i,l})\approx \tilde{\Theta}(1)$ while linear combinations of the remaining kernels will keep small in $\tilde{O}(1/polyk)$. This obviously leads to good performance in downstream classification tasks.
>
> **Q3. ... Is there any way we can quantitatively measure how many semantics in each class are captured by kernels?**
>
> **Reply:** To be honest, it is hard to quantitatively measure how many semantics in each class are captured by kernels, since 1) for real data, we do not know how many semantics a class contains; 2) there are so many kernels in the network and a kernel at most captures a single semantic.  Therefore, we use indirect methods, including the visualization and also test accuracy, which at least can justify more semantics captured by MRP over supervised learning.  We agree that a quantitative measure can make the comparison more transparent and better support our theoretical implication, and will continue to explore such a measure for further improving our work.
>
>
> **Q4. As you mentioned... Is that possible that we remove the nonlinear part and make it analyzable?**
>
> **Reply:** If we only remove the softmax function in attention, it becomes $XW_qW_k^TX^TXW_v/\sqrt{d_k}$. For this form, there still exists correlated manipulations, which is hard to analyze. Besides, there exist some experimental results when we remove the softmax in Transformer, the performance has a large degradation, which makes our theoretical analysis meaningless.
>
> **Q5. Some typos such as the pretraining data set as $\mathcal{Z}$ in Assumption1.**
>
> **Reply:**  Thanks. Here we denote the pretraining dataset as $\mathcal{Z}$, which consists of $N$ samples i.i.d. drawn from the data distribution $\mathcal{D}$. But we have checked through the paper to correct other typos.

---

### Decision · Program_Chairs · 2023-01-20

**Decision:**

Accept: poster

**Justification For Why Not Higher Score:**

This paper was quite dense and difficult for me (a proxy for a general audience) to understand. I do not think it will have sufficiently broad accessibility or appeal to the rest of the community to deserve a spotlight or oral.

**Justification For Why Not Lower Score:**

The reviewers universally agreed that this paper met the bar for ICLR and was technically sound.

**Metareview: Summary, Strengths And Weaknesses:**

**Summary:** This paper provides a theoretical prespective for understanding mask-reconstruction pretraining techniques that are popular for self-supervised computer vision pre-training. I am an empiricist and I won't pretend to fully understand the theoretical details of the paper, but the reviewers universally agreed that this paper was technically sound, novel, and an advance in our understanding of self-supervised learning, and I defer to their judgment.

**Note From Pc:**

if the above contains the word "oral" or "spotlight" please see: "oral" presentation means -> notable-top-5% and "spotlight" means -> notable-top-25%. As stated in our emails, we are disassociating presentation type from AC recommendations

**Summary Of Ac-Reviewer Meeting:**

N/A